

# Gravity from thermodynamics:
# Optimal transport and negative effective dimensions

**Giuseppe Bruno De Luca[1]⋆, Nicolò De Ponti[2]†,**
**Andrea Mondino[3]‡ and Alessandro Tomasiello[4]◦**

**1** Stanford Institute for Theoretical Physics, Stanford University,
382 Via Pueblo Mall, Stanford, CA 94305, United States
**2** International School for Advanced Studies (SISSA), Via Bonomea 265, 34136 Trieste, Italy
**3** Mathematical Institute, University of Oxford, Andrew-Wiles Building,
Woodstock Road, Oxford, OX2 6GG, UK
**4** Dipartimento di Matematica, Università di Milano–Bicocca, Via Cozzi 55,
20126 Milano, Italy; INFN, sezione di Milano–Bicocca

⋆ gbdeluca@stanford.edu , † ndeponti@sissa.it ,
‡ andrea.mondino@maths.ox.ac.uk , ◦ alessandro.tomasiello@unimib.it

## Abstract

We prove an equivalence between the classical equations of motion governing vacuum gravity compactifications (and more general warped-product spacetimes) and a concavity property of entropy under time evolution. This is obtained by linking the theory of optimal transport to the Raychaudhuri equation in the internal space, where the warp factor introduces effective notions of curvature and (negative) internal dimension. When the Reduced Energy Condition is satisfied, concavity can be characterized in terms of the cosmological constant $\Lambda$; as a consequence, the masses of the spin-two Kaluza-Klein fields obey bounds in terms of $\Lambda$ alone. We show that some Cheeger bounds on the KK spectrum hold even without assuming synthetic Ricci lower bounds, in the large class of infinitesimally Hilbertian metric measure spaces, which includes D-brane and O-plane singularities. As an application, we show how some approximate string theory solutions in the literature achieve scale separation, and we construct a new explicit parametrically scale-separated AdS solution of M-theory supported by Casimir energy.

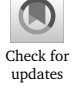

# 1 Introduction

The mathematical field of optimal transport was originally inspired by the concrete problem of how to best move a distribution of mass from one configuration to another. In recent years, this field has grown in various directions, incorporating ideas from Riemannian geometry and information theory.

In this paper, we apply ideas from this field to the physics of gravity, and in particular to its compactifications. At a technical level, these applications stem from the fact that the particular tensor

$$(\mathrm{Ric}_f^N)_{mn} := R_{mn} - \nabla_m \nabla_n f + \frac{1}{n-N} \nabla_m f \nabla_n f \tag{1}$$

plays a role in both contexts. In optimal transport, the function $f$ defines a measure $\sqrt{g}\mathrm{e}^f$, and $\mathrm{Ric}_f^N$ controls the distortion of measures along Wasserstein geodesics which, as we will see, describe mathematically the optimal way to transport probability distributions. While $n$ is the actual dimension of space, we will see that the number $N \in \mathbb{R} \cup \{\infty\}$ will play the role of an effective dimension, for reasons related to the Raychaudhuri equation. In gravity compactifications, $\mathrm{Ric}_f^N$ appears in the internal Einstein equations: $f$ is proportional to the warping function multiplying $d$-dimensional macroscopic spacetime, $n$ is the internal dimension, and $N = 2 - d$.

The fact that the effective dimension $N$ is often negative for compactifications might look unsettling at first. In an earlier paper [1] we had reorganized the equations of motion so as to use the $N = \infty$ limit $\mathrm{Ric}_f^\infty$, and exploted this fact to find applications of optimal transport to Kaluza-Klein masses (with some initial steps provided in [2]). However, recent mathematical work has shown that the $N < 0$ case also makes sense and is rich of geometric/analytic consequences [3–8]; in this paper we will see that it leads to cleaner and broader results.

Although our motivations come from the study of compactifications of higher dimensional gravitational theories, we stress that our results apply to general warped products in arbitrary number of dimensions, such as (warped) $1+n$ decompositions of static space-times.

## 1.1 Gravity and entropy

To a distribution of particles on a space $M_n$, we can associate a probability density distribution $\rho(x)$. If the total mass is $m$, the mass in a region of space $U$ is $m \int_U \mathrm{d}x \sqrt{g}\rho(x)$. One of the most intriguing results of optimal transport in curved spaces regards the behavior of the Shannon entropy

$$\mathcal{S}[\rho] = -\int_{M_n} \sqrt{g}\rho \log \rho \,,$$

for a distribution of particles that move geodesically. The *second* derivative of $\mathcal{S}$ with respect to time evolution turns out to be negative if and only if the ordinary Ricci tensor $R_{mn}$ is positive [9–12]. In other words, in this situation the entropy is concave as a function of time.[1]

The Einstein equations now imply an inequality relating this second derivative to an integral of the stress-energy tensor. This inequality becomes in fact *equivalent* to the Einstein equations if we add the information that it can be saturated on delta-like distributions.[2] This striking result has been rigorously proved for the Lorentzian vacuum Einstein equations in [14, 15]: [14] is focused on the Hawking-Penrose strong energy condition and time-like Ricci lower bounds, [15] treats general upper and lower time-like Ricci bounds and thus the Einstein equations. We rederive such an optimal transport characterization of Einstein's equations formally[3] in Sec. 4, using the tools of optimal transport we recall in Sec. 3.

Our description of this result in terms of $\frac{d^2 \mathcal{S}}{dt^2}$ is a bit of an oversimplification. In optimal transport one actually tends to focus on concavity rather than on the sign of $\frac{d^2 \mathcal{S}}{dt^2}$, because it

---

[1] The entropy is also known to be concave in the space of probability distributions, i.e. $\mathcal{S}[t\rho_0 + (1-t)\rho_1] \geqslant t\mathcal{S}[\rho_0] + (1-t)\mathcal{S}[\rho_1]$, $\forall \rho_0, \rho_1$ and $t \in [0,1]$.

[2] At its core, this is similar in spirit to the earlier observation in [13], where the Einstein equations were related to the behavior of a small sphere of particles in free fall.

[3] We use the adjective "formally" with its standard mathematical denotation, where it is contrasted to "rigorously".

makes sense even when $\mathcal{S}$ is not smooth, thus allowing to include in the treatment also spaces with singularities. This is important for physics applications, as we will see below.

As we mentioned earlier, a natural modification in this context is to introduce a weighted measure $\sqrt{g}e^f$ that differs from the standard Riemannian volume measure $\sqrt{g}$. Concavity of the Shannon entropy now becomes equivalent to positivity of $\mathrm{Ric}_f^\infty$. It is also natural to study other notions of entropy considered in the literature. As we review in Sec. 5.1, a famous possibility is the *Tsallis* entropy [16]

$$\mathcal{S}_\alpha[\rho] = \frac{1}{\alpha-1}\left(1 - \int_{M_n}\sqrt{g}e^f\rho^\alpha\right),$$

obtained by replacing one of the axioms characterizing the Shannon entropy (related to its extensive property) to a homogeneous property in terms of $\alpha$. Under the choice $\alpha = 1 - 1/N$, concavity of $\mathcal{S}_\alpha$ is related to positivity of $\mathrm{Ric}_f^N$ (cf. [17–19] for $N \in (1,\infty)$, and [5] for $N < 0$).

The aforementioned appearance of $\mathrm{Ric}_f^N$ in the internal Einstein equations suggests that they too might be reformulated in terms of generalized concavity properties for the Tsallis entropy $\mathcal{S}_{1-1/N}$, with weighted measure $\sqrt{g}e^f$. We establish this new result at the formal level in Sec. 5.3, leaving a fully rigorous mathematical proof to a later publication [20]. In Sec. 5.2 we also show that the external Einstein equation can be reformulated in terms of the *first* derivative of the ordinary Shannon entropy.

This reformulation also provides a rigorous mathematical definition of the low-energy Einstein equations for certain classes of singular space-times where the standard analytical geometrical definition breaks down. In Sec. 6 we showcase some of the advantages of this approach by proving rigorous theorems about the masses of the spin-two fluctuations around backgrounds that include localized classical sources, such as D$p$-brane singularities in supergravity. This suggests that this mathematical definition agrees at least partially with the UV completion of classical supergravity provided by string theory.

In physics, it is more customary to define an entropy by integrating a probability distribution in phase space. Our integrals over $M_n$ are entropies in the more general sense of information theory: they parameterize our ignorance about the position of particles that propagate geodesically on $M_n$. When the latter is the internal space of a compactification, our entropy measures the ignorance of a low-dimensional observer regarding a particle distribution along the internal dimensions.

There is of course a long history of connections between gravity and thermodynamical ideas, starting with black hole physics. A famous argument derives the Einstein equations from the assumption that the entropy is proportional to the area of any local Rindler horizon [21]. A later argument derived them by using the Ryu–Takayanagi formula [22] for holographic entanglement entropy [23]. An even more ambitious idea views gravity as an entropic force [24]. In contrast, we stress that our reformulation only uses *classical* physics of probe particles in free fall (that is, only subject to gravity). Nevertheless, it would be very interesting to investigate any relationship with those earlier results.

## 1.2  Bounds on KK masses and separation of scales

A notable and frequent application of global inequalities on the Ricci tensor is to obtain bounds on the eigenvalues of various geometrical operators, such as the Laplace–Beltrami. In gravity compactifications, the masses of Kaluza–Klein (KK) fields are also obtained as eigenvalues of geometrical operators. While unfortunately there is no general expression for those (other than in simple classes such as Freund–Rubin), an exception is the tower of spin-two masses, which is obtained from a version of Laplace–Beltrami weighted by the warping function (which is in turn proportional to the $f$ above).

In our reformulation of the Einstein equations, the second derivative of the entropy $\frac{d^2S}{dt^2}$ is related to an integral of a certain combination of the internal and external stress-energy tensor. This combination also appeared in [2], where it was shown to be positive for many common forms of matter; this was dubbed there the Reduced Energy Condition (REC).

This observation was already enough in [1, 2] to prove several upper and lower bounds on the spin-two masses. However, those results were using the $N = \infty$ effective dimension, and as a consequence many of the resulting bounds depended on the upper bound $\sup|dA|$ of the gradient of the warping function $A$. In some situations this can get large, and make the bounds less useful. The inequality we obtain here in terms of negative effective dimensions is much simpler:

$$(\text{Ric}_f^N)_{mn} \geqslant \Lambda g_{mn} \,, \tag{2}$$

with $f = (D-2)A$.

This makes no reference to the warping, and in turn improves the bounds on the spin-two KK masses. For example, we exploit and generalize a result in the literature [25] to show rigorously that in general the smallest mass satisfies (Th. 6.16):

$$m_1^2 \geqslant \frac{\alpha(\text{diam}\sqrt{-K})}{\text{diam}^2} \,, \tag{3}$$

where the diameter diam is the largest distance between any two internal points, and $\alpha(\text{diam}\sqrt{-K}) > 0$ is a constant that only depends on the product $\text{diam}\sqrt{-K}$, where $K < 0$ is a lower bound on the $N$-Ricci curvature. When $K = \Lambda = (1-d)/L_{\text{AdS}}^2$ as in (2), $\text{diam}\sqrt{-K} \propto \text{diam}/L_{\text{AdS}}$. For the applications, an important property of such a constant $\alpha$ is the limiting behaviour (Rem. 6.18):

$$\lim_{\text{diam}\sqrt{-K}\to 0} \alpha = \pi^2 \,.$$

In particular, this proves that in any warped compactification with matter content that satisfies the REC there is a large hierarchy between $m_1$ and the scale of the cosmological constant when $\text{diam}/L_{\text{AdS}}$ is small. The lower bound (3) is of course intuitive (at least at the qualitative level), but here we are providing a precise statement with a general rigorous argument, valid for any higher-dimensional gravity with an Einstein–Hilbert kinetic term. Optimal transport plays a key role in the proof of Th. 6.16, based on the so-called "$L^1$-localization method": the basic idea is that using $L^1$-optimal transport (i.e. optimal transport with cost function given by the distance function), it is possible to partition the (possibly singular) space $X$ (up to a set of measure zero) into geodesics $\{X_\alpha\}_\alpha$, each $X_\alpha$ being endowed with a Borel non-negative measure $\mathfrak{m}_\alpha$, and to reduce the proof of the desired inequality (3) in $X$ to proving a family of corresponding inequalities on the 1-dimensional weighted spaces $(X_\alpha, \mathfrak{m}_\alpha)$. Such a dimension reduction argument is very powerful, it has its roots in [26], it was formalised in highly symmetric spaces in [27, 28] by using iterative bisections, and was then developed via optimal transport tools for smooth Riemannian manifolds in [29] and for (possibly non-smooth) metric measure spaces satisfying synthetic Ricci lower bounds and dimensional upper bounds in [30].

As we mentioned above, optimal transport can handle certain singularities, called RCD spaces, by focusing on concavity properties for $\mathcal{S}$ rather than on its second derivative [17, 18, 31–34]; it turns out that this applies to the famous string theory objects called *D-branes*, as was checked in [1] for the $N \in (n, \infty]$ case and extended here to $N < 0$ (Sec. 6.4). As mentioned above, the advantage of considering negative $N$ here is that it allows for a neat control of the weighted $N$-Ricci curvature lower bound (2). So our proof of (3) applies to compactifications with brane singularities as well.

Some interesting compactifications of string theory also contain a type of source called *O-plane*. Unfortunately these turn out to be outside the RCD class, as we rigorously show in Sec. 6.4.1.[4] To handle them, we consider the broader class of *infinitesimally Hilbertian* metric measure spaces [32, 33]. We are able to show that some of the KK lower bounds appearing in [1] are also valid in this larger class, and hence apply to compactifications with O-plane singularities as well. In particular, we obtain (Thm. 6.6)

$$m_1^2 \geqslant h_1^2/4 \,, \tag{4}$$

where $h_1$ is the so-called *Cheeger constant* of $(M_n, f)$, which is small when the weighted manifold has a 'neck' almost separating it in two pieces (as reviewed at length in [1]). We also prove some higher order generalization of the previous result, obtaining bounds on the whole tower of spin-two masses (Thm. 6.7 and Thm. 6.8).

While the mathematical study of the $N < 0$ case is developed enough for us to obtain the results presented here, it is at present not as mature as the $N > 0$ and $N = \infty$ cases. Because of this, in this paper we have not improved the upper bound on $m_1^2$ mentioned in [1, Th. 4.2] and first [35, Cor. 1.2], itself a generalization of the so-called Buser inequality. We hope to return to this in the future, as mathematical techniques improve further.

We end the paper in Sec. 7 with some considerations about the problem of scale separation. First we discuss how the bounds (3) and (4) on $m_1$ can be used to show that this mass is much larger than $|\Lambda|$ for certain approximate string theory vacua [36, 37]. Second, we show how a simple violation of the REC, Casimir energy, can lead to such a scale separation as well, by constructing a AdS$_4 \times T^7$ solution with a parametric hierarchy between the KK modes and the scale of the cosmological constant. This background appears to be in tension with conjectures in the literature discussing the behavior of the KK spectrum as $|\Lambda| \to 0$, inviting further study.

## 2 Equations of motion and weighted Ricci tensor

We start by showing how the equations of motion for general warped-product space-times are naturally organized in terms of a generalization of the Ricci curvature tensor that better captures the geometrical properties of the space when the warping is non-trivial. The results in this section are partially based on the analysis of the equations of motion performed in [2, Sec. 2], to which we refer the reader for more details.

### 2.1 Effective curvature and dimension

Our setup consists of any possible gravitational theory that at low energy reduces to $D$-dimensional Einstein gravity, with some prescribed matter content. In particular, our analysis applies to compactifications of string and M-theory but it is not restricted to those.

We normalize the Einstein–Hilbert term in the action as $S_{\mathrm{EH}} = \frac{1}{\kappa^2} \int \sqrt{-g_D} R_D$ with $\kappa^2 = (2\pi)^{D-3} \ell_D^{D-2}$, where $\ell_D$ is the $D$-dimensional Planck length. In such a theory, the Einstein equations for the $D$-dimensional metric can be written as

$$R_{MN} = \frac{1}{2} \kappa^2 \left( T_{MN} - g_{MN} \frac{T}{D-2} \right) := \hat{T}_{MN} \,, \tag{5}$$

where $T_{MN} := -\frac{2}{\sqrt{-g}} \frac{\delta S_{\mathrm{mat}}}{\delta g^{MN}}$ is the stress-energy tensor of the $D$-dimensional theory. We are interested in studying general, possibly warped, $d$-dimensional vacuum compactifications. That

---

[4]Intuitively, this is due to geodesics being repelled by the singularity (due to its negative tension) and thus having to focus on it if one wants to hit an antipodal target. This is to be contrasted with D-branes, to which geodesics are attracted and thus spread out before refocusing.

is, the $D$-dimensional space-time has the form[5]

$$ds_D^2 = e^{2A}(ds_d^2 + ds_n^2). \tag{6}$$

The *warping* function $A$ only varies over the $n$-dimensional internal space; $ds_d^2$ is a maximally-symmetric space, with curvature normalized as $R_{\mu\nu}^{(d)} = \Lambda g_{\mu\nu}^{(d)}$.

Plugging (6) in (5), and specializing to external and internal directions, we get two sets of equations, which can be combined and re-organized as

$$\frac{1}{D-2}e^{-f}\Delta(e^f) = \frac{1}{d}\hat{T}^{(d)} - \Lambda, \tag{7}$$

$$R_{mn} - \nabla_m\nabla_n f + \frac{1}{D-2}\nabla_m f\nabla_n f = \Lambda g_{mn} + \tilde{T}_{mn}, \tag{8}$$

where we have defined the combinations

$$\hat{T}^{(d)} := g^{(d)\mu\nu}\hat{T}_{\mu\nu}, \qquad \tilde{T}_{mn} := \frac{1}{2}\kappa^2\left(T_{mn}^{(D)} - \frac{1}{d}g_{mn}T^{(d)}\right), \qquad f = (D-2)A. \tag{9}$$

Equation (8) highlights a particular combination of the Ricci tensor and the derivatives of the warping. For a given $N \in \mathbb{R}$ we can define the $N$-Bakry–Émery Ricci tensor

$$(\text{Ric}_f^N)_{mn} := R_{mn} - \nabla_m\nabla_n f + \frac{1}{n-N}\nabla_m f\nabla_n f, \tag{10}$$

and in terms of this object the equations of motion take the simple-looking form

$$\frac{1}{D-2}e^{-f}\Delta(e^f) = \frac{1}{d}\hat{T}^{(d)} - \Lambda, \tag{11a}$$

$$(\text{Ric}_f^{(2-d)})_{mn} = \Lambda g_{mn} + \tilde{T}_{mn}, \tag{11b}$$

where

$$N = 2 - d. \tag{12}$$

At this stage, the definition in (10) might appear purely algebraic and far from any geometrical meaning. However, crucially for the rest of our analysis, this generalization of the Ricci curvature tensor has already been considered and extensively studied in the literature of optimal transport, where (10) has been shown to be the notion of curvature that captures the analytic/geometric properties of *weighted* Riemannian manifolds with an effective notion of dimension $N < 0$ [3–6]. We will explore this more in detail in Sec. 3.4.

Finally, we stress that, even though our main motivation is the study of vacuum compactifications, all our analysis and results apply to any space-time that can be written in the form (6), including, for example, $1 + n$ splittings of static space-times.

## 2.2 Reduced energy condition and Ricci lower bounds

From (11b) we see that lower bounds on $\tilde{T}_{mn}$ directly translate to lower bounds on $(\text{Ric}_f^{(2-d)})_{mn}$, which we can then exploit to derive constraints on physical properties of vacuum compactifications.

At first sight, the peculiar combination of stress energy tensors appearing in the definition (9) for $\tilde{T}_{mn}$ can seem hard to estimate in general, as it might depend on the details of the energy sources. However, in [2] it has been noticed that for a large class of matter content

---

[5]We use upper case Latin letters to denote $D$-dimensional indices, lower-case Greek letters for indices along the directions of the $d$-dimensional vacuum and lower-case Latin letters for indices in the $n$-dimensional internal space, with $n = D - d$. Also, compared to references [1,2] we are suppressing here the bar on top of $g_n$ i.e. $\bar{g}_n^{\text{there}} \to g_n^{\text{here}}$.

it is actually non-negative. This condition, being like an energy condition naturally emerging from reducing the theory, has been named Reduced Energy Condition (REC):[6]

$$\tilde{T}_{mn} = \frac{1}{2}\kappa^2\left(T_{mn}^{(D)} - \frac{1}{d}g_{mn}T^{(d)}\right) \geqslant 0\,, \qquad \text{REC.} \tag{13}$$

More precisely, [2] has shown that the REC is satisfied by higher dimensional scalar fields, general $p$-form fluxes (including 0-forms such as the Romans mass in type mIIA) and localized sources with positive tension. Moreover, one can easily check that any potential for a collection fields $\varphi_i$ of the form $S^{\text{pot.}} = \int \sqrt{-g_D}\,V(\{\varphi_i\})$, with $V(\{\varphi_i\})$ independent of the metric, does not affect the REC since its contribution cancels from the combination (13).

When the REC is satisfied by the matter content of the $D$-dimensional theory, we have a simple lower bound for the synthetic curvature:

$$\text{Ric}_f^{(2-d)} \geqslant \Lambda\,, \tag{14}$$

which we can exploit to bound physical properties of gravity compactifications in terms of $\Lambda$, as we will do in Sec. 6 where we bound the masses of spin-two Kaluza-Klein fluctuations around general vacua that satisfy the REC. However, many interesting physical sources violate the REC, such as O-planes in string theory or quantum effects. In Sec. 7 we will analyze explicit examples in which these sources allow the construction of scale-separated solutions, i.e. solutions for which the masses of the Kaluza-Klein modes are parametrically larger than $|\Lambda|$.

## 3 Optimal transport

As we anticipated, the tensor (10) appearing in the equations of motion has a natural interpretation in the field of optimal transport. In this section we will give a brief review of some aspects of this field that are important for the rest of the paper. In particular we will show why (10) is a natural combination, and in what sense $N$ can be considered an effective dimension. In this section we will mostly consider smooth (weighted) spaces, while the non-smooth case will be treated later in Section 6.

### 3.1 Probability and optimal transport

Take a distribution of probe particles on a *space X* that at time $t = 0$ has a certain shape $\mu_0(x)$. We can think of it as an actual mass distribution of many particles or as a probability distribution for a single particle; in both cases we normalize $\int_X \mu_0(x) = 1$. Assume that on $X$ there is a notion of distance and that each bit of the distribution starts moving along a geodesic. The initial shape $\mu_0$ is thus being distorted, and we call $\mu(x, t)$ the probability distribution at $t \geqslant 0$, with $\mu(x, 0) := \mu_0(x)$. Since the individual bits of mass are moving along geodesics, certainly $\mu(x, t)$ has some information about the geometry of $X$; we may wonder if it knows enough, for example to give information about the curvature of $X$. The answer turns out to be affirmative: specifying the evolution of the *relative entropy* $\mathcal{S}[\mu]$ of $\mu$ with respect to the volume form on $X$ is *equivalent* to prescribing the Ricci tensor. This observation can be then exploited to encode all the Einstein equations in an evolution equation for $\mathcal{S}[\mu]$; cf. [14, 15].

To explicitly derive and state this equivalence we need tools to handle the evolution of probability distributions. Luckily, these have been extensively developed in the context of optimal transport theory. Most of the informal discussion below is based on [19, Chapp. 6,

---

[6]Unlike most energy conditions, the REC only makes sense for compactifications. Another such condition, dubbed IEC, was considered in [38, (2.32)], but it only concerns traces of the stress-energy tensor.

14, 15, 16]: we adopt the formalism introduced by Otto [39], now know as *Otto calculus*; cf. [40, Sec. 2.2] and [1, Sec. 2.3].

To make contact with this framework, assume we are on a more general metric space $(X, \mathrm{d})$ and that we are given the task of moving mass on $X$ in order to morph an initial distribution $\mu_0(x)$ into a distribution $\mu_1(y)$, while minimizing the total cost, when the cost of moving one bit of mass from $x$ to $y$ is given by the squared distance $\mathrm{d}(x, y)^2$. This problem induces a distance on the space $\mathcal{P}_2(X)$ of Borel probability measures over $X$ with finite second moment, called 2-Kantorovich–Wasserstein, or *Wasserstein distance* for short:

$$W_2(\mu_0, \mu_1) := \inf \sqrt{\left\{ \iint_{X \times X} \mathrm{d}(x, y)^2 \, \mathrm{d}\pi(x, y) : \ \pi \text{ coupling of } \mu_0 \text{ and } \mu_1 \right\}}, \qquad (15)$$

where a *coupling* is a probability distribution $\pi$ on $X \times X$ whose *marginals* $\int_X \mathrm{d}y \, \pi(x, y)$ and $\int_X \mathrm{d}x \, \pi(x, y)$ are respectively equal to $\mu_0(x)$ and $\mu_1(y)$. These requirements impose that all the mass that is going to $y$ comes from $\mu_0$ and that all the mass moved out from $x$ goes into $\mu_1$, i.e. no mass has been lost or created in the process.

When $X$ is an $n$-dimensional Riemannian manifold equipped with a metric $g$, which induces the distance $\mathrm{d}$ and an invariant measure $\sqrt{g}\mathrm{d}x^n$, we can formally think of $\mathcal{P}_2(X)$ as an infinite dimensional Riemannian manifold equipped with a scalar product that induces the distance (15). We can write this metric in terms of its action on tangent vectors $\dot{\mu}$ on $\mathcal{P}_2(X)$, which are characterized as follows. Since the total mass is being preserved in this process, a continuity equation holds on $X$, and we can specify $\dot{\mu}$ at a given time $t$ in terms of a vector field $\xi(\cdot, t) \in T(X)$ as

$$\dot{\mu} := -\nabla \cdot (\mu \xi) := -\nabla \cdot (\mu \nabla \eta). \qquad (16)$$

The time-dependent vector field $\xi(x, t)$, which describes the direction on $X$ along which the bit of mass at $x$ is moving at time $t$, can be written as the gradient of a real function $\eta(\cdot, t)$. With this definition, it can be shown that the Wasserstein distance can be represented as

$$W_2(\mu_0, \mu_1) = \inf_{\{\mu(t)\}_{t \in [0,1]}} \sqrt{\int_0^1 \mathrm{d}t \int_X \mu |\xi|^2}, \qquad \text{with} \quad \mu(0) = \mu_0, \quad \mu(1) = \mu_1. \qquad (17)$$

Thus, given two tangent vectors (on $\mathcal{P}_2(X)$) at $\mu$, $\dot{\mu}_1$ and $\dot{\mu}_2$, their scalar product is

$$\langle \dot{\mu}_1, \dot{\mu}_2 \rangle_W := \int_X \mu \, \xi_1 \cdot \xi_2. \qquad (18)$$

This defines our formal Riemannian metric on $\mathcal{P}_2(X)$.

Now that we have tangent vectors and a Riemannian metric on $\mathcal{P}_2(X)$, we can ask when the curve $\mu(\cdot, t) : [0, 1] \to \mathcal{P}_2(X)$ is a geodesic with respect to the metric (18), i.e. when it locally minimizes the distance (15). The answer is well-known: such geodesics are characterized in terms of solutions of the Hamilton–Jacobi equation on $X$. Specifically, $\mu(x, t)$ describes a geodesics in $\mathcal{P}_2(X)$ if

$$\dot{\eta} = -\frac{|\xi|^2}{2} = -\frac{|\nabla \eta|^2}{2}, \qquad (19)$$

with $\xi$ and $\eta$ defined from $\dot{\mu}$ through the continuity equation (16). In Appendix C we show how (19) implies that each bit of mass composing $\mu$ moves along a geodesics on $X$.

## 3.2 Derivatives of functionals in Wasserstein space

Equipped with the formal machinery developed in the previous section, we are now ready to compute the derivative of functionals on $X$ along geodesics on $\mathcal{P}_2(X)$. From now on, we

will perform formal computations specializing to the cases in which $X$ is an $n$-dimensional Riemannian manifold equipped with a metric $g$, which induces the distance d and an invariant measure $\sqrt{g}dx^n$. We will return to singular spaces in Sec. 6, where we show that one can rigorously take into account the classical singular backreactions of physical sources.

Given a probability distribution $\mu$, we define its density $\rho$ as

$$\mu := \rho\sqrt{g}dx^n, \tag{20}$$

and represent a generic functional $\mathcal{F}$ as[7]

$$\mathcal{F}[\mu] := \int_X \sqrt{g}F(\rho). \tag{21}$$

When $\mu$ changes in time, so will $\mathcal{F}[\mu]$, and an explicit computation reveals that along the curve described by the continuity equation (16) its rate of change is

$$\frac{\mathrm{d}}{\mathrm{d}t}\mathcal{F}[\mu] = \int_X \sqrt{g}P(\rho)\Delta\eta, \tag{22}$$

where we have defined $P(\rho) := \rho F'(\rho) - F(\rho)$. We can go further and compute the second derivative along the curve $\mu(\cdot, t)$, taking into account that this curve is a geodesic on $\mathcal{P}_2(X)$ and thus imposing (19). Doing so we get

$$\frac{\mathrm{d}^2}{\mathrm{d}t^2}\mathcal{F}[\mu] = -\int_X \sqrt{g}P(\rho)\left[\Delta\left(\frac{|\nabla\eta|^2}{2}\right) - \nabla(\Delta\eta)\cdot\nabla\eta\right] + \int_X \sqrt{g}P_2(\rho)(\Delta\eta)^2, \tag{23}$$

where we defined $P_2(\rho) := \rho P'(\rho) - P(\rho)$ and $\Delta := -\nabla^2$. More details on the derivations of (22) and (23) can be found in App. B. We can simplify the quantity in the brackets by using the Bochner formula

$$-\left[\Delta\left(\frac{|\nabla\eta|^2}{2}\right) - \nabla(\Delta\eta)\cdot\nabla\eta\right] = R_{mn}\nabla^n\eta\nabla^m\eta + \nabla_m\nabla_n\eta\nabla^m\nabla^n\eta. \tag{24}$$

In Appendix A we review how (24) is a close relative of the Raychaudhuri equation on $X$, which connects the behavior of families of geodesics to the Ricci curvature. Plugging it in (23) we can finally write

$$\frac{\mathrm{d}^2}{\mathrm{d}t^2}\mathcal{F}[\mu] = \int_X \sqrt{g}P(\rho)[R_{mn}\nabla^n\eta\nabla^m\eta + \nabla_m\nabla_n\eta\nabla^m\nabla^n\eta] + \int_X \sqrt{g}P_2(\rho)(\Delta\eta)^2. \tag{25}$$

Equations (22) and (25) are the main ingredients we need to rewrite the Einstein equations in terms of derivatives of an entropy.

We conclude this section by noticing that knowledge of time derivatives of function(als) along geodesics can be used to extract their spatial derivatives, i.e. gradients and Hessians. Indeed, consider first the finite-dimensional case of a function $F : M \to \mathbb{R}$, with $M$ a smooth manifold. We can extract the gradient of $F$ at $x_0$ along the direction $\xi_0$ by evaluating the derivative of $F$ along a curve that at $t = 0$ passes through $x_0$ with tangent vector $\xi_0$. Indeed, $\frac{\mathrm{d}}{\mathrm{d}t}F(x(t)) = \langle \dot{x}^i, \mathrm{grad}F \rangle_g$, where $\mathrm{grad}F$ is the gradient vector field of $F$. Evaluating it at $t = 0$ results in the expression

$$\left.\frac{\mathrm{d}}{\mathrm{d}t}\right|_{t=0}F(x(t)) = \langle \xi_0, \mathrm{grad}F(x_0) \rangle_g. \tag{26}$$

---

[7]For simplicity of notation we suppress $\mathrm{d}x^n$ from integrals in the remainder of this section.

In a similar way we can extract the Hessian. This time we need the second derivative of $F$ along $x(t)$, when the latter describes a geodesic on $M$. This gives $\frac{d^2}{dt^2}F(x(t)) = \ddot{x}^i \partial_i F + \dot{x}^i \dot{x}^j \partial_i \partial_j F = \dot{x}^i \dot{x}^j \nabla_i \nabla_j F$, which evaluated at $t = 0$ results in

$$\frac{d^2}{dt^2}\bigg|_{t=0} F(x(t)) = \langle \xi_0, \mathrm{Hess} F(x_0)\xi_0 \rangle_g. \tag{27}$$

Formally, the same relations are true in the Wasserstein space $\mathcal{P}_2(X)$, so that (22) and (25) evaluated at $t = 0$ represent the gradient and the Hessian of $\mathcal{F}$ at $\mu$ along the direction $\dot{\mu}$.

## 3.3 Weighted measures

The discussion of the previous section can be generalized to the case in which the Riemannian volume form is weighted by a positive function. In this situation the measure that equips our metric-measure space is a more general $e^f \sqrt{g}$.[8] Given a probability distribution $\mu$ on $X$ it is then more natural to define its density $\rho$ with respect to the weighted volume form as

$$\mu := \rho \sqrt{g} e^f dx^n. \tag{28}$$

We can then represent a generic functional $\mathcal{F}$ as

$$\mathcal{F}[\mu] := \int_X \sqrt{g} e^f F(\rho). \tag{29}$$

When $\mu$ changes in time according to the continuity equation (16) the derivative of $\mathcal{F}$ is given by

$$\frac{d}{dt}\mathcal{F}[\mu] = \int_X \sqrt{g} e^f P(\rho)\Delta_f(\eta), \tag{30}$$

which differs from the unweighted case (22) by the fact that the integral is weighted and the Laplacian is replaced by the weighted Laplacian

$$-\Delta_f \eta := e^{-f}\nabla^m(e^f \nabla_m \eta) = \nabla^2 \eta + \nabla f \cdot \nabla \eta := \nabla_f^2 \eta. \tag{31}$$

Equation (30) and the ones that follow are derived in App. B. Taking another derivative and using (19) to evaluate the resulting expression along geodesics in Wasserstein space, after some manipulations we get:

$$\frac{d^2}{dt^2}\mathcal{F}[\mu] = -\int_X \sqrt{g} e^f P(\rho)\left[\Delta_f\left(\frac{|\nabla\eta|^2}{2}\right) - \nabla(\Delta_f(\eta))\cdot\nabla\eta\right] + \int_X \sqrt{g} e^f P_2(\rho)(\Delta_f(\eta))^2. \tag{32}$$

To simplify the term in square brackets in (32), we need the weighted analogue of the Bochner equation (24):

$$-\left[\Delta_f\left(\frac{|\nabla\eta|^2}{2}\right) - \nabla(\Delta_f(\eta))\cdot\nabla\eta\right] = (R_{mn} - \nabla_m\nabla_n f)\nabla^n\eta\nabla^m\eta + \nabla_m\nabla_n\eta\nabla^m\nabla^n\eta. \tag{33}$$

All in all, in the weighted case, for the second derivative of a generic functional of the form (29) along a geodesic in Wasserstein space we have:

$$\frac{d^2}{dt^2}\mathcal{F}[\mu] = \int_X \sqrt{g} e^f P(\rho)[(R_{mn} - \nabla_m\nabla_n f)\nabla^n\eta\nabla^m\eta + \nabla_m\nabla_n\eta\nabla^m\nabla^n\eta]$$
$$+ \int_X \sqrt{g} e^f P_2(\rho)(\Delta_f\eta)^2. \tag{34}$$

---

[8]Also in the weighted case we are describing the theory in the smooth setting for simplicity but the results can be shown to hold in more general metric measure spaces.

In the next section we will obtain a physical picture of the Bochner identities by relating them to Raychaudhuri equations and highlighting the effect of the weight function in introducing an effective notion of curvature of dimension.

### 3.4 Effective dimension

Let us now focus on the term $\nabla_m \nabla_n \eta \nabla^m \nabla^n \eta = \nabla_m \xi_n \nabla^m \xi^n$ appearing in both (25) and (34). As we noticed already, the origin of these terms is from the Bochner or the related Raychaudhuri equations (App. A). We can bound this term by using the inequality

$$\mathrm{Tr}M^2 \geqslant \frac{1}{n}(\mathrm{Tr}M)^2\,, \tag{35}$$

which follows from the Cauchy–Schwarz inequality by considering the inner product of $M$ and $1_n$ in the space of $n$-dimensional matrices. In particular we get

$$\nabla_m \nabla_n \eta \nabla^m \nabla^n \eta \geqslant \frac{1}{n}(\nabla^2 \eta)^2\,. \tag{36}$$

Using this, the Raychaudhuri equation becomes

$$-\nabla_\xi \theta \geqslant R_{mn}\xi^m \xi^n + \frac{1}{n}\theta^2\,, \tag{37}$$

where recall $\theta = \nabla_m \xi^m$ is the expansion. In many physics applications, actually the bound is even more stringent. The matrix $M_{mn} = \nabla_m \nabla_n \eta = \nabla_m \xi_n$ can have rank $r < n$; we can apply Cauchy–Schwarz to $M$ and the projector orthogonal to $\ker M$, which results in

$$\frac{1}{n} \rightarrow \frac{1}{r}\,, \tag{38}$$

in both (35) and (37). For example, in Lorentz signature, for timelike geodesics the matrix $\nabla_m \xi_n$ is orthogonal to $\xi$ itself, so it has rank $r = n-1$. For lightlike geodesics, $r = n-2$.

We can achieve an even more dramatic change in dimension. By using the identity

$$\frac{x_1^2}{a_1} - \frac{x_2^2}{a_2} + \frac{(x_1 + x_2)^2}{a_2 - a_1} = \frac{1}{a_2 - a_1}\left(\sqrt{\frac{a_2}{a_1}}x_1 + \sqrt{\frac{a_1}{a_2}}x_2\right)^2\,, \tag{39}$$

with $a_1 = -N$, $a_2 = n-N$, $x_1 = -\nabla_f^2 \eta$, $x_2 = \nabla f \cdot \nabla \eta$, we obtain

$$\frac{1}{n}(\nabla^2 \eta)^2 - \frac{1}{N}(\nabla_f^2 \eta)^2 - \frac{1}{n-N}(\nabla f \cdot \nabla \eta)^2 = -\frac{n-N}{nN}\left(\nabla^2 \eta + \frac{n}{n-N}\nabla f \cdot \nabla \eta\right)^2\,. \tag{40}$$

For $N < 0$ the right-hand side is positive. Combining this information with (36), we can bound the expression appearing in the first line of (34) and in the right-hand side of the weighted Bochner identity (33) as follows:

$$(R_{mn} - \nabla_m \nabla_n f)\xi^m \xi^n + \nabla_m \xi_n \nabla^m \xi^n \left(R_{mn} - \nabla_m \nabla_n f + \frac{1}{n-N}\nabla_m f \nabla_n f\right)\xi^m \xi^n + \frac{1}{N}(\nabla_f^2 \eta)^2\,.$$

The first term on the right-hand side is $(\mathrm{Ric}_f^N)_{mn}$ as defined in (10), thus explaining its relevance in optimal transport. In particular the weighted Raychaudhuri (A.11) now implies

$$-\nabla_\xi \theta_f \geqslant (\mathrm{Ric}_f^N)_{mn}\xi^m \xi^n + \frac{1}{N}\theta_f^2\,,$$

with $\theta_f := -d_f^\dagger \xi$ as defined in (A.10). Comparing with (37), we see that the dimension $n$ has now been replaced by $N$, which can thus be thought of as an *effective dimension*.

In other words, $N$ plays the role usually played by $d-1 = 3$ for massive geodesics or $d-2 = 2$ for massless geodesics in applications of the Raychaudhuri equation to $d = 4$ general relativity.

## 4 Shannon entropy and Einstein equations

The Einstein equations can be equivalently rewritten in terms of concavity properties of appropriate entropy functionals $\mathcal{S}$ defined on space-time:

$$\text{Ric} = T \qquad \Longleftrightarrow \qquad \mathrm{d}_t^2\mathcal{S} \leqslant -T\,, \quad \text{with saturated inequality in the limit for measures}$$
$$\text{concentrated towards Dirac deltas.} \tag{41}$$

To show the equivalence, we apply the methods in Sec. 4.1. We will first consider $1+n$ space-times and unwarped compactifications, where (41) will take the form of Theorem 4.1. We then review in Sec. 4.2 the more general Lorentzian case (Th. 4.2), before addressing general warped compactifications in Sec. 5.

### 4.1 Time+space and unwarped compactifications

The analysis that follows is a formal re-derivation of the results rigorously proved in [10–12] (respectively in [41]) regarding the optimal transport characterization of lower (resp. upper) bounds on the Ricci curvature for smooth Riemannian manifolds.

Given a probability distribution $\mu$ with density $\rho$ as defined in (20), we can compute its Shannon entropy

$$\mathcal{S} := \mathcal{S}[\mu|\sqrt{g}] = -\int_X \sqrt{g}\rho \ln\rho + \gamma\,, \tag{42}$$

where $\gamma$ is a normalization constant. Def. (42) can also be interpreted as relative entropy between $\mu$ and the uniform distribution on $X$, where "uniform" has to be defined with respect to the volume to have a coordinate-independent meaning. We will use these two denominations for the entropy interchangeably. In any case, (42) measures how spread out $\mu$ is compared to $\sqrt{g}$. Indeed, (42) reaches its maximum for the uniform distribution while approaching $-\infty$ for a very localized $\mu$ approaching a delta distribution.

Specializing (25) to $F = -\rho \ln\rho$, we then have an expression for the time evolution of $\mathcal{S}$:

$$\frac{\mathrm{d}^2}{\mathrm{d}t^2}\mathcal{S} = -\int_X \sqrt{g}\rho\left[R_{mn}\nabla^n\eta\nabla^m\eta + \nabla_m\nabla_n\eta\nabla^m\nabla^n\eta\right]\,, \tag{43}$$

which we will use to obtain the Einstein equations from this notion of entropy.

While the discussion so far focused on the Riemannian case, where particles are transported along Riemannian geodesics, and thus it does not describe general gravitational systems, it is nevertheless sufficient to completely characterize the Einstein equations for $D$-dimensional product space-times of the form $\mathcal{M}_D = M_d \times X_n$, with product metrics

$$\mathrm{d}s_D^2 = \mathrm{d}s_\Lambda^2 + \mathrm{d}s_n^2\,, \tag{44}$$

where $M_d$ is a $d$-dimensional vacuum ($\text{AdS}_d$, $\text{Mink}_d$, $\text{dS}_d$) with cosmological constant $\Lambda$ and $X_n$ is an $n$-dimensional space. Here $d \geqslant 1$, with the $d = 1$ case corresponding to an $1 + n$ decomposition of the $D$-dimensional space-time:

$$\mathrm{d}s_D^2 = -\mathrm{d}t^2 + \mathrm{d}s_n^2\,. \tag{45}$$

In situations like these, the Riemannian geodesics on $X_n$ can immediately be lifted to geodesics on $\mathcal{M}_D$, either massive or massless, upon an appropriate identification between the "time" coordinate along the Riemannian geodesic and a local time-coordinate on $M_d$. The Riemannian

formalism developed so far thus applies directly to massive and massless particles on product space-times, and we use this simplified scenario as a first illustration of the how the Einstein equations follow from entropy concavity, before reviewing the general Lorentzian case in Sec. 4.2 and the extending to general warped products in Sec. 5.

The $D$-dimensional Einstein equations specialized to space-times of the form (44) are (7) (8) with $f = 0$, which read:

$$\Lambda = \frac{1}{d}\hat{T}^{(d)}, \tag{46}$$

$$R_{mn} = \Lambda g_{mn} + \tilde{T}_{mn} := \hat{T}_{mn}, \tag{47}$$

where $R_{mn}$ is the Ricci tensor on $X_n$.

Equation (46) determines the $d$-dimensional cosmological constant, and our goal is to obtain (47) as a concavity equation for $\mathcal{S}$. If we think of a situation like (44) as a compactification on $X_n$ (or a more general reduction of the higher dimensional gravitational theory), $\mathcal{S}$ has a natural interpretation as a quantification of the ignorance of a lower-dimensional observer about the internal degrees of freedom. Indeed, classically, a $d$-dimensional observer can localize a $D$-dimensional particle approximately as a point on $M_d$, but they cannot do the same on $X_n$ if the Kaluza-Klein scale is much smaller than the energies they are able to probe; they will describe such a particle in terms of a probability distribution $\mu$ on $X_n$, with $\mathcal{S}$ quantifying their uncertainty about the internal position. Similarly, not being able to measure the masses of the KK excitation beyond the compactification scale, a lower-dimensional observer can reconstruct a higher-dimensional scalar field only up to a probability distribution in the internal space.

Crucially, the lower-dimensional observer need not to be aware of the gravitational nature of the sector they cannot probe to be able to characterize it completely. Indeed, the internal Einstein equations can be traded completely for an evolution equation for the information the lower dimensional observer has about the system, as in the following

**Theorem 4.1.** *Let $(X, g)$ be a smooth Riemannian manifold. The following statements are equivalent:*

1. *The metric $g$ on $X$ satisfies the equation of motion*

$$R_{mn} = \hat{T}_{mn}. \tag{48}$$

2. *i) For any probability distribution $\mu$ on $X$, evolving along a geodesic in the space $\mathcal{P}_2(X)$ of probability distributions, with tangent vector $\dot{\mu} = -\nabla \cdot (\mu \nabla \eta)$, its Shannon entropy (42)(the relative entropy between $\mu$ and the volume form of $g$) satisfies*

$$\left.\frac{d^2}{dt^2}\right|_{t=0} \mathcal{S} \leqslant -\int_X \mu \, \hat{T}_{mn} \nabla^m \eta \nabla^n \eta. \tag{49}$$

*ii) In addition, the inequality (49) becomes saturated whenever $\mu$ is concentrated at a point, and for a suitably chosen $\eta$. Namely, for any point $x_0 \in X$ and any tangent vector $\xi_0$ at $x_0$, there exists an $\eta$ such that $\nabla\eta|_{x_0} = \xi_0$ and such that (49) becomes an equality asymptotically for distributions $\mu$ very localized at $x_0$.*

*More precisely, for every point $x_0 \in X$ there exists a function $\omega$ with $\lim_{\varepsilon \to 0} \omega(\varepsilon) = 0$ such that the following holds: for any tangent vector $\xi_0$ at $x_0$ of unit norm $\|\xi_0\| = 1$, there exists a smooth function $\eta$ with $\nabla\eta|_{x_0} = \xi_0$ such that*

$$\left| \frac{d^2}{dt^2}\Big|_{t=0} \mathcal{S} + \int_X \mu \hat{T}_{mn} \nabla^m \eta \nabla^n \eta \right| \leq \omega(\varepsilon),$$

*for every probability measure $\mu$ supported in $B_\varepsilon(x_0)$.*

*Proof.* $1 \Rightarrow 2$: We plug (48) in (43), obtaining

$$\frac{\mathrm{d}^2}{\mathrm{d}t^2}\bigg|_{t=0} \mathcal{S} = -\int_X \mu \left[ \hat{T}_{mn} \nabla^n \eta \nabla^m \eta + \nabla_m \nabla_n \eta \nabla^m \nabla^n \eta \right]. \tag{50}$$

Then i) follows from the fact that $\nabla_m \nabla_n \eta \nabla^m \nabla^n \eta$ is non-negative. For ii), in the limit where $\mu \to \delta(x - x_0)$ the integral (50) localizes at $x = x_0$; using Lemma B.1 with $f = 0$ the second term vanishes and we get the result.

$2 \Rightarrow 1$: Combining (49) and (43) we obtain

$$\int_X \mu \left( E_{mn} \nabla^m \eta \nabla^n \eta + \nabla_m \nabla_n \eta \nabla^m \nabla^n \eta \right) \geqslant 0, \tag{51}$$

where we wrote (48) as $E_{mn} = 0$. Again, in the limit $\mu \to \delta(x - x_0)$ the integral (51) localizes at $x_0$. For an arbitrary tangent vector $\xi_0$ at $x_0$, using Lemma B.1 we then get $E_{mn} \xi_0^m \xi_0^n \geqslant 0$. Since $x_0$ and $\xi_0$ are arbitrary this implies

$$E_{mn} \geqslant 0. \tag{52}$$

Then, since by hypothesis for $\mu$ localized at any $x_0$ the inequality can be saturated by a certain $\eta$ such that $\nabla \eta(x_0) = \xi_0$, with arbitrary $\xi_0$, for such a choice of $\eta$ (51) implies

$$\left( E_{mn} \xi_0^m \xi_0^n + \nabla^m \xi_0^n \nabla_m \xi_{0n} \right)(x_0) = 0. \tag{53}$$

But from (52) both terms are non-negative, so for the equality to hold they have to vanish independently. Arbitrariness of $x_0$ and $\xi_0$ then ensures $E_{mn} = 0$. □

## 4.2 Einstein's equations in Lorentzian manifolds

In this section we show how also in Lorentzian signature the Einstein equations can be rewritten in terms of concavity properties of entropy functionals, characterizing in this way the whole-space time (and not just the internal part, as in vacuum compactifications). The analysis that follows is a formal re-derivation of the results rigorously proved in [14, 15].

On the whole $D$-dimensional Lorentzian space-time,[9] we seek to reproduce the Einstein equations, in the form (5)

$$R_{MN} = \hat{T}_{MN}, \tag{54}$$

from a concavity property of an appropriate notion of entropy for test particles. Since there is no analogue of warping, it is natural to guess that the relevant quantity will be the Shannon entropy, similarly to the unwarped Riemannian analysis of Sec. 4.1. This guess will turn out to be correct, but an important difference due to the signature will arise in the need to restrict the transport only along physical geodesics. In the following we will focus on the massive (time-like) case. In addition, even in this class, it is not guaranteed the squares of tensors appearing in the expression have definite sign (so that they can be discarded in the derivation of inequalities) and this technical difference will require us to carefully define the transport by switching to a more general non-linear framework. Let us see in practice how this works.

Given a time-like curve $\gamma : [0, 1] \to M$, define the $A_p$ actions

$$A_p[\gamma] := -\frac{1}{p} \int_0^1 \mathrm{d}\sigma \left( -g_{MN} \dot{\gamma}^M \dot{\gamma}^N \right)^{p/2}, \tag{55}$$

where $p \in (0, 1)$. The *cost* of moving a particle from $x$ to $y$ is then defined to be

$$c_p(x, y) := \inf \left\{ A_p[\gamma] \mid \gamma(0) = x, \gamma(1) = y \right\}. \tag{56}$$

---

[9] We can take $D = 4$ if we are working in a 4-dimensional Einstein theory or $D = 10, 11$ for string/M-theory.

The minus sign in (55) is introduced so that the cost (56) can still be formulated as a minimization problem. This is just as for the usual particle action in curved space, where $S/m = -\tau = -\int d\sigma \left(-g_{MN}\dot{\gamma}^M\dot{\gamma}^N\right)^{1/2}$, which would be recovered for $p \to 1$. Albeit here we are restricting only to $p \in (0,1)$, since for these values of $p$ the $A_p$ actions will have good convexity properties that we can exploit in our derivations, this technical choice does not change the physical picture. Indeed the extremizers of $A_p$ for $p \in (0,1)$ coincide with the extremizers of $A_1 = -\tau$ parametrized such that the tangent vector along a geodesic is parallel transported. This is similar to how in the Riemannian case extremizers of the energy functional $E[\gamma] := \int_0^1 |\dot{\gamma}|^2$ coincide with extremizers of the length functional $L[\gamma] := \int_0^1 |\dot{\gamma}|$ for a preferred parametrization of the coordinate along $\gamma$.

As in the definition of Wasserstein distance (15), we can lift the notion of cost (56) for moving massive particles in the space-time to a notion of cost for moving distributions of massive particles, by defining the family of functionals

$$\mathcal{C}_p(\mu_0, \mu_1) := \inf\left\{ \int_{M \times M} c_p(x, y)\, d\pi(x, y) \ : \ \pi \text{ coupling of } \mu_0 \text{ and } \mu_1 \right\}, \tag{57}$$

where, as in the Riemannian case (15), a *coupling* is a probability distribution $\pi$ on $M \times M$ whose marginals are equal to $\mu_0$ and $\mu_1$, which are Borel probability measures with compact support, $\mu_0, \mu_1 \in \mathcal{P}_c(M)$ in short.

Notice that $(-\mathcal{C}_p(\mu_0, \mu_1))^{1/p}$ is non-negative and satisfies a reverse triangle inequality; thus, in a broad sense, it is lifting the Lorentzian distance from $M$ to $\mathcal{P}_c(M)$. This kind of $p$-Lorentz-Wasserstein distances have been studied in [14, 15, 42–44].

The non-linearities introduced by the choice $p \neq 1$ will enter in the various expressions governing the evolution of a generic probability distribution $\mu$ through a non-linear redefinition of the gradient. Specifically, in terms of the conjugate exponent $q$ to $p$:

$$\frac{1}{p} + \frac{1}{q} = 1, \tag{58}$$

we define the $q$-gradient of a function $h$ with time-like gradient as:

$$\nabla_M^q h := -|\nabla h|^{q-2}\nabla_M h, \quad \text{with} \quad |\nabla h| := \sqrt{-g_{MN}\nabla^N h \nabla^N h}. \tag{59}$$

Then, the continuity (16) and geodesic (19) equations are modified, respectively, as

$$\dot{\mu} = -\nabla \cdot (\mu \nabla^q \eta), \qquad \dot{\eta} = -\frac{1}{q}|\nabla \eta|^q. \tag{60}$$

With these tools we can now compute derivatives of functionals along massive geodesics. The derivation is technically more involved as a consequence of the non-linearity, and we quickly sketch the relevant formulas in App. D.

Given a probability distribution of massive particles $\mu$ on a space-time $M$, we define its Shannon entropy to be

$$\mathcal{S} := \mathcal{S}[\mu|\sqrt{-g}] = -\int_M \sqrt{-g}\,\rho \ln \rho, \quad \text{with} \quad \mu := \sqrt{-g}\rho. \tag{61}$$

This is a measure of how much the distribution $\mu$ is spread in *space-time*, compared to the uniform distribution $\sqrt{-g}$. Using the formulas in App. D, we obtain for its second derivative along time-like geodesics the expression

$$\frac{d^2}{dt^2}\mathcal{S} = -\int_M \sqrt{-g}\,\rho \left[ R_{MN}\nabla_q^M \eta \nabla_q^N \eta + \nabla_M \nabla_N^q \eta \nabla^M \nabla_q^N \eta \right]. \tag{62}$$

The main difference compared to the Riemannian formula (43) is the appearance of the non-linear $q$-gradients. Equipped with formula (62) we can now characterize the Einstein equation as in the following

**Theorem 4.2.** *Let $(M, g)$ be a smooth space-time and fix $p \in (0, 1)$. The following statements are equivalent:*

1. *The Lorentzian metric $g$ on the space-time $M$ satisfies the equation of motion*

$$R_{MN} = \hat{T}_{MN}. \tag{63}$$

2. *i) For any probability distribution $\mu$ on $M$ evolving along a time-like geodesic (w.r.t. $\mathcal{C}_p$) in the space of probability distributions on $M$, with tangent vector $\dot{\mu} = -\nabla \cdot (\mu \nabla^q \eta)$, its Shannon entropy (61)(the relative entropy between $\mu$ and the volume form of $g$) satisfies*

$$\left. \frac{\mathrm{d}^2}{\mathrm{d}t^2} \right|_{t=0} \mathcal{S} \leqslant - \int_M \mu \, \hat{T}_{MN} \nabla_q^M \eta \nabla_q^N \eta. \tag{64}$$

*ii) In addition, the inequality (64) becomes saturated whenever $\mu$ is concentrated at a point, and for a suitably chosen $\eta$. Namely, for any point $x_0 \in M$ and any time-like tangent vector $\xi_0$ at $x_0$, there exists an $\eta$ such that $\nabla^q \eta|_{x_0} = \xi_0$ and such that (64) becomes an equality asymptotically for distributions $\mu$ very localized at $x_0$.*

*Proof.* The proof closely follows the one of the Riemannian theorem 4.1, and we highlight here only the differences. An important fact we used to prove both implications in the Riemannian case is that the quantity $\nabla_m \eta \nabla_n \eta \nabla^m \eta \nabla^n \eta$ appearing in the integrand of the second derivative of the entropy was manifestly non-negative. In the Lorentzian expression (62) this is replaced by $\nabla_M \nabla_N^q \eta \nabla^M \nabla_q^N \eta$, which is not immediately so. However, using Lemma D.1 this term is non-negative for $q < 1$, and so in particular for $p \in (0, 1)$. Moreover, a Lorentzian counterpart of (the Riemannian) Lemma B.1 holds (see [15, Lemma 3.2 (1)] or [14, Lemma 8.3]). We can thus follow the proof of the Riemannian theorem 4.1, *mutatis mutandis*. $\square$

## 5 Tsallis entropy and warped compactifications

We are now ready to describe one of our main results: the reformulation of the equations (11) for warped compactifications in terms of optimal transport. For this we will need the notion of *Tsallis entropy*, which as we review in section 5.1 is a natural generalization of the more usual Shannon one. In section 5.2 we show how to reformulate (11a) in terms of a relative entropy, while in section 5.3 we show how (11b) can be reformulated in terms of Tsallis entropy, along the lines of the previous section.[10]

### 5.1 Various definitions of entropy

We have already used the definition (42) of Shannon entropy $\mathcal{S}$ associated to a probability density $\rho$. This is famously related to the Gibbs entropy, to which it reduces when $\rho$ is defined on phase space. However, it is natural to wonder what properties single out $\mathcal{S}$ among the possible functionals of the form (21).

A set of such properties was provided by Khinchin [45] and Faddeev [46] for the case of probability distributions $p = (p_1, \ldots, p_n)$ on finite spaces of any cardinality $n$. Consider a function $\mathcal{S}(p) = \mathcal{S}(p_1, \ldots, p_n)$. If we demand that

---

[10]It would be interesting to try to reformulate in a similar fashion the equations of motion for all other fields as well; we will not attempt this here.

1. (Continuity) In the $n = 2$ case, $\mathcal{S}(p, 1-p)$ is continuous in $p \in [0, 1]$;

2. (Symmetry) $\mathcal{S}$ is a completely symmetric function of its entries (i.e. it remains the same if any two of the $p_i$ are exchanged);

3. (Additivity) $\mathcal{S}(tp_1, (1 - t)p_1, p_2, \ldots, p_n) = \mathcal{S}(p_1, \ldots, p_n) + p_1 \mathcal{S}(t, 1 - t)$, for any $t \in [0, 1]$;

then it can be shown that $\mathcal{S}$ is proportional to the discrete Shannon entropy

$$\mathcal{S} = -\sum_i p_i \log_2 p_i + \gamma. \tag{65}$$

The constant $\gamma$ can be fixed by also demanding a normalization, such as $\mathcal{S}(1/2, 1/2) = 1$.

The last property implies the more general

$$\mathcal{S}(p_1 q_{11}, \ldots, p_1 q_{1k_1}, p_2 q_{21}, \ldots, p_2 q_{2k_2}, \ldots, p_n q_{n1}, \ldots, p_n q_{nk_n})$$
$$= \mathcal{S}(p_1, \ldots, p_n) + \sum_i p_i \mathcal{S}(q_{i1}, \ldots, q_{ik_i}). \tag{66}$$

The particular case $q_{ij} = q_j$ yields the property that the entropy of a direct product of two probability distributions, which describes two independent events, is the sum of the entropy for the two events. This is the usual *extensivity* property. Symbolically we can write

$$\mathcal{S}(p \times q) = \mathcal{S}(p) + \mathcal{S}(q), \tag{67}$$

where we defined $p \times q = (p_1 q_1, \ldots, p_1 q_k, p_2 q_1, \ldots, p_2 q_k, \ldots, p_n q_1, \ldots, p_n q_k)$.

The idea of the proof that the axioms above lead to (65) is that $\mathcal{S}(p_1, \ldots, p_n)$ can be reduced to the $n = 2$ case using the Additivity axiom. (66) also gives $\mathcal{S}(r/s, 1 - r/s) = F(s) - r/s F(r) + (1 - r/s) F(s - r)$, where $F(n) := \mathcal{S}(1/n, \ldots, 1/n)$. Using (66) and the Continuity axiom one finds $F(nm) = F(n) + F(m)$ and $\lim_{n \to \infty}(F(n+1) - F(n)) = 0$. One can prove that this implies $F(n) \propto \log(n)$ [47, Lemma, Sec. 1]. Collecting all these observations one arrives at the Shannon entropy (65).

(67) is weaker than (66) and than the Additivity axiom; indeed there exist additional entropies that satisfy Continuity, Symmetry and (67), such as the *Rényi entropy* [47]

$$s_\alpha := \frac{1}{1 - \alpha} \log_2 \sum_i p_i^\alpha. \tag{68}$$

If one replaces the Additivity axiom with [48, 49]

3'. (Generalized Additivity) $\mathcal{S}(tp_1, (1 - t)p_1, p_2, \ldots, p_n) = \mathcal{S}(p_1, \ldots, p_n) + p_1^\alpha \mathcal{S}(t, 1 - t)$

then instead of the Shannon entropy one gets the *Tsallis entropy* [16]

$$\mathcal{S}_\alpha = \frac{1}{\alpha - 1}\left(1 - \sum_i p_i^\alpha\right). \tag{69}$$

The overall constant is chosen such that the limit $\alpha \to 1$ reduces to (65). Notice that

$$\mathcal{S}_\alpha = \frac{1}{\alpha - 1}\left(2^{(1-\alpha)s_\alpha} - 1\right). \tag{70}$$

(69) was originally introduced in the hope of describing distributions beyond the usual Boltzmann one, for examples with longer tails. It is extremized in the equiprobable case $p_i = 1/n$; this extremum is a maximum for $\alpha > 0$, a minimum for $\alpha < 0$. Notice however

that Generalized Additivity means that (66) also needs to be modified by $p_i \rightarrow p_i^\alpha$ in the second term in the right-hand side, and that in turn means that the extensivity property (67) is no longer satisfied: this is also evident from the relation (70) with the Rényi entropy, which is extensive. Rather we have

$$\mathcal{S}_\alpha(p \times q) = \mathcal{S}_\alpha(p) + \mathcal{S}_\alpha(q) + (1-\alpha)\mathcal{S}_\alpha(p)\mathcal{S}_\alpha(q). \tag{71}$$

A reformulation of these characterizations was suggested in [50]. To any $f$, a probability-preserving map between two sets with probability distributions $p$ and $q$, one associates a number $F(f)$ obeying three axioms called Functoriality, Linearity, Continuity. It can then be proven that $F(f) = \mathcal{S}(p) - \mathcal{S}(q)$, where $\mathcal{S}$ is again proportional to the Shannon entropy. Thus the function $F$ quantifies the loss of information associated to the map $f$. If Linearity is replaced by a different Homogeneity axiom, the Tsallis entropy (69) is recovered.

## 5.2 Warping equation and relative entropy

We begin our reformulation with the equation (11a) for the warping.

We can think of the warping as defining a measure on $(X, g_n)$, with density $e^f$ with respect to the distribution $\sqrt{g}dx^n$. We denote by $\mathfrak{f} := e^f \sqrt{g}\, dx^n$ the corresponding measure.

As in (42), we can define its relative entropy compared to $\sqrt{g}$ as[11]

$$\mathcal{S}_f := -\int_X \sqrt{g}e^f f + \gamma, \tag{72}$$

if $f$ is integrable on $(X, \mathfrak{f})$ (we used that $\rho = e^f$), and $+\infty$ otherwise. Now, assume that $\mathfrak{f}$ changes in time, with velocity $\dot{\mathfrak{f}}$ with compact support (or, fast decreasing at infinity). Then, applying equation (22) with $P(\rho) = -\rho = -e^f$ we get

$$\frac{d}{dt}\bigg|_{t=0} \mathcal{S}_f = -\int_X \sqrt{g}e^f \Delta\eta = -\int_X \sqrt{g}\eta\Delta(e^f). \tag{73}$$

Comparing with (18) we see that the right hand side is the scalar product

$$\frac{d}{dt}\bigg|_{t=0} \mathcal{S}_f = \langle \dot{\mathfrak{f}}, -\sqrt{g}\Delta(e^f)\rangle_W. \tag{74}$$

Comparing with (26) we have obtained

$$\nabla_W \mathcal{S}_f = -\sqrt{g}\Delta(e^f), \tag{75}$$

where $\nabla_W$ denotes the gradient in Wasserstein space $\mathcal{P}_2(X)$. With this relation, we can finally write the warping equation (11a) as

$$\nabla_W \mathcal{S}_f = -(D-2)\sqrt{g}e^f\left[\frac{1}{d}\hat{T}^{(d)} - \Lambda\right]. \tag{76}$$

To summarize, the warping equation (76) fixes the warping by constraining the gradients of its relative entropy compared to the Riemannian volume form.

---

[11]We could simply call $\mathcal{S}_f$ a Shannon entropy; however, beginning with section 3.3 we have seen that the natural measure in our context is the weighted $e^f\sqrt{g}$, not the usual Riemannian $\sqrt{g}$. For this reason we prefer calling $\mathcal{S}_f$ a relative entropy.

### 5.3 Internal Einstein equations as entropy concavity

We now turn to the internal equation (11b). We consider the Tsallis entropy (of $\mu = \rho \sqrt{g} e^f$ with respect to the reference $\sqrt{g} e^f$):

$$\mathcal{S}_\alpha = \frac{1}{\alpha - 1} \left( 1 - \int_{M_n} \sqrt{g} e^f \rho^\alpha \right). \tag{77}$$

Since we integrate only along $M_n$, this is measuring our ignorance about the internal position of a particle. If it is massless and moves geodesically, then its internal trajectory will follow an internal geodesic (App. C). We will show now that (8) is equivalent to an equation about the second time derivative of $\mathcal{S}_\alpha$ for a probability distribution of such particles.

**Theorem 5.1.** *Let $(X, g)$ be a smooth Riemannian manifold. The following statements are equivalent:*

1. *The Ricci tensor of $g$ satisfies the equation of motion (11b).*

2. *i) For any probability distribution $\mu$ on $X$ moving along a geodesic in the space $\mathcal{P}_2(X)$ of probability distributions, the Tsallis entropy (77) with $\alpha = \frac{N-1}{N}$ satisfies*

$$\frac{d^2}{dt^2}\bigg|_{t=0} \mathcal{S}_\alpha \leqslant - \int_X \sqrt{g} e^f \rho^{\frac{N-1}{N}} (\Lambda g_{mn} + \tilde{T}_{mn}) \nabla^m \eta \nabla^m \eta, \tag{78}$$

*where $\dot{\mu} = -\nabla \cdot (\mu \nabla \eta)$.*
*ii) In addition, the inequality in (78) becomes saturated if $\mu$ is concentrated at a point, and for a suitably chosen $\eta$. Namely, for any point $x_0 \in X$ and any tangent vector $\xi_0$ at $x_0$, there exists an $\eta$ such that $\nabla \eta|_{x_0} = \xi_0$ such that (78) becomes an equality asymptotically for distributions $\mu$ very localized at $x_0$.*
*More precisely, for every point $x_0 \in X$ there exists a function $\omega$ with $\lim_{\varepsilon \to 0} \omega(\varepsilon) = 0$ such that the following holds: for any tangent vector $\xi_0$ at $x_0$ of unit norm $\|\xi_0\| = 1$, there exists a smooth function $\eta$ with $\nabla \eta|_{x_0} = \xi_0$ such that*

$$\left| \frac{d^2}{dt^2}\bigg|_{t=0} \mathcal{S}_\alpha + \int_X \sqrt{g} e^f \rho^{\frac{N-1}{N}} (\Lambda g_{mn} + \tilde{T}_{mn}) \nabla^m \eta \nabla^m \eta \right| \leq \omega(\varepsilon),$$

*for every probability measure $\mu$ supported in $B_\varepsilon(x_0)$.*

*Proof.* $1 \Rightarrow 2$: For i), we take $F = \rho^{\frac{N-1}{N}}$ in (21):

$$\mathcal{F}[\mu] = \int_{M_n} \sqrt{g} e^f \rho^{\frac{N-1}{N}} = 1 + \frac{1}{N} \mathcal{S}_\alpha, \qquad \alpha = \frac{N-1}{N}. \tag{79}$$

From the definitions below (22) and (23), we see that $P = -F/N$, $P_2 = F/N^2$. We need (B.8), the weighted version of (23); replacing in it the weighted Bochner identity (A.9) we obtain

$$\frac{d^2}{dt^2}\bigg|_{t=0} \mathcal{S}_\alpha = - \int_{M_n} \sqrt{g} e^f \rho^{\frac{N-1}{N}} \left[ (R_{mn} - \nabla_m \nabla_n f) \nabla^m \eta \nabla^n \eta + \nabla_m \nabla_n \eta \nabla^m \nabla^n \eta - \frac{1}{N} (\nabla_f^2 \eta)^2 \right] \tag{80a}$$

$$= - \int_{M_n} \sqrt{g} e^f \rho^{\frac{N-1}{N}} \left[ (\text{Ric}_f^N)_{mn} \nabla^m \eta \nabla^n \eta + H_{mn}^0 H_0^{mn} - \frac{n-N}{nN} \left( \nabla^2 \eta + \frac{n}{N-n} \nabla f \cdot \nabla \eta \right)^2 \right]. \tag{80b}$$

$H_{mn}^0 := (\nabla_m\nabla_n - \frac{1}{n}g_{mn}\nabla^2)\eta$ is the traceless part of the Hessian of $\eta$. In the second step we have used the definition (10), and again the identity (40). Recalling that for us $N = 2 - D < 0$, the second and third terms in the parenthesis in (80b) are positive. Using (11b) we arrive at (78). For ii), we use Lemma B.1, which makes the second and third terms in (80b) vanish asymptotically in the limit where $p \sim \delta(x - x_0)$.

$2 \Rightarrow 1$: Suppose now we know (78). Using (80b), we obtain

$$\int \sqrt{g}\,e^f \rho^{\frac{N-1}{N}}[E_{mn}\nabla^m\eta\nabla^n\eta + X(\eta)] \geqslant 0\,,$$

where we wrote (11b) as $E_{mn} = 0$, and $X(\eta)$ represents the second and third terms in (80b). Now it follows that $E_{mn}$ is semi-positive definite everywhere. (If this were not the case, there would exist a $x_0$ and a $\hat{\xi}_0$ such that $E_{mn}\hat{\xi}_0^m\hat{\xi}_0^n < 0$. By Lemma B.1, there would now exist $\hat{\eta}$ such that $\nabla\hat{\eta} = \hat{\xi}$ and $X^2 = 0$; taking now $\rho \sim \delta(x - x_0)$ we arrive at a contradiction.)

Now, again taking the measure to be concentrated at one point $x_0$, we take $\eta$ as in ii). Since by hypothesis the inequality becomes saturated, we can write

$$[E_{mn}\nabla^m\eta\nabla^n\eta + X(\eta)](x_0) = 0\,.$$

We know that $E_{mn}$ is semi-positive definite, so all terms are $\geqslant 0$; it follows that they are all zero. In particular $E_{mn}(x_0)\xi_0^m\xi_0^n = 0$. Since $x_0$ and $\xi_0$ are arbitrary, $E_{mn} = 0$ everywhere. $\qquad\square$

# 6 Effective negative dimensions and KK bounds

As another application of negative effective dimensions to warped compactifications, we will now obtain new bounds on their spin-two KK masses. Recall [51, 52] that these are eigenvalues of the weighted (or Bakry–Émery) Laplacian

$$\Delta_f(\psi) := -\frac{1}{\sqrt{g}}e^{-f}\partial_m\left(\sqrt{g}\,g^{mn}e^f\partial_n\psi\right) = \Delta\psi - \nabla f \cdot \mathrm{d}\psi\,, \tag{81}$$

with $f = (D-2)A$. In [1, 2] optimal transport techniques were already used to find bounds on these eigenvalues, but using the $N = \infty$ effective dimension. A lower bound on $\mathrm{Ric}^{\infty,f}$ could be obtained, but in terms of $\sigma := \sup|\mathrm{d}A|$, which unfortunately can get quite large in some solutions. The advantage of considering negative effective dimensions is that the bound (14) is in terms of the cosmological constant $\Lambda$ alone, thus avoiding the dependence on $\sigma$.

## 6.1 D$p$-branes

The possibility to work with some non-smooth spaces is quite powerful for string theory, as several important compactifications have singularities in their low-energy description due to the back-reaction of extended objects. Recall for example that O-plane singularities (and/or quantum effects) are necessary in order to obtain dS compactifications [53, 54]. D-brane singularities also appear often in AdS vacua, where they are holographically dual to flavor symmetries. We review here the singularities associated to D- and M-branes.

In the supergravity approximation, D-branes play the role of localized sources for the gravitational and higher-form electromagnetic fields. The presence of such a localized object produces a singularity in the classical fields it sources; this is in complete analogy to black holes in pure general relativity or for electrons in classical electrodynamics. While on the one hand, such singularities are expected to be resolved in a full quantum theory, on the other hand they are a general feature of low-energy descriptions. It is thus useful to develop mathematical tools that allow to handle such non-smooth spaces.

In the setting of ten-dimensional supergravity theories, D-branes are identified with a ten-dimensional Lorentzian metric that, in Einstein frame, has the following asymptotics:

$$\mathrm{d}s_{10}^2 \sim H^{\frac{p-7}{8}}\left(\mathrm{d}x_{p+1}^2 + H(\mathrm{d}r^2 + r^2\mathrm{d}s_{\mathbb{S}^{8-p}}^2)\right), \qquad \text{for } r \to 0. \tag{82}$$

Here $p \in \{0, 1, \ldots, 7\}$, $r$ is a radial coordinate in the transverse directions to the singular object, $\mathrm{d}x_{p+1}^2$ denotes the $p+1$ dimensional Lorentzian metric (in case $p = 0$, simply $-\mathrm{d}x_1^2$) corresponding to the subspace appearing in the singular limit $r \to 0$, and $\mathrm{d}s_{\mathbb{S}^{8-p}}^2$ denotes the round metric on the unit $8-p$-dimensional sphere $\mathbb{S}^{8-p}$; the function $H$ is harmonic on the transverse space and introduces the singularity.

In order to preserve maximal symmetry in vacuum compactifications, a D$p$-brane has to be extended along all the $d$ vacuum directions; however, in addition, it can also be extended in some of the internal directions. Comparing (6) and (82), we obtain that the internal metric $\mathrm{d}s_n^2$ has the following asymptotics

$$\mathrm{d}s_n^2 \sim \mathrm{d}x_{p+1-d}^2 + H(\mathrm{d}r^2 + r^2\mathrm{d}s_{\mathbb{S}^{8-p}}^2), \qquad \text{for } r \to 0, \tag{83}$$

where $\mathrm{d}x_{p+1-d}^2$ denotes the flat metric of the $(p+1-d)$-dimensional Euclidean space. Again from (6), we also get that the weight function $f$ satisfies

$$\mathrm{e}^f = \mathrm{e}^{8A} \sim H^{\frac{p-7}{2}}, \qquad \text{for } r \to 0. \tag{84}$$

Near the singularity, the harmonic function has the following asymptotics:

$$H \sim \begin{cases} (r/r_0)^{p-7}, & 0 \le p < 7, \\ -\frac{2\pi}{g_s}\log(r/r_0), & p = 7, \end{cases} \qquad \text{for } r \to 0, \tag{85}$$

where $r_0^{7-p} = g_s(2\pi l_s)^{7-p}/((7-p)\mathrm{Vol}(\mathbb{S}^{8-p}))$ for $p < 7$; as usual $g_s$ is the string coupling (a value for $\mathrm{e}^\phi$ at a reference point, often infinity) and $l_s$ is the string length.

The next definition, where (for some values of $p$) we allow singularities that are *asymptotic* to D$p$-branes, is slightly more general than the one given in our previous work [1] where we considered *exact* D-brane singularities.

**Definition 6.1** (Asymptotically D-brane metric measure spaces)**.** We define an *asymptotically D-brane metric measure space* a smooth and compact Riemannian manifold $(X, g)$ that is glued (in a smooth way) to a finite number of ends where the metric $g$ is asymptotically of the form (83) in a neighborhood of the closed singular set $\{r = 0\}$, depending on value of $p$ in the following precise sense:

- Case $p = 0, 1, \ldots, 5$. In the end the metric can be written as

$$g = \mathrm{d}x_{p+1-d}^2 + \left(\frac{r_0}{r}\right)^{7-p}\left(\mathrm{d}r^2 + r^2\mathrm{d}s_{\mathbb{S}^{8-p}}^2\right) + \omega(\Theta, r), \tag{86}$$

  with $r_0^{7-p} = g_s(2\pi l_s)^{7-p}/((7-p)\mathrm{Vol}(\mathbb{S}^{8-p}))$ and $\omega$ is a quadratic form in $\mathrm{d}\Theta, \mathrm{d}r$ (and independent of the variable $x$), satisfying

$$\sup_{\Theta \in \mathbb{S}^{8-p}} \limsup_{r \to 0} r^{7-p}|\omega|(\Theta, r) = 0.$$

- Case $p = 6$. In a neighborhood $\{r < \epsilon\}$ of the closed singular set $\{r = 0\}$, the metric is of the form (86) with

$$\sup_{\Theta \in \mathbb{S}^2} \limsup_{r \to 0} \frac{|\omega|(\Theta, r)}{r} = 0.$$

- Case $p = 7$. In a neighborhood $\{r < \epsilon\}$ of the closed singular set $\{r = 0\}$, the metric is of the form

$$g = dx_{8-d}^2 - \frac{2\pi}{g_s}(\log(r/r_0) - \eta(x,r))\left(dr^2 + r^2 ds_{\mathbb{S}^1}^2\right),\tag{87}$$

  where $\eta$ is a non-negative real valued function.

- Case $p = 8$. In a neighborhood $\{|r| < \epsilon\}$ of the closed singular set $\{r = 0\}$, the metric is of the form

$$g = dx_{9-d}^2 + (1 - h_8|r|)dr^2,\tag{88}$$

  where $h_8 > 0$ is a positive constant, and the measure is given by

$$\mathfrak{m}\llcorner_{\{|r|<\epsilon\}} = \sqrt{1 - h_8|r|}\, \mathrm{dvol}_g\llcorner_{\{|r|<\epsilon\}},$$

  where $\mathrm{dvol}_g$ is the Riemannian volume measure associated to $g$.

In all the above cases, we endow $X$ with a weighted measure, and view it as a metric measure space $(X, d, \mathfrak{m})$ where:

- The distance $d$ between two points $p, q \in X$ is given by

$$d(p,q) := \inf_{\gamma \in \Gamma(p,q)} \int g\left(\gamma'(t), \gamma'(t)\right) dt,$$

  where $\Gamma(p, q)$ denotes the set of absolutely continuous curves joining $p$ to $q$.

- The measure $\mathfrak{m}$ is a weighted volume measure $\mathfrak{m} := e^f \, \mathrm{dvol}_g$, with the function $e^f$ smooth outside the tips of the ends and gives zero mass to the singular set.

We say that $(X, d, \mathfrak{m})$ is an *(exactly) D-brane metric measure space* if, for each end, the error $\omega$ (resp. $\eta$) vanishes on a neighbourhood $\{r < \epsilon\}$ of the singular set $\{r = 0\}$.

**Remark 6.2** (Other localized sources). Let us briefly comment on other localized sources. First of all, fundamental strings (F1) and NS five-branes (NS5), have exactly the same expansion as D1 and D5 branes, respectively; this is indeed a consequence of the invariance under type IIB S-duality (or more generally under the SL(2, $\mathbb{Z}$) symmetry) of the asymptotic 10-dimensional Einstein metric (82).

For M2 and M5 branes in M-theory, the internal metric has again the asymptotic form (83), now with $H \sim (r/r_0)^{q-8}$, $q = 2, 5$. Notice it enters in the first case of Def. 6.1; in particular, for both M2s and M5s, the singularity is at infinite distance.

## 6.2 Some basics on metric measure spaces

Motivated by the appearance of singularities as discussed in the section above, we enlarge the class of spaces under consideration. We thus leave the framework of smooth weighted Riemannian manifold and enter the more general setting of metric measure spaces. Let us start with some basics. (For a longer introduction to some of these ideas see also Sec. 2.3 in our earlier [1].)

In the sequel $(X, d)$ will be a complete and separable metric space.
By *geodesic* over $(X, d)$ we mean a constant speed (length minimizing) geodesic, i.e. a curve $\gamma : [0, 1] \to X$ such that

$$d(\gamma(s), \gamma(t)) = |t - s|d(\gamma(0), \gamma(1)), \qquad \forall s, t \in [0, 1].$$

The space of all geodesics over a space $X$ will be denoted by $\Gamma(X)$. The evaluation map $e_t : \Gamma(X) \to X$, $t \in [0, 1]$, is defined as $e_t(\gamma) := \gamma(t)$.

The space $\mathcal{P}(X, \mathsf{d})$ is the space of Borel probability measures over $X$. When the distance $\mathsf{d}$ is clear by the context, we will simply write $\mathcal{P}(X)$. The space $\mathcal{P}_2(X) \subset \mathcal{P}(X)$ is the subset of probability measures with finite second moment. We endow $\mathcal{P}_2(X)$ with the 2-Wasserstein distance $W_2$ defined as in (15).

A measure $\pi$ realising the infimum in (15) is called an *optimal coupling*. A measure $\nu \in \mathcal{P}(\Gamma(X))$ is called an *optimal dynamical plan* if the probability measure $(e_0, e_1)_\sharp(\nu)$ is an optimal coupling between its own marginals, and we denote by $\mathrm{OptGeo}(\mu_0, \mu_1)$ the set of all the optimal dynamical plans between $\mu_0$ and $\mu_1$.

A *metric measure space* is a triple $(X, \mathsf{d}, \mathfrak{m})$ where $(X, \mathsf{d})$ is a complete and separable metric space and $\mathfrak{m}$ is a non-negative Borel measure which is finite on balls, i.e. $\mathfrak{m}(B_r(x)) < +\infty$ for every $r > 0$ and $x \in X$, where $B_r(x) := \{y \in X : \mathsf{d}(x, y) < r\}$.

Denote by $\mathrm{Lip}(X)$ (resp. $\mathrm{Lip}_{bs}(X)$) the space of Lipschitz functions on $(X, \mathsf{d})$ (resp. with bounded support). For a function $f \in \mathrm{Lip}(X)$ the slope at a point $x \in X$ is defined as

$$|\nabla f|(x) := \limsup_{y \to x} \frac{|f(y) - f(x)|}{\mathsf{d}(y, x)}, \qquad \text{if } x \text{ is an accumulation point},$$

and $|\nabla f|(x) := 0$ if $x$ is isolated.

### 6.2.1 Cheeger energy, Laplacian and heat flow

Given a function $f \in L^2(X, \mathfrak{m})$, the *Cheeger energy* $\mathsf{Ch}(f)$ is defined by [55] (see also [56])

$$\mathsf{Ch}(f) := \inf \left\{ \liminf_{n \to \infty} \frac{1}{2} \int_X |\nabla f_n|^2 \mathrm{d}\mathfrak{m} : f_n \in \mathrm{Lip}_{bs}(X), f_n \to f \text{ in } L^2(X, \mathfrak{m}) \right\},$$

with finiteness domain given by the vector space

$$W^{1,2}(X, \mathsf{d}, \mathfrak{m}) := \{f \in L^2(X, \mathfrak{m}) : \mathsf{Ch}(f) < +\infty\}.$$

We endow $W^{1,2}(X, \mathsf{d}, \mathfrak{m})$ with the norm $\|f\|_{W^{1,2}}^2 := \|f\|_{L^2}^2 + 2\mathsf{Ch}(f)$. The Cheeger energy is a convex, 2-homogenous and lower semicontinuous functional on $L^2(X, \mathfrak{m})$.

For $f \in W^{1,2}(X, \mathsf{d}, \mathfrak{m})$, $\mathsf{Ch}(f)$ can be represented in terms of the *minimal relaxed gradient* $|Df|$ as

$$\mathsf{Ch}(f) := \frac{1}{2} \int_X |Df|^2 \, \mathrm{d}\mathfrak{m}. \tag{89}$$

The minimal relaxed gradient is a local object in the sense that $|Df| = |Dg|$ $\mathfrak{m}$-a.e. (namely, almost everywhere with respect to $\mathfrak{m}$) on the set $\{f - g = c\}$, $c \in \mathbb{R}$, for all $f, g \in W^{1,2}(X, \mathsf{d}, \mathfrak{m})$. For more details on the minimal relaxed gradient we refer to [56].

If $\varphi : J \subset \mathbb{R} \to \mathbb{R}$ is Lipschitz, and $J$ is an interval containing the image of $f$ (with $\varphi(0) = 0$ if $\mathfrak{m}(X) = \infty$), then

$$\varphi(f) \in D(\mathsf{Ch}), \quad \text{and} \quad |D\varphi(f)| \leq |\varphi'(f)||Df|, \qquad \mathfrak{m}\text{-a.e. } \forall \varphi \in \mathrm{Lip}(X), \tag{90}$$

where the inequality makes sense thanks to the locality of the minimal relaxed gradient.

Thanks to [56, Lemma 4.3], for any function $f \in L^2(X, \mathfrak{m})$ with $|Df| \in L^2(X, \mathfrak{m})$ it is possible to find a sequence $(f_n)$ of Lipschitz functions with $f_n \to f$ and $|\nabla f_n| \to |Df|$ strongly in $L^2(X, \mathfrak{m})$. By a standard cutoff argument, we can further assume that $(f_n) \subset \mathrm{Lip}_{bs}(X)$. In other words, the class $\mathrm{Lip}_{bs}(X)$ is *dense in energy* in $W^{1,2}(X, \mathsf{d}, \mathfrak{m})$.

For every $f \in L^2(X, \mathfrak{m})$ the *heat flow* of $f$ is the unique locally Lipschitz curve $t \mapsto H_t(f)$ from $(0, \infty)$ to $L^2(X, \mathfrak{m})$ such that

$$\begin{cases} \frac{\mathrm{d}}{\mathrm{d}t} H_t(f) \in -\partial^- \mathsf{Ch}(H_t(f)), & \text{for a.e. } t \in (0, \infty), \\ H_t(f) \to f, & \text{as } t \to 0^+. \end{cases}$$

Here $\partial^- \mathrm{Ch} \subset L^2(X, \mathfrak{m})$ is the subdifferential of the Cheeger energy, i.e. given $f \in L^2(X, \mathfrak{m})$ it holds

$$\ell \in \partial^- \mathrm{Ch}(f) \iff \int_X \ell(g - f) d\mathfrak{m} + \mathrm{Ch}(f) \leqslant \mathrm{Ch}(g), \quad \text{for all } g \in L^2(X, \mathfrak{m}).$$

For a function $f \in L^2(X, \mathfrak{m})$, we write $f \in D(\Delta)$ if $\partial^- \mathrm{Ch}(f) \neq \emptyset$; when $f \in D(\Delta)$ we denote by $\Delta f$ the element of minimal $L^2$-norm in $\partial^- \mathrm{Ch}(f)$ and we refer to it as the *Laplacian* of $f$.

## 6.3 Cheeger bounds in infinitesimally Hilbertian metric measure spaces

The goal of this section is to prove some bounds on the spectrum of the Laplacian in the high generality of infinitesimally Hilbertian metric measure spaces, framework which include several singularities appearing in gravity compactifications (e.g. D$p$-branes, $O$-planes, etc.).

### 6.3.1 Infinitesimally Hilbertian metric measure spaces

Notice that, in general, the Laplacian is 1-homogenous but may not be linear (for instance in $\mathbb{R}^n$ endowed with a non-euclidean norm, or more generally on a Finsler manifold). This is equivalent to say that the heat flow $H_t : L^2(X, \mathfrak{m}) \to L^2(X, \mathfrak{m})$ in general is 1-homogenous but may not be linear, or, still equivalently, that the Cheeger energy is 2-homogenous but may not be a quadratic form, or, still equivalently, that $W^{1,2}(X, d, \mathfrak{m})$ in general is a Banach space but may not be a Hilbert space.

When the latter of two options is satisfied, i.e. when we have a "Riemannian" behaviour as opposed to a "Finslerian" one, we say that the space is *infinitesimally Hilbertian* (see [32,33]). Below is the precise definition.

**Definition 6.3.** A metric measure space $(X, d, \mathfrak{m})$ is said to be *infinitesimally Hilbertian* if the Cheeger energy satisfies the parallelogram identity, i.e.

$$\mathrm{Ch}(f + g) + \mathrm{Ch}(f - g) = 2\mathrm{Ch}(f) + 2\mathrm{Ch}(g), \qquad \forall f, g \in W^{1,2}(X, d, \mathfrak{m}). \tag{91}$$

As mentioned above, if $(X, d, \mathfrak{m})$ is infinitesimally Hilbertian, then the heat flow and the Laplacian are linear, $W^{1,2}(X, d, \mathfrak{m})$ is a Hilbert space, and, using (90), the quadratic form

$$\mathcal{E}(f) := 2\mathrm{Ch}(f),$$

defines a Dirichlet form, i.e. a $L^2(X, \mathfrak{m})$-lower semicontinuous quadratic form that satisfies the Markov property $\mathcal{E}(\varphi(f)) \leqslant \mathcal{E}(f)$ for every 1-Lipschitz function $\varphi : \mathbb{R} \to \mathbb{R}$ with $\varphi(0) = 0$. By construction, $H_t$ and $-\Delta$ correspond respectively to the (sub)-Markov semigroup and the infinitesimally generator associated to the form (see for instance [57] as a general reference on Dirichlet forms). Moreover, the heat flow is a self-adjoint operator on $L^2(X, \mathfrak{m})$, as well as the Laplacian $\Delta$ which becomes a non-negative, densely defined, self-adjoint operator. If the measure of the space is finite (or, more generally, if $\mathfrak{m}(B_r(\bar{x})) \leq A \exp(B r^2)$ for some $A, B > 0$, $\bar{x} \in X$ and every $r > 0$) the semigroup $H_t$ is also mass preserving, i.e. for every $f \in L^1 \cap L^2(X, \mathfrak{m})$ it holds (see for instance [56, Th. 4.16, Th. 4.20]):

$$\int_X H_t f \, d\mathfrak{m} = \int_X f \, d\mathfrak{m}, \quad \text{for every } t \geq 0. \tag{92}$$

Another important property of this class of spaces is the density of $\mathrm{Lip}_{bs}(X)$ in $W^{1,2}(X, d, \mathfrak{m})$, that follows easily from (91) using the $L^2$-lower semicontinuity of the Cheeger energy and the already stated density in energy of the Lipschitz functions.

The next proposition will allow to include several interesting singularities (e.g. both D$p$-branes and $O$-planes) in the framework of infinitesimally Hilbertian spaces.

**Proposition 6.4.** *Let* $(X, \mathrm{d}, \mathfrak{m})$ *be a metric measure space. Assume that* $(X, \mathrm{d}, \mathfrak{m})$ *is a smooth weighted Riemannian manifold out of a closed singular set of measure zero:*

- *there exists a closed subset* $\Sigma \subset X$ *with* $\mathfrak{m}(\Sigma) = 0$,

- *there exists a smooth (open) weighted Riemannian manifold* $(M, g, \mathrm{e}^f \sqrt{g})$,

- *there exists a measure-preserving isometry* $\Phi : X \setminus \Sigma \to M$, *i.e.*

$$\mathrm{d}_g(\Phi(x), \Phi(y)) = \mathrm{d}(x, y), \quad \text{for all } x, y \in X \setminus \Sigma, \qquad \Phi_\sharp(\mathfrak{m} \llcorner_{X \setminus \Sigma}) = \mathrm{e}^f \sqrt{g} \mathrm{d} x^n,$$

*where* $\sqrt{g} \mathrm{d} x^n$ *denotes the Riemannian volume measure of* $(M, g)$.

*Then* $(X, \mathrm{d}, \mathfrak{m})$ *is infinitesimally Hilbertian.*

*Proof.* First of all, since the Riemannian scalar product $g$ satisfies the parallelogram rule, it holds that

$$
\begin{aligned}
&\int_M \sqrt{g} \mathrm{e}^f g(\nabla(u' + v'), \nabla(u' + v')) + \int_M \sqrt{g} \mathrm{e}^f g(\nabla(u' - v'), \nabla(u' - v')) \\
&= 2 \int_M \sqrt{g} \mathrm{e}^f g(\nabla u', \nabla u') + 2 \int_M \sqrt{g} \mathrm{e}^f g(\nabla v', \nabla v') \quad \forall u', v' \in W^{1,2}(M, g, \mathrm{e}^f \sqrt{g}),
\end{aligned}
\tag{93}
$$

where $\nabla u'$ denotes the weak gradient of a Sobolev function $u' \in W^{1,2}(M, g, \mathrm{e}^f \sqrt{g})$ in the classical distributional sense.

Let now $u, v \in W^{1,2}(X, \mathrm{d}, \mathfrak{m})$. We have to check the validity of the parallelogram identity (91). Using the representation formula (89) and the fact that $\mathfrak{m}(\Sigma) = 0$, this is equivalent to show that

$$\int_{X \setminus \Sigma} |D(u + v)|^2 \mathrm{d}\mathfrak{m} + \int_{X \setminus \Sigma} |D(u - v)|^2 \mathrm{d}\mathfrak{m} = 2 \int_{X \setminus \Sigma} |Du|^2 \mathrm{d}\mathfrak{m} + 2 \int_{X \setminus \Sigma} |Dv|^2 \mathrm{d}\mathfrak{m}. \tag{94}$$

Since by assumption $\Sigma$ is a closed set and $X \setminus \Sigma$ is isomorphic to the smooth weighted Riemannian manifold $(M, g, \mathrm{e}^f \sqrt{g})$, we have that the relaxed gradient of a Sobolev function restricted to $X \setminus \Sigma$ coincides with the modulus of the classical weak gradient (in Sobolev sense). Thus the validity of (94) follows from (93). $\square$

**Remark 6.5.** The framework encompassed by the assumptions of Proposition 6.4 is very general, and includes most (if not all) the singularities appearing in the low-energy description of string theory as localised sources: for instance singular metrics which are asymptotic to D-branes, M-branes and O-planes near the singular set fit into this setting, since the closure of the singular set has measure zero.

### 6.3.2 Spectrum of the Laplacian and Cheeger constants in infinitesimally Hilbertian spaces

In this section we assume $(X, \mathrm{d}, \mathfrak{m})$ to be infinitesimally Hilbertian. We have seen in the previous section that the Laplacian is a non-negative, densely defined, self-adjoint operator, and thus it enters in the classical framework for spectral theory.

The *regular values* of $\Delta$ are the values $\lambda \in \mathbb{C}$ such that $(\lambda \mathrm{Id} - \Delta)$ has a bounded inverse. Its *spectrum* $\sigma(\Delta)$ is the set of numbers $\lambda \in [0, \infty)$ that are not regular values. A non-zero function $f \in D(\Delta)$ is an *eigenfunction* of $\Delta$ of *eigenvalue* $\lambda$ if $\Delta f = \lambda f$. The set of all eigenvalues constitutes the *point spectrum* while the *discrete spectrum* $\sigma_d(\Delta)$ is the set of eigenvalues

which are isolated in the point spectrum and with finite dimensional eigenspace. Finally the *essential spectrum* is defined as $\sigma_{ess}(\Delta) := \sigma(\Delta) \setminus \sigma_d(\Delta)$.

Recall that the self-adjointness of the Laplacian implies that eigenfunctions relative to different eigenvalues are orthogonal. For spaces of finite measure, constant functions are eigenfunctions relative to $\lambda_0 = 0$, and thus any other eigenfunction has null mean value.

Given $f \in W^{1,2}(X, \mathrm{d}, \mathfrak{m})$, $f \not\equiv 0$, its Rayleigh quotient is defined as

$$\mathcal{R}(f) := \frac{2\mathsf{Ch}(f)}{\int_X |f|^2 \, \mathrm{d}\mathfrak{m}}. \tag{95}$$

Notice that for any eigenfunction $f_\lambda$ of eigenvalue $\lambda$, it holds $\lambda = \mathcal{R}(f_\lambda)$.

The infimum of the essential spectrum plays an important role in the sequel, since the set of eigenvalues below $\inf \sigma_{ess}(\Delta)$ is at most countable and, listing them in an increasing order $\lambda_0 < \lambda_1 \leqslant ... \leqslant \lambda_k \leqslant ...$, it holds

$$\lambda_k = \min_{V_{k+1}} \max_{f \in V_{k+1}, f \not\equiv 0} \mathcal{R}(f), \tag{96}$$

where $V_k$ denotes a $k$-dimensional subspace of $W^{1,2}(X, \mathrm{d}, \mathfrak{m})$.

The *perimeter* of a Borel subset $B \subset X$ with $\mathfrak{m}(B) < \infty$ is defined by

$$\mathrm{Per}(B) := \inf \left\{ \liminf_{n \to \infty} \int_X |\nabla f_n| \, \mathrm{d}\mathfrak{m} : f_n \in \mathsf{Lip}_{bs}(X), f_n \to \chi_B \text{ in } L^1(X, \mathfrak{m}) \right\}.$$

Using the notion of perimeter, one can define the *k-Cheeger constant* (or *k-way isoperimetric constant*) as

$$h_k(X) := \inf_{B_0, .., B_k} \max_{0 \leqslant i \leqslant k} \frac{\mathrm{Per}(B_i)}{\mathfrak{m}(B_i)}, \tag{97}$$

where the infimum runs over all collections of $k + 1$ disjoint Borel sets $B_i \subset X$ such that $0 < \mathfrak{m}(B_i) < \infty$. Notice that $h_k(X) \leqslant h_{k+1}(X)$ for every $k \in \mathbb{N}$ and, when $\mathfrak{m}(X) < \infty$, $h_0(X) = 0$.

We also recall the following characterization of $h_1(X)$, valid for spaces of finite measure and that easily follows from the definitions recalling that the perimeter of a set coincides with the perimeter of its complement:

$$h_1(X) = \inf \left\{ \frac{\mathrm{Per}(B)}{\mathfrak{m}(B)} : B \subset X \text{ Borel subset with } 0 < \mathfrak{m}(B) \leqslant \mathfrak{m}(X)/2 \right\}. \tag{98}$$

### 6.3.3 Generalization of Cheeger bounds

First of all, the celebrated Cheeger inequality [58] holds in the high generality of infinitesimal Hilbertian spaces. We recall the statement below. For the proof we refer to [35, App. A]; see also [59] where it is shown that the inequality is strict in a large class of singular spaces.

**Theorem 6.6.** *Let $(X, \mathrm{d}, \mathfrak{m})$ be an infinitesimally Hilbertian metric measure space. If $\mathfrak{m}(X) < \infty$ and $\lambda_1 < \inf \sigma_{ess}(\Delta)$ then*

$$\frac{h_1(X)^2}{4} \leq \lambda_1. \tag{99}$$

*If $\mathfrak{m}(X) = \infty$ and $\lambda_0 < \inf \sigma_{ess}(\Delta)$, (99) holds replacing $\lambda_1$ by $\lambda_0$ and $h_1(X)$ by $h_0(X)$.*

We will now extend some theorems proved in [1] (after [60, 61]) from the class of RCD spaces to the more general framework of infinitesimally Hilbertian metric measure spaces (i.e. without any curvature assumption), and during the proofs we will focus on the modifications needed to address this case. In particular, all the results in this section apply to a very large class of singular metrics including D-branes, M-branes, O-planes (see Remark 6.5).

**Theorem 6.7.** *Let* $(X, \mathrm{d}, \mathfrak{m})$ *be an infinitesimally Hilbertian metric measure space. Let* $k \in \mathbb{N}^+$ *and let us suppose that* $\lambda_k < \inf \sigma_{ess}(\Delta)$. *If* $\mathfrak{m}(X) < \infty$ *then*

$$h_1(X)^2 \lambda_k \leqslant 128 k^2 \lambda_1^2. \tag{100}$$

*If* $\mathfrak{m}(X) = \infty$ *then* (100) *holds replacing* $\lambda_1$ *with* $\lambda_0$ *and* $h_1(X)$ *with* $h_0(X)$.

*Proof.* We consider a non-null function $f \in \mathrm{Lip}_{bs}(X)$ and set

$$\phi(f) := \inf_{t \geqslant 0} \frac{\mathrm{Per}(\{x : f(x) > t\})}{\mathfrak{m}(\{x : f(x) > t\})}.$$

In [1] the following bound has been proved

$$\phi(f) \leqslant 8\sqrt{2} \frac{k}{\sqrt{\lambda_k}} \frac{\||\nabla f|\|_{L^2}^2}{\|f\|_{L^2}^2}, \quad \text{for all } k \in \mathbb{N}^+, \tag{101}$$

and we take it for granted since its proof requires only the variational characterization of the eigenvalues (96) and the co-area inequality for Lipschitz functions, results that hold on infinitesimally Hilbertian metric measure spaces without requiring any curvature bound.

First, suppose $\mathfrak{m}(X) = \infty$. Since the class $\mathrm{Lip}_{bs}(X)$ is dense in energy, we can find a sequence of functions $(f_n) \in \mathrm{Lip}_{bs}(X)$ such that $f_n \to f$ and $|\nabla f_n| \to |Df|$ in $L^2(X, \mathfrak{m})$, where $f$ is an eigenfunction of eigenvalue $\lambda_0$. The result thus follows by applying (101) to such a sequence $(f_n)$, using the trivial fact $h_0(X) \leqslant \phi(f_n)$ for any $n$, and passing to the limit.

If $\mathfrak{m}(X) < \infty$ we argue in a similar way just by applying (101) to two sequences of functions $f_n, h_n \in \mathrm{Lip}_{bs}(X)$ that converge in $L^2(X, \mathfrak{m})$ respectively to the positive and negative parts $f^+, f^-$ of an eigenfunction $f$, of eigenvalue $\lambda_1$ with $|\nabla f_n| \to |Df^+|$ and $|\nabla h_n| \to |Df^-|$ in $L^2(X, \mathfrak{m})$. The existence of such sequences is again a consequence of the density in energy of $\mathrm{Lip}_{bs}(X)$, noticing that $f^+, f^- \in W^{1,2}(X, \mathrm{d}, \mathfrak{m})$ by (90). Recalling the definition of $h_1(X)$ given in (97) and that $\lambda_1 = \mathcal{R}(f^+) = \mathcal{R}(f^-)$ (see [62]) the result follows. $\square$

**Theorem 6.8.** *Let* $(X, \mathrm{d}, \mathfrak{m})$ *be an infinitesimally Hilbertian metric measure space and assume that there exists constants* $A, B > 0$ *such that, for some (and thus for all)* $\bar{x} \in X$, *it holds*

$$\mathfrak{m}(B_r(\bar{x})) \leq A \exp(B r^2), \quad \text{for all } r > 0. \tag{102}$$

*Then, there exists an absolute constant* $C > 0$ *with the following property: for any* $k \in \mathbb{N}^+$ *such that* $\lambda_k < \sigma_{ess}(\Delta)$, *it holds*

$$h_k(X)^2 \leqslant C k^6 \lambda_k. \tag{103}$$

*Proof.* First of all, recall that under the assumption (102) the heat flow is mass preserving, i.e. (92) holds (see [56, Th.4.16, Th. 4.20]).

- Case $\mathfrak{m}(X) < \infty$: Since the heat flow $H_t : L^2(X, \mathfrak{m}) \to L^2(X, \mathfrak{m})$, $t \geqslant 0$, is mass preserving and satisfies comparison principles (i.e. for any $c \in \mathbb{R}$: $f \geqslant c$ $\mathfrak{m}$-a.e. implies $H_t f \geqslant c$ $\mathfrak{m}$-a.e., and $f \leqslant c$ $\mathfrak{m}$-a.e. implies $H_t f \leqslant c$ $\mathfrak{m}$-a.e., see [56, Th.4.16]), then $H_t$ is Markovian, i.e. if $L^2(X, \mathfrak{m}) \ni f \geqslant 0$ $\mathfrak{m}$-a.e. then $H_t(f) \geqslant 0$ $\mathfrak{m}$-a.e., and $H_t 1 = 1$ where 1 denotes the function equal to 1 $\mathfrak{m}$-a.e.

  We can thus appeal to a result of Miclo [61, page 325] (as we did in [1]) and infer that

$$\frac{\tilde{C}}{k^6} \Lambda_k \leqslant \lambda_k, \tag{104}$$

for an absolute constant $\tilde{C} > 0$, where

$$\Lambda_k := \min\left\{\max_j \lambda_0(B_j) : \{B_j\}_{j=0}^k \text{ are pairwise disjoint Borel sets, } \mathfrak{m}(B_j) > 0\right\},$$

and we are defining $\lambda_0(B)$ as

$$\lambda_0(B) := \inf\left\{\mathcal{R}(f) : f \in W^{1,2}(X, \mathsf{d}, \mathfrak{m}), f = 0 \text{ } \mathfrak{m}\text{-a.e. on } B^c, f \not\equiv 0\right\}. \tag{105}$$

Let $B \subset X$ be a Borel set with $\mathfrak{m}(B) \in (0, \mathfrak{m}(X))$. We now fix $\varepsilon > 0$ and let $f \in W^{1,2}(X, \mathsf{d}, \mathfrak{m})$, $f = 0$ $\mathfrak{m}$-a.e. on $B^c$, be such that

$$\varepsilon + \lambda_0(B) \geqslant \frac{\int_X |Df|^2 \, d\mathfrak{m}}{\int_X |f|^2 \, d\mathfrak{m}}.$$

We fix now a Borel representative of $f$ and we set $B_t := \{x \in X : |f^2(x)| > t\} \subset B$, $t > 0$. Reasoning as in the proof of [59, Th. 4.6] (using that $f^2$ is a BV function to which we can apply the co-area formula and noticing that all the arguments involved do not require $(X, \mathsf{d}, \mathfrak{m})$ to be an RCD space) we can conclude that

$$2[\varepsilon + \lambda_0(B)]^{1/2} \geqslant 2\left(\frac{\int_X |Df|^2 \, d\mathfrak{m}}{\int_X |f|^2 \, d\mathfrak{m}}\right)^{1/2} \geqslant \frac{\int_0^{\text{ess sup} f^2} \text{Per}(B_t) \, dt}{\int_0^{\text{ess sup} f^2} \mathfrak{m}(B_t) \, dt}. \tag{106}$$

Now let $\phi_2(f) := \inf\left\{\frac{\text{Per}(B_t)}{\mathfrak{m}(B_t)} : t \in (0, \text{ess sup} f^2)\right\}$ and notice that $\phi_2(f) \in [0, \infty)$. By definition of infimum we find a $\bar{t} \in (0, \text{ess sup} f^2)$ such that

$$\frac{\int_0^{\text{ess sup} f^2} \text{Per}(B_t) \, dt}{\int_0^{\text{ess sup} f^2} \mathfrak{m}(B_t) \, dt} \geqslant \phi_2(f) \geqslant \frac{\text{Per}(B_{\bar{t}})}{\mathfrak{m}(B_{\bar{t}})} - \varepsilon. \tag{107}$$

Putting (106) and (107) together, it follows that for any Borel set $B \subset X$, $\mathfrak{m}(B) \in (0, \mathfrak{m}(X))$, there exists a Borel set $B_{\bar{t}} \subset B$, $\mathfrak{m}(B_{\bar{t}}) > 0$, such that

$$\varepsilon + 2[\varepsilon + \lambda_0(B)]^{1/2} \geqslant \frac{\text{Per}(B_{\bar{t}})}{\mathfrak{m}(B_{\bar{t}})}. \tag{108}$$

Since $B \subset X$ and $\varepsilon > 0$ are arbitrary, by (108) we obtain $h_k^2(X) \leqslant 4\Lambda_k$ from the definition (97) of $h_k(X)$. Together with (104) this leads to the desired conclusion, where $C := 4/\tilde{C}$.

- Case $\mathfrak{m}(X) = \infty$: For a Borel set $E \subset X$ with finite (non-zero) measure, we introduce the notation $H_{t,E}$ for the heat semigroup restricted to $E$:

$$H_{t,E} : L^2(\mathfrak{m}_E) \to L^2(\mathfrak{m}_E), \qquad H_{t,E}(f) := \chi_E H_t(\chi_E f),$$

where $\mathfrak{m}_E := \mathfrak{m}(E)^{-1} \mathfrak{m} \llcorner E$ is the conditional expectation of $\mathfrak{m}$ with respect to $E$. Using the assumption (102) and arguing by approximation, one can show that $H_{t,E} : L^2(\mathfrak{m}_E) \to L^2(\mathfrak{m}_E)$ is sub-Markovian in the sense that $H_{t,E} 1 \leqslant 1$ for any $t \geqslant 0$ (i.e. one approximates the infinite measure $\mathfrak{m}$ by an increasing sequence of measures $\mathfrak{m}_k$ where one can apply comparison principle, and then pass to the limit; see for instance the proof of [56, Th. 4.20]). Moreover, $H_{t,E}$ is a continuous self-adjoint semigroup in $L^2(\mathfrak{m}_E)$ (see [61, page 325]).

We can then follow verbatim the proof of the corresponding case given in [1, Theorem 4.9], noticing that the variational characterization of the eigenvalues $\lambda_0, ..., \lambda_k$ as well as the density of $\text{Lip}_{bs}(X)$ in $W^{1,2}(X, \mathsf{d}, \mathfrak{m})$ still hold under the infinitesimally Hilbertianity assumption.

$\square$

## 6.4 Curvature-dimension conditions

We have seen in Sec. 2.2 that when the REC is satisfied (and in particular for compactifications of string/M-theory) the weighted Ricci curvature satisfies the simple bound (14), $\mathrm{Ric}_f^{(2-d)} \geqslant \Lambda$. In Sec. 3.4 we saw that the number $N = 2 - d$ could be interpreted as an effective dimension. We will now introduce a class of spaces called RCD($K, N$) (for *Riemannian Curvature-Dimension*) that reduces to (14) on smooth manifolds for $K = \Lambda$, $N = 2 - d$, but includes more general singular spaces. We will show in Corollary 6.15 that the class of allowed singularities includes those induced by D$p$-branes. In our previous paper [1, Sec. 3] we treated the case $N \in (1, +\infty]$ (paying the price of not explicitly controlling the lower Ricci bound $K \in \mathbb{R}$), the treatment for $N < 0$ presents both similarities and differences. The main advantage of considering negative $N$ is that it allows for a very neat control of the Ricci lower bound, since $K$ coincides with the cosmological constant $\Lambda$, when the REC is satisfied.

For $N \in (-\infty, 0)$, $\kappa \in \mathbb{R}$ and $\theta \geqslant 0$, denote

$$
\mathfrak{s}_\kappa(\theta) := \begin{cases} \frac{1}{\sqrt{\kappa}} \sin(\sqrt{\kappa}\theta), & \text{if } \kappa > 0, \\ \theta, & \text{if } \kappa = 0, \\ \frac{1}{\sqrt{-\kappa}} \sinh(\sqrt{-\kappa}\theta), & \text{if } \kappa < 0, \end{cases} \tag{109}
$$

For $t \in [0, 1]$, define the quantity

$$
\sigma_\kappa^{(t)}(\theta) := \frac{\mathfrak{s}_\kappa(t\theta)}{\mathfrak{s}_\kappa(\theta)}, \qquad \sigma_\kappa^{(t)}(0) := t, \tag{110}
$$

with $\theta > 0$ if $\kappa \leqslant 0$ and $\theta \in (0, \pi/\sqrt{\kappa})$ if $\kappa > 0$.

Finally, for $t > 0$, consider the functions

$$
\tau_{K,N}^{(t)}(\theta) := t^{1/N} \sigma_{K/(N-1)}^{(t)}(\theta)^{(N-1)/N}, \tag{111}
$$

with $\theta > 0$ if $K \geqslant 0$ and $\theta \in \left(0, \pi\sqrt{\frac{N-1}{K}}\right)$ if $K < 0$. For $t = 0$, set $\tau_{K,N}^{(0)}(\theta) := 0$. For $K < 0$, we also set $\sigma_{K,N}^{(t)}(\theta) := \infty$ and $\tau_{K,N}^{(t)}(\theta) := \infty$ if $\theta \geq \pi\sqrt{\frac{N-1}{K}}$.

An important role will be played by the Tsallis entropy (77), for the choice $\alpha = (N-1)/N$ and $N \in (-\infty, 0)$. Let us briefly recall it in the metric measure notation. Let $(X, \mathrm{d}, \mathfrak{m})$ be a metric measure space and fix $\alpha > 1$. Given $\mu \in \mathcal{P}(X)$, the $\alpha$-Tsallis entropy of $\mu$ with respect to $\mathfrak{m}$ is defined by

$$
\mathcal{S}_\alpha(\mu) := \begin{cases} \frac{1}{\alpha-1}\left(1 - \int_X \rho^\alpha \mathrm{d}\mathfrak{m}\right), & \text{if } \mu = \rho\mathfrak{m} \ll, \mathfrak{m} \text{ and } \rho^\alpha \in L^1(X, \mathfrak{m}), \\ +\infty, & \text{otherwise.} \end{cases} \tag{112}
$$

Recall that $\mu \ll \mathfrak{m}$ means that $\mu$ is absolutely continuous with respect to $\mathfrak{m}$: any Borel set $E$ that has $\mathfrak{m}(E) = 0$ also has $\mu(E) = 0$.

We are now ready to recall the definition of the *curvature-dimension condition* CD($K, N$), for $N < 0$, given in [5]. See also [63–65] for some recent developments in the theory of CD($K, N$) spaces, for negative $N$: stability properties, existence of optimal transport maps and local-to-global property (for the reduced condition).

**Definition 6.9.** A metric measure space $(X, \mathrm{d}, \mathfrak{m})$ satisfies the curvature-dimension condition CD($K, N$), $K \in \mathbb{R}$ and $N < 0$, if for any couple of absolutely continuous measures $\mu_0 = \rho_0\mathfrak{m}$, $\mu_1 = \rho_1\mathfrak{m} \in \mathcal{P}_2(X)$ there exists $\nu \in \mathrm{OptGeo}(\mu_0, \mu_1)$ such that for all $t \in [0, 1]$ and all $N' \in [N, 0)$, denoting by $\mu_t := (\mathrm{e}_t)_\sharp \nu$, it holds

$$
\mathcal{S}_{\frac{N'-1}{N'}}(\mu_t) \geq -N' + N' \int_{X \times X} \left[ \tau_{K,N'}^{(1-t)}(\mathrm{d}(x,y))\rho_0(x)^{-\frac{1}{N'}} + \tau_{K,N'}^{(t)}(\mathrm{d}(x,y))\rho_1(y)^{-\frac{1}{N'}} \right] \mathrm{d}\pi(x,y), \tag{113}
$$

for all $t \in (0,1)$, for some $W_2$-optimal coupling $\pi$ from $\mu_0$ to $\mu_1$.

We say that $(X, \mathsf{d}, \mathfrak{m})$ satisfies the $\mathsf{RCD}(K, N)$ condition if it is infinitesimally Hilbertian and satisfies the $\mathsf{CD}(K, N)$ condition.

**Remark 6.10.** • The definition given in [5] involves a slightly different expression of the entropy, namely $\int_X \rho^{\frac{N-1}{N}} \mathrm{d}\mathfrak{m}$ is considered instead of the Tsallis form $-N \left( 1 - \int_X \rho^{\frac{N-1}{N}} \mathrm{d}\mathfrak{m} \right)$. However the two definitions are equivalent.

• If $(X, \mathsf{d}, \mathfrak{m})$ is a smooth metric measure space, then it satisfies $\mathsf{CD}(K, N)$ in the sense of Def. 6.9 if and only if its $N$-Bakry–Émery-Ricci tensor is bounded below by $K$ (cf. (10)), as proved in [5, Th. 4.20].

In the sequel, we will consider metrics in polar coordinates with points denoted as $x = (\Theta, r) \in \mathbb{S}^n \times (0, \infty)$, while the origin will be denoted by $\mathsf{O} := \{r = 0\}$.

**Lemma 6.11.** *Let $(X, \mathsf{d})$ be a metric space associated to a smooth, compact manifold glued smoothly with an end where the Riemannian metric (not smooth at the origin $\mathsf{O} \in X$) is of the form*

$$g = \mathrm{d}r^2 + \ell(r)^2 \mathrm{d}s_{\mathbb{S}^n}^2 + \omega(\Theta, r),$$

*with*

$$\limsup_{r \to 0} \ell(r)/r < \frac{2}{\pi}, \qquad \sup_{\Theta \in \mathbb{S}^n} \limsup_{r \to 0} |\omega|(\Theta, r)/r^2 = 0. \tag{114}$$

*Then the origin $\mathsf{O}$ cannot be in the interior of any geodesic of $X$, i.e. if $\gamma : [0,1] \to X$ is a geodesic and $\mathsf{O} \in \gamma([0,1])$ then either $\mathsf{O} = \gamma_0$ or $\mathsf{O} = \gamma_1$.*

*Proof.* Assume by contradiction that there exists a geodesic $\gamma : [0,1] \to X$ such that $\gamma_t = \mathsf{O}$ for some $t \in (0,1)$. Up to restricting and reparametrizing $\gamma$, we can assume without loss of generality that

$$\gamma([0,1]) \subset B_\varepsilon(\mathsf{O}), \qquad r(\gamma_0) = r(\gamma_1) = \varepsilon, \tag{115}$$

where $\varepsilon > 0$ is a small parameter to be fixed later in the proof. Since $\gamma$ passes through the origin $\mathsf{O}$, by triangle inequality we have that

$$\mathrm{Length}(\gamma) \geq \mathsf{d}(\gamma_0, \mathsf{O}) + \mathsf{d}(\mathsf{O}, \gamma_1) = r(\gamma_0) + r(\gamma_1) + o(\varepsilon) = 2\varepsilon + o(\varepsilon). \tag{116}$$

Using (114) and assuming $\varepsilon > 0$ is small enough, we have that

$$\begin{aligned}
\mathsf{d}(\gamma_0, \gamma_1) &= \mathsf{d}\big((\Theta(\gamma_0), r(\gamma_0)), (\Theta(\gamma_1), r(\gamma_1))\big) \\
&\leq |r(\gamma_0) - r(\gamma_1)| + \pi \min\{\ell(r(\gamma_0)), \ell(r(\gamma_1))\} + o(\varepsilon) \\
&< 2\varepsilon + o(\varepsilon).
\end{aligned} \tag{117}$$

The combination of (116) and (117) contradicts the identity $\mathrm{Length}(\gamma) = \mathsf{d}(\gamma_0, \gamma_1)$ given by the assumption that $\gamma$ is a geodesic. $\qquad \square$

The next proposition is inspired by [66, Th. 4]. We first describe the physics interpretation. On a space of the form given in the proposition below, geodesics tend to be attracted by the origin and bend towards it. Indeed we will check later in this section that D-brane singularities, which have positive tension, are of this form. Two antipodal points can be connected by one of these bended geodesics rather than by one that goes through the origin. Heuristically, this suggests that a distribution of particles moving towards the origin will spread out before refocusing on the other side; moreover, a single particle belonging to it will hit the origin with probability zero.

**Proposition 6.12.** *Let $(X, \mathsf{d}, \mathfrak{m})$ be a metric measure space associated to a smooth, compact, weighted manifold glued smoothly with an end where the Riemannian metric g can be written in polar coordinates around the (possibly non-smooth) origin O, and assume the following:*

1. *The distance $\mathsf{d}$ is the length distance associated to the metric g on the end of the form*

$$g := \mathrm{d}r^2 + \ell(r)^2 \mathrm{d}s_{\mathbb{S}^n}^2, \quad with \quad \ell(r)^2 \leqslant r^2. \tag{118}$$

2. *On the end, the measure $\mathfrak{m}$ is absolutely continuous with respect to the standard product measure of $\mathbb{S}^n \times (0, a)$, $a > 0$, and if $O \in X$ it holds $\mathfrak{m}(\{O\}) = 0$.*

*Then for every optimal dynamical plan $\nu$ such that $(\mathrm{e}_i)_\sharp \nu \ll \mathfrak{m}$, $i = 0, 1$, we have $\nu(\Gamma_O) = 0$, where $\Gamma_O := \{\gamma \in \Gamma(X) : \gamma_t = O \text{ for some } t \in (0, 1)\}$.*

*Proof.* In the proof we will consider points on the end of the manifold, and accordingly we will use the polar coordinates to denote them.

First of all, notice that for every $x_0 = (\Theta_0, r_0)$, $x_1 = (\Theta_1, r_1)$ we have $\mathsf{d}(x_0, O) = r_0$ and, as a consequence of the fact that $\ell(r)^2 \leqslant r^2$, we infer that $\mathsf{d}(x_0, x_1) \leqslant \mathsf{d}_{C(\mathbb{S}^n)}(x_0, x_1)$, where

$$\mathsf{d}_{C(\mathbb{S}^n)}(x_0, x_1) := \sqrt{r_0^2 + r_1^2 - 2r_0 r_1 \cos(\mathsf{d}_{\mathbb{S}^n}(\Theta_0, \Theta_1))}$$

is the standard cone distance for $C(\mathbb{S}^n)$.

The result will be a consequence of the following:

**Claim** $(*)$: For every $r > 0$ there exists at most one $\Theta \in \mathbb{S}^n$ such that $\gamma_0 = (\Theta, r)$ is the starting point of some geodesic $\gamma \in \mathrm{supp}(\nu) \cap \Gamma_O$.

Once the claim is settled, the proposition can be proved by contradiction. Since the restriction of an optimal dynamical plan is still optimal (see for instance [19, Th. 7.30 (ii)]), we can assume that $\nu$ is concentrated on $\Gamma_O$. Using the fact that $(\mathrm{e}_i)_\sharp \nu \ll \mathfrak{m}$, $i = 0, 1$, and since $\mathfrak{m}$ gives zero mass to $O$ we can also assume that $\gamma_0 \neq O$ and $\gamma_1 \neq O$ for $\nu$-a.e. $\gamma$. Equivalently, that the set of geodesics not starting nor ending in $O$ is of measure zero with respect to $\nu$. The claim $(*)$ implies that the measure $(\mathrm{e}_0)_\sharp \nu$ is concentrated on a set of the form $C_h := \{(h(r), r) : r > 0\}$ for some function $h$, and thus $\mathfrak{m}(C_h) = 0$ which contradicts the fact that $(\mathrm{e}_0)_\sharp \nu \ll \mathfrak{m}$.

Thus, it remains to prove the claim $(*)$. We split its proof in three steps.

**Step 1**. We first show that if $\gamma : [0, 1] \to X$ is a non-constant geodesic with endpoints $\gamma_0 = (\Theta_0, r_0)$ and $\gamma_1 = (\Theta_1, r_1)$ such that $\gamma_t = O$ for some $t \in (0, 1)$, then $\Theta_0$ and $\Theta_1$ are antipodal as points in $\mathbb{S}^n$.

Indeed, by the fact that the curve $\gamma$ is a geodesic we obtain $r_0 = t\mathsf{d}(\gamma_0, \gamma_1)$ and $r_1 = (1 - t)\mathsf{d}(\gamma_0, \gamma_1)$ which implies $r_1 = \frac{1-t}{t} r_0$. In particular

$$\frac{r_0^2}{t^2} = \mathsf{d}^2(\gamma_0, \gamma_1) \leqslant \mathsf{d}_{C(\mathbb{S}^n)}^2(\gamma_0, \gamma_1) = r_0^2 + \frac{(1-t)^2}{t^2} r_0^2 - 2\frac{(1-t)}{t} r_0^2 \cos(\mathsf{d}_{\mathbb{S}^n}(\Theta_0, \Theta_1)), \tag{119}$$

from which $\cos(\mathsf{d}_{\mathbb{S}^n}(\Theta_0, \Theta_1)) \leqslant -1$, i.e. $\Theta_0$ and $\Theta_1$ are antipodal, as desired.

**Step 2**. We show that for every $t \in (0, 1)$ there exists at most one geodesic $\gamma \in \mathrm{supp}(\nu)$ with $\gamma_t = O$.

Let us consider two geodesics $\gamma, \gamma' \in \mathrm{supp}(\nu)$ that at time $t$ pass through $O$. We have $\gamma_0 = (\Theta_0, tr)$, $\gamma_1 = (\Theta_1, (1-t)r)$ for some $\Theta_0, \Theta_1 \in \mathbb{S}^n$ and similarly $\gamma_0' = (\Theta_0', tr')$, $\gamma_1' = (\Theta_1', (1-t)r')$, and we can assume that $\Theta_0, \Theta_1$ (resp. $\Theta_0', \Theta_1'$) are antipodal by what we have previously proved. By the cyclical monotonicity and the triangle inequality for $\mathsf{d}_{C(\mathbb{S}^n)}$, we know that

$$0 \leqslant d^2(\gamma_0, \gamma_1') + d^2(\gamma_0', \gamma_1) - d^2(\gamma_0, \gamma_1) - d^2(\gamma_0', \gamma_1')$$
$$\leqslant d_{C(\mathbb{S}^n)}^2(\gamma_0, \gamma_1') + d_{C(\mathbb{S}^n)}^2(\gamma_0', \gamma_1) - r^2 - r'^2$$
$$\leqslant [tr + (1-t)r']^2 + [tr' + (1-t)r]^2 - r^2 - r'^2 = -2t(1-t)(r-r')^2,$$

which implies $r = r'$. By taking advantage of this information, we can derive

$$0 \leqslant d^2(\gamma_0, \gamma_1') + d^2(\gamma_0', \gamma_1) - d^2(\gamma_0, \gamma_1) - d^2(\gamma_0', \gamma_1')$$
$$\leqslant d_{C(\mathbb{S}^n)}^2(\gamma_0, \gamma_1') + d_{C(\mathbb{S}^n)}^2(\gamma_0', \gamma_1) - 2r^2$$
$$= -2r^2 t(1-t)[2 + \cos(d_{\mathbb{S}^n}(\Theta_0, \Theta_1')) + \cos(d_{\mathbb{S}^n}(\Theta_0', \Theta_1))],$$

and in particular $\Theta_0$ and $\Theta_1'$ are antipodal and thus $\Theta_0 = \Theta_0'$ and $\Theta_1 = \Theta_1'$.

**Step 3**. Conclusion of the proof of the claim $(*)$.

Consider $\gamma, \gamma' \in \text{supp}(\nu) \cap \Gamma_O$ with $\gamma_0 = (\Theta_0, r)$, $\gamma_0' = (\Theta_0', r)$ for some $r > 0$. By assumption $\gamma$ and $\gamma'$ are geodesics passing through the origin, so that $\gamma_1 = (\Theta_1, r_1)$, $\gamma_1' = (\Theta_1', r_1')$ where $d_{\mathbb{S}^n}(\Theta_0, \Theta_1) = \pi$, $d_{\mathbb{S}^n}(\Theta_0', \Theta_1') = \pi$ for $r_1$, $r_1'$ positive real numbers. Again by the cyclical monotonicity and the properties of d we have

$$0 \leqslant d^2(\gamma_0, \gamma_1') + d^2(\gamma_0', \gamma_1) - d^2(\gamma_0, \gamma_1) - d^2(\gamma_0', \gamma_1')$$
$$\leqslant d_{C(\mathbb{S}^n)}^2(\gamma_0, \gamma_1') + d_{C(\mathbb{S}^n)}^2(\gamma_0', \gamma_1) - (r + r_1)^2 - (r + r_1')^2$$
$$= -2rr_1'[1 + \cos(d_{\mathbb{S}^n}(\Theta_0, \Theta_1'))] - 2rr_1[1 + \cos(d_{\mathbb{S}^n}(\Theta_0', \Theta_1))].$$

That forces $\Theta_0$ and $\Theta_1'$ to be antipodal and thus $\Theta_0 = \Theta_0'$. $\qquad\square$

**Remark 6.13.** From Step 1 in the proof, it follows that geodesics passing through the singular point O do not branch; i.e. if two geodesics coincide for a finite time, they coincide for ever. Moreover, since out of O the space is smooth, we conclude that if $(X, d)$ is as in the assumptions of Proposition 6.12, then $(X, d)$ is non-branching.

**Theorem 6.14.** *Let $(X, d, \mathfrak{m})$ be a metric measure space associated to a smooth, compact, weighted manifold glued smoothly with an end where the metric is non-smooth at a point $O \in X$. Let us suppose that $\text{Ric}_f^N \geqslant K$ on $X \setminus \{O\}$, where $f$ is the weight function, $K \in \mathbb{R}$ and $N < 0$. Let us also suppose that for every optimal dynamical plan $\nu$ such that $(e_i)_\sharp \nu \ll \mathfrak{m}$, $i = 1, 2$, we have $\nu(\Gamma_O) = 0$, where $\Gamma_O := \{\gamma \in \Gamma(X) : \gamma_t = O \text{ for some } t \in (0, 1)\}$. Then $(X, d, \mathfrak{m})$ is an $\text{RCD}(K, N)$ space.*

*Proof.* Let $N' \in [N, 0)$, $\mu_0 = \rho_0 \mathfrak{m}$, $\mu_1 = \rho_1 \mathfrak{m}$ be two measures in $\mathcal{P}_2(X)$, and $\nu \in \text{OptGeo}(\mu_0, \mu_1)$. Since $\nu(\Gamma_O) = 0$, following [5, Th. 4.10] for $\nu$-a.e. geodesic $\gamma$ emanating from $x$ with velocity $v = \nabla\varphi(x)$ and for every $t \in [0, 1]$ it holds

$$\mathbf{J}_t(x)^{\frac{1}{N}} \leqslant \tau_{K,N}^{(1-t)}(|v|) + \tau_{K,N}^{(t)}(|v|)\mathbf{J}_1(x)^{\frac{1}{N}}, \tag{120}$$

where $\mathbf{J}_t$ is the Jacobian of $\gamma(t) = \mathbf{T}_t(x) := \exp(tv)$ and $(\mu_t)_{t \in [0,1]}$ is the unique minimal geodesic $\mu_t = \rho_t \mathfrak{m} = (\mathbf{T}_t)_\sharp \mu_0$ connecting $\mu_0$ to $\mu_1$. Integrating (120) with respect to $\nu$ we obtain (113) with $N' = N$. Since $\text{Ric}_f^{N'} \geqslant \text{Ric}_f^N$ on $X \setminus \{O\}$, the conclusion follows. $\qquad\square$

In the following corollary we specify the previous results to the case of D$p$-branes.

**Corollary 6.15.** *Let $(X, d, \mathfrak{m})$ be an asymptotically D-brane metric measure space in the sense of Def. 6.1.*

- *Fix $K, N \in \mathbb{R}$. Assume that, on the smooth part of $X$, the $N$-Bakry–Émery Ricci tensor* (10) *is bounded below by $K$. Then $(X, \mathsf{d}, \mathfrak{m})$ is an $\mathrm{RCD}(K, N)$ space.*

- *In particular, if $(X, \mathsf{d}, \mathfrak{m})$ is an asymptotically D-brane metric measure space satisfying (on the smooth part) the Einstein equations* (5)*, and the Reduced Energy Condition* (13) *holds, then $(X, \mathsf{d}, \mathfrak{m})$ is an $\mathrm{RCD}(K, N)$ space for $K = \Lambda$ (the cosmological constant) and $N = 2-d$ (where $d$ is the dimension of the extended space-time).*

*If, more strongly, $(X, \mathsf{d}, \mathfrak{m})$ is an (exactly) D-brane metric measure space, then it also satisfies the $\mathrm{RCD}(K', N')$ condition for some $K' \in \mathbb{R}$ and $N' \in (1, \infty)$,*

*Proof.* We first observe that, since (by the very definition 6.1) the singular set of an asymptotically D-brane metric measure space $(X, \mathsf{d}, \mathfrak{m})$ is closed and has zero $\mathfrak{m}$-measure, then $(X, \mathsf{d}, \mathfrak{m})$ is infinitesimally Hilbertian thanks to Proposition 6.4. In order to get that $(X, \mathsf{d}, \mathfrak{m})$ is an $\mathrm{RCD}(K, N)$ space it is thus enough to prove it satisfies the $\mathrm{CD}(K, N)$ condition. We discuss it case by case below.

- **Case $0 \leqslant p \leqslant 5$.** The distance between $O$ and any other point is infinite. Thus, we do not include the point $O$ in $X$ so that the space $(X, \mathsf{d})$ is a complete metric space. The $\mathrm{CD}(K, N)$ condition then follows by the compatibility with the smooth setting, see Remark 6.10.

- **Case $p = 6$.** With the change of coordinates $\rho = 2\sqrt{r_0}\sqrt{r}$ the metric near $O$ takes the form $g = \mathrm{d}\rho^2 + \frac{\rho^2}{4}\mathrm{d}s_{\mathbb{S}^2}^2 + \tilde{\omega}(\Theta, \rho)$ with $\sup_\Theta \limsup_{\rho \to 0} \frac{|\tilde{\omega}|(\Theta, \rho)}{\rho^2} = 0$. The conclusion follows from Lemma 6.11 and Th. 6.14, together with the bound on $\mathrm{Ric}_f^N$ on the smooth part.

- **Case $p = 7$.** With the change of coordinates $\rho = \sqrt{\frac{2\pi}{g_s}} \int_0^r \sqrt{-\log(\frac{s}{r_0})}\,\mathrm{d}s$ the metric near $O$ takes the form $g = \mathrm{d}\rho^2 + \ell(\rho)^2 \mathrm{d}s_{\mathbb{S}^1}^2$ with $\ell(\rho)^2 < \rho^2$ for $\rho > 0$ small enough. The conclusion follows from Proposition 6.12 and Th. 6.14, together with the bound on $\mathrm{Ric}_f^N$ on the smooth part.

- **Case $p = 8$.** In the previous work [1] we have already proved that a $D8$-brane metric measure space is an $\mathrm{RCD}(0, N)$ space for some $N \in (1, \infty)$. The claim follows by the fact that $\mathrm{RCD}(0, N)$ for some $N \in (1, \infty)$ implies $\mathrm{RCD}(0, N')$ for all $N' \in (-\infty, 0)$.

The second part of the statement follows from the first one, once we recall that the REC (13) coupled with the Einstein equations (5) implies that $\mathrm{Ric}_f^{(2-d)} \geqslant \Lambda$ (see (14)). The final claim was proved in our previous work [1, Th 3.2] □

### 6.4.1 O-planes are not $\mathrm{CD}(K, N)$, even for negative $N$

The heuristic reason why O-planes are not CD is in a sense the opposite of that behind Prop. 6.12 (see the informal discussion above it, and Fig. 1). For O-planes, we will see now that in (118) we need to take $l(r) > r$. One can now check that geodesics tend to be repelled by the origin $O$, which is intuitively due to O-planes having negative tension. If we send a distribution of particles towards the origin with the aim of making it reform with the same density on the other side, it will actually tend to focus near $O$, (say at time $t = 1/2$) before spreading out. As a consequence, two antipodal points are only connected by the geodesic going through the origin, in contrast with the positive-tension case in Prop. 6.12. Since the reference measure (123) of small balls centred at the origin goes to zero as the radius of the ball goes to zero and the entropy is super-linear for $N < 0$, it will follow that the entropy tends

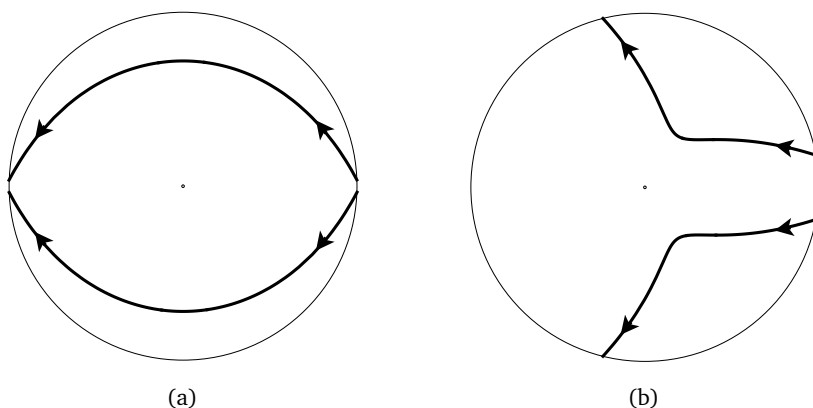

Figure 1: (a) Geodesics near a D$p$ singularity get attracted by it, and curve around it. (Here the $p = 6$ is depicted.) (b) An O$p$-plane has negative tension and repels geodesics, as one sees with (122).

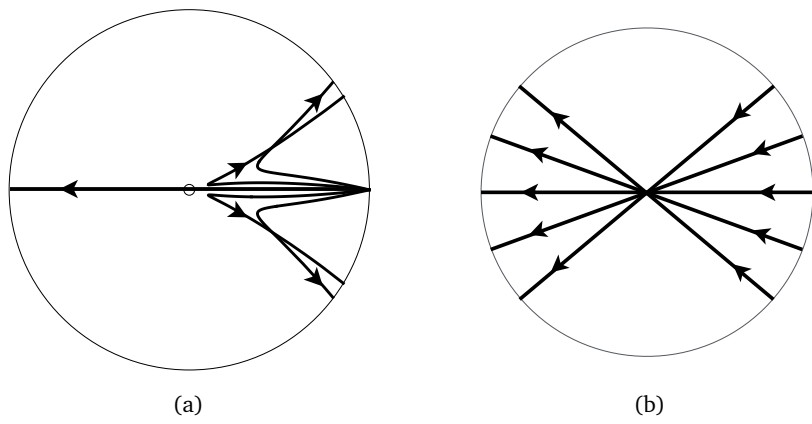

Figure 2: (a) Geodesics in an O$p$ geometry get deflected more as they get closer to the origin; the only way to get to the other side is to go straight through it. (b) As a consequence, the optimal plan that connects two antipodal distributions that are localized enough consists of straight geodesics going through the origin.

to negative infinity at the time $t = 1/2$ when the distribution of particles is concentrated near the origin. See Fig. 2 for a visualization of this behavior of geodesics in the O$p$-geometry.

Let us now turn such heuristics into a rigorous argument.

First of all observe that the CD$(K, N)$ condition (for some $K \in \mathbb{R}$ and $N \in (-\infty, 0)$, see Def. 6.9) on a metric measure space $(X, \mathrm{d}, \mathfrak{m})$ implies that there exists a constant $C = C(K, N) > 0$ with the following property: for any couple of absolutely continuous probability measures $\mu_0, \mu_1 \in \mathcal{P}_2(X)$ there exists a $W_2$-geodesic $(\mu_t)_{t \in [0,1]}$ such that

$$\frac{\mathcal{S}_{(N-1)/N}(\mu_{1/2})}{\max\left(\mathcal{S}_{(N-1)/N}(\mu_0), \mathcal{S}_{(N-1)/N}(\mu_1)\right)} \geq -C(K, N). \tag{121}$$

Recall that the (internal part of the) metric for an O-plane singularity is asymptotic to

$$\mathrm{d}s^2 = \mathrm{d}r^2 + \rho_0^2 r^{2/3} \mathrm{d}s^2_{\mathbb{S}^{8-p}}, \tag{122}$$

and the weighted measure is asymptotic to

$$d\mathfrak{m}(\Theta, r) = \rho_0 r^{1/3} dr \, d\Theta, \tag{123}$$

for some constant $\rho_0 > 0$ and some $p = 2, \ldots, 7$, where

$$\Theta = (\Theta_0, \Theta_1, \ldots, \Theta_{8-p}) \in \mathbb{S}^{8-p} \subset \mathbb{R}^{9-p},$$

denotes the unit vector in $\mathbb{R}^{9-p}$ parametrizing the unit sphere $\mathbb{S}^{8-p}$.

For $\varepsilon, \delta, r_0 > 0$ small, consider the probability measures

$$\mu_i^{\varepsilon,\delta,r_0} := \mathfrak{m}(A_i^{\varepsilon,\delta,r_0})^{-1} \mathfrak{m} \llcorner_{A_i^{\varepsilon,\delta,r_0}}, \quad i = 0, 1,$$

obtained by normalizing the restriction of $\mathfrak{m}$ to the sets

$$\begin{aligned}
A_0^{\varepsilon,\delta,r_0} &:= \{\Theta_0 \in [1-\delta, 1], \Theta_1, \ldots, \Theta_{8-p} \in [0, \delta], r \in [r_0 - \varepsilon, r_0], \}, \\
A_1^{\varepsilon,\delta,r_0} &:= \{\Theta_0 \in [-1, -1+\delta], \Theta_1, \ldots, \Theta_{8-p} \in [0, \delta], r \in [r_0 - \varepsilon, r_0]\}.
\end{aligned} \tag{124}$$

Notice that $A_0^{\varepsilon,\delta,r_0}$ and $A_1^{\varepsilon,\delta,r_0}$ are antipodal with respect to the singular origin

$$\{O\} := \{r = 0\}.$$

Moreover, since the power of $r$ in (122) is $2/3 < 2$ (where the exponent 2 would correspond to the cone metric), then, for $r > 0$ sufficiently small, the unique geodesic connecting the antipodal points $(\Theta, r)$ and $(-\Theta, r)$ passes through $O$.

It can be checked that, for $r_0 > 0$ and $\delta > 0$ small enough, the unique $W_2$-geodesic $(\mu_t^{\varepsilon,\delta,r_0})_{t \in [0,1]}$ from $\mu_0^{\varepsilon,\delta,r_0}$ to $\mu_1^{\varepsilon,\delta,r_0}$ passes through $O$ at time $t = 1/2$. More precisely, one can check that there exists $\rho = \rho(\varepsilon, \delta, r_0) > 0$ with

$$\lim_{(\delta,r_0) \to (0,0)} \lim_{\varepsilon \to 0} \frac{\rho(\varepsilon, \delta, r_0)}{r_0} = 0, \tag{125}$$

such that $\text{supp}(\mu_{1/2}^{\varepsilon,\delta,r_0}) \subset B_\rho(O)$. The expression of the measure (123) and (125) yield

$$\lim_{(\delta,r_0) \to (0,0)} \lim_{\varepsilon \to 0} \frac{\mathfrak{m}(B_\rho(O))}{\mathfrak{m}(A_0^{\varepsilon,\delta,r_0}) + \mathfrak{m}(A_1^{\varepsilon,\delta,r_0})} = 0.$$

Since the integrand of the entropy $\mathcal{S}_{(N-1)/N}$ is super-linear for $N < 0$, it follows that

$$\lim_{(\delta,r_0) \to (0,0)} \lim_{\varepsilon \to 0} \frac{\mathcal{S}_{(N-1)/N}(\mu_{1/2}^{\varepsilon,\delta,r_0})}{\max\left(\mathcal{S}_{(N-1)/N}(\mu_0^{\varepsilon,\delta,r_0}), \mathcal{S}_{(N-1)/N}(\mu_1^{\varepsilon,\delta,r_0})\right)} = -\infty. \tag{126}$$

Clearly, the last equation (126) is in contradiction with (121). We conclude that O-plane singularities (122) are not $\text{CD}(K, N)$ spaces for any value of $K \in \mathbb{R}$ and $N \in (-\infty, 0)$.

## 6.5 A lower bound in terms of the diameter

The statement below was proved in the Ph.D. thesis of E. Calderon [25, Th. 5.2.1] in the framework of smooth weighted Riemannian manifolds. By using 1-dimensional localization we can extend it to the non-smooth metric measure setting that includes D$p$-brane singularities. To this aim, we either assume that the metric measure space is asymptotically D-brane (see Def. 6.1), or we will have two RCD conditions: we assume the validity of both an RCD$(K, N)$

condition for some $N \in (-\infty, -1]$, $K < 0$, and an $\mathrm{RCD}(K', N')$ condition for some $K' \in \mathbb{R}$ and $N' \in (1, +\infty)$. This should be read as follows: while the former is giving the synthetic curvature-dimension condition we are actually interested in (so the bound we obtain will be in terms of $K$ and $N$), the latter should be read as a *qualitative regularity* assumption on the metric measure space to make the proof work (thus we do not want $K'$ and $N'$ to appear in the thesis).

**Theorem 6.16.** *Assume that $(X, \mathrm{d}, \mathfrak{m})$ is an $\mathrm{RCD}(K, N)$ space for some $N \in (-\infty, -1]$ and $K < 0$, whose diameter satisfies $0 < \mathrm{diam}(X) \leq \pi\sqrt{\frac{N-1}{K}}$. Assume moreover that*

- *either $(X, \mathrm{d}, \mathfrak{m})$ is an asymptotically D-brane metric measure space in the sense of Def. 6.1,*

- *or $(X, \mathrm{d}, \mathfrak{m})$ satisfies also an $\mathrm{RCD}(K', N')$ condition for some $K' \in \mathbb{R}$ and $N' \in (1, +\infty)$.*

*Then, the smallest eigenvalue $\lambda_1$ of the Laplacian satisfies*

$$\lambda_1 \geqslant \frac{\alpha(\mathrm{diam}(X)\sqrt{-K})}{\mathrm{diam}^2(X)}, \tag{127}$$

*where $\alpha(\mathrm{diam}(X)\sqrt{-K})$ is the minimum of*

$$\frac{\int_{-1/2}^{1/2} \mathrm{d}z\, \mathrm{e}^f (\partial_z \psi)^2}{\int_{-1/2}^{1/2} \mathrm{d}z\, \mathrm{e}^f \psi^2}, \qquad \mathrm{e}^f := \cos^{N-1}\left(\mathrm{diam}(X)\sqrt{\frac{K}{N-1}}z\right), \tag{128}$$

*among the functions $\psi$ which are smooth and have vanishing weighted average $\int_{-1/2}^{1/2} \mathrm{e}^f \psi$.*

*Proof.* The proof in the smooth weighted setting given in [25, Th. 5.2.1] can be summarised in two steps: first show that the desired bound can be reduced to a family of inequalities on weighted intervals (of topological dimension 1), second establish such a family of inequalities on weighted intervals.

The first step goes under the name of 1-dimensional localisation; in the setting of smooth weighted manifolds, such a dimensional reduction was obtained in [29]. The second step is the contribution of [25].

In [30], the 1-dimensional localisation was established in the framework of essentially non-branching $\mathrm{CD}(K', N')$ metric measure spaces for $N' \in (1, \infty)$, setting which includes $\mathrm{RCD}(K', N')$ spaces, for $N' \in (1, \infty)$, thanks to [67].

This gives the following. Given $u \in L^1(X, \mathfrak{m})$ with $\int_X u\, \mathrm{d}\mathfrak{m} = 0$, there exists a partition of $X$ as

$$X = \mathcal{N} \mathring{\cup} \mathcal{Z} \mathring{\cup} \left(\mathring{\bigcup}_{\alpha \in Q} X_\alpha\right), \tag{129}$$

where

- $\mathring{\cup}$ denotes a disjoint union,

- $\mathfrak{m}(\mathcal{N}) = 0$,

- for $\mathfrak{m}$-a.e. $z \in \mathcal{Z}$ it holds that $u(z) = 0$,

- $Q$ is a suitable set of indices and, for all $\alpha \in Q$, $X_\alpha$ is a geodesic in $X$.

Moreover, associated to the above partition of $X$, we have a disintegration of the measure $\mathfrak{m}$ as

$$\mathfrak{m} = \mathfrak{m}\llcorner_{\mathcal{Z}} + \int_Q \mathfrak{m}_\alpha\, \mathrm{d}q(\alpha), \tag{130}$$

where

- $\mathfrak{q}$ is a suitable measure on the set of indices $Q$,

- for $\mathfrak{q}$-a.e. $\alpha \in Q$, the measure $\mathfrak{m}_\alpha$ is concentrated on the geodesic $X_\alpha$ and the one-dimensional metric measure space $(X_\alpha, |\cdot|, \mathfrak{m}_\alpha)$ satisfies $\mathsf{CD}(K', N')$,

- for $\mathfrak{q}$-a.e. $\alpha \in Q$, it holds that $\int_{X_\alpha} u \, \mathrm{d}\mathfrak{m}_\alpha = 0$.

Using now that the ambient space satisfies also the $\mathsf{CD}(K, N)$ condition for $N < 0$ in the sense of Def. 6.9, one can follow verbatim the proof of [30, Th. 4.2] and infer that one-dimensional metric measure space $(X_\alpha, |\cdot|, \mathfrak{m}_\alpha)$ satisfies $\mathsf{CD}(K, N)$ as well, for $\mathfrak{q}$-a.e. $\alpha \in Q$.

Once these information are at disposal, the proof of the spectral bound under the additional assumption that $(X, \mathsf{d}, \mathfrak{m})$ is also an $\mathsf{RCD}(K', N')$ space for some $K' \in \mathbb{R}$, $N' \in (1, \infty)$ is obtained verbatim as in [25] (the modifications for the metric measure setting are now completely analogous to the proof of [68, Th. 4.4], setting $p = 2$).

Let us now briefly sketch the proof in the case when $(X, \mathsf{d}, \mathfrak{m})$ is an asymptotically $D$-brane metric measure space. By the proof of Corollary 6.15, the only case when some geodesic can pass through the singular set is for $p = 7$ or $p = 8$. In the latter case we already know that the singularity satisfies $\mathsf{RCD}(0, N')$ for some $N' \in (1, \infty)$ (at least locally in that end) and thus we can argue as above.

The only case remained to discuss is then for $p = 7$. From Remark 6.13 we get that the singular space corresponding to an asymptotic D7-brane is non-branching. From [69, Th. 3.3.5], we infer that the transport set (associated to the $L^1$-optimal transport problem used in the 1-d localization) is endowed by an equivalence relation whose equivalence classes are the geodesics $X_\alpha$, $\alpha \in Q$, mentioned above. Moreover, since it is easily seen that the cut locus has measure zero, we get that $\mathfrak{m}(X \setminus \bigcup_{\alpha \in Q} X_\alpha) = 0$. It follows that, up to a set of measure zero, all the transport set used in the 1-d localization is contained in the smooth part of the space. The result follows then by the smooth arguments in [25]. $\qquad\square$

**Remark 6.17.**  - Notice that $\alpha$ can also be interpreted as the lowest eigenvalue of the operator $-\mathrm{e}^{-f} \partial_{\check{z}}(\mathrm{e}^f \partial_{\check{z}}(\cdot))$ on the interval $[-1/2, 1/2]$, with Neumann boundary conditions.

- The assumptions of Th. 6.16 are natural thanks to Cor. 6.15: we proved that the validity of the Einstein equations and of the REC imply that an asymptotically D-brane metric measure space satisfies $\mathsf{RCD}(K, N)$ for $K = \Lambda$ (the cosmological constant) and $N = 2 - d < 0$ ($d$ is the dimension of the extended space-time); if, more strongly, $(X, \mathsf{d}, \mathfrak{m})$ is an exactly D-brane metric measure space then it satisfies also $\mathsf{RCD}(K', N')$ for some $K' \in \mathbb{R}$ and $N' \in (1, \infty)$.

- The bound is sharp, in the sense that there exist examples that saturate it. In particular it improves on a result [70, Th. 3], mentioned in [2, Th. 3], both because the bound does not contain $K$ (and hence the warping, in our physical application), and because it allows for singularities of $\mathsf{RCD}(N < -1, K)$ type, which as we saw in section 6.4 includes D-branes.

- The result in [25, Th. 5.2.1] actually also contains diameter bounds on other $N$ and $K$. In particular, for $N \in [-1, 0]$ and $K < 0$, the function cos is replaced by sin and the interval is shifted from $[-1/2, 1/2]$ to $[\varepsilon, 1 + \varepsilon]$. Also, for $N < 0$ and $K = \Lambda > 0$, the function cos is replaced by cosh; there is now no further limit on $K$. Unfortunately, $\Lambda > 0$ compactifications require the use of O-planes (or quantum corrections), so that part of the theorem is not of immediate application.

**Remark 6.18.** • As recalled above, for us $N = 2 - d$; when the REC is satisfied, the lower Ricci bound (14) holds, so we can choose $K = \Lambda$. Defining as usual the AdS radius $L_{\text{AdS}}$ via the identity $\Lambda = (1-d)/L_{\text{AdS}}^2$, the weight function becomes $e^f = \cos^{N-1}(\frac{\text{diam}}{L_{\text{AdS}}}z)$. Since from (81) the first eigenvalue corresponds to the mass of the first spin 2 Kaluza-Klein mode, in this situation we get the bound

$$\frac{m_1^2}{|\Lambda|} \geqslant \frac{\alpha(\text{diam}(X)/L_{\text{AdS}})}{d-1} \frac{L_{\text{AdS}}^2}{\text{diam}^2(X)}. \tag{131}$$

While we have not proven this rigorously, in $d = 4$ the minimum of (128) appears to be attained on $\psi = \sin(\pi z)$; the resulting $\alpha$ is a function of $\text{diam}(X)/L_{\text{AdS}} \in (0, \pi)$ such that

$$\lim_{\text{diam}(X)/L_{\text{AdS}} \to 0} \alpha = \pi^2, \qquad \lim_{\text{diam}(X)/L_{\text{AdS}} \to \pi} \alpha = 0, \tag{132}$$

monotone decreasing in between. In particular, the bound is most effective when $\text{diam}(X) \ll L_{\text{AdS}}$, and loses efficacy when $\text{diam}(X) \to \pi L_{\text{AdS}}$. A numerical analysis shows that a similar conclusion also holds in $d \neq 4$.

• When the REC is not satisfied, $K$ is not necessarily equal to $\Lambda$, but we can still consider the limit where the internal space is flat (or positively curved). Setting $K = -|\varepsilon| \to 0^-$, since in this limit $\alpha \to \pi^2$, we obtain

$$\frac{m_1^2}{|\Lambda|} \geqslant \frac{\pi^2}{d-1} \frac{L^2}{\text{diam}^2(X)}, \tag{133}$$

where $L^2 = (d-1)/|\Lambda|$ is the curvature radius of the vacuum (with either sign of the cosmological constant). (133) is now valid also for compactifications where the REC is violated, provided they are either smooth or at most with D/M-brane sources, and it proves that such vacua are automatically scale-separated when $\text{diam} \ll L$. We will construct such an example in Sec. 7.2.

# 7 Scale separation

## 7.1 Applications of KK bounds

Most of the theorems we have seen in Sec. 6 are lower bounds on the eigenvalues of the weighted Laplacian. These have a natural application to the issue of scale separation, namely the problem of finding solutions in which $m_{\text{KK}} \gg \sqrt{|\Lambda|}$.

In particular, (127) and the Remark 6.18 imply that a solution without O-planes with small diameter,

$$\text{diam} \ll L_{\text{AdS}}, \tag{134}$$

has scale separation at least in its spin-two tower. While this is intuitively expected, Theorem 6.16 establishes it rigorously. Recall that O-planes are not included because we have shown in section 6.4.1 that they do not satisfy the RCD condition, even for negative $N$. The theorem [70, Th. 3] quoted in [2, Th. 3] was not quite as conclusive, because one could have in principle weakened its conclusions by finding compactifications with large dilaton gradients. Moreover, as we mentioned, [70, Th. 3] only allowed smooth manifolds.

Most proposed solutions with scale separation do include O-planes. An exception is suggested in [37]. This construction originates in IIA, with a $T^2$-fibration over $T^4$ as internal space and intersecting O6-planes, which can be made approximately localized in the limit of small $\Lambda$.

However, the Romans mass vanishes (unlike the more famous [36]), so an uplift to M-theory is expected to exist, where the O6-planes become purely geometric features, locally described by the smooth Atiyah–Hitchin metric. (The uplift of an O6 intersection is not known, but one would expect it to be geometrized as well, at worst involving a mild singularity allowed in supergravity, such as that of an orbifold.) The sizes of the $T^2$, $T^4$ and M-theory circle can be made to scale with different powers as $\Lambda \to 0$; the diameter is expected to scale in this limit as the largest of these three, which can still be much smaller than $L_{\text{AdS}}$. Thus Theorem 6.16 implies that the M-theory version of this solution should have scale separation. It would thus be very interesting to test further the approximations made in finding it.[12]

We next consider a similar application for Theorem 6.6. This now implies that any solution with

$$\frac{1}{h_1} \ll L_{\text{AdS}} \tag{135}$$

is scale separated. Recall that the inequality (99) was also present in [1], but that here we stated it under the weaker assumption that the space is infinitesimally Hilbertian (Def. 6.3). This framework includes O-plane singularities (see Remark 6.5).[13]

While the Cheeger constant $h_1$ looks less familiar than the diameter, just like the diameter it is a non-local quantity that can be at least estimated if not precisely computed. Recall from (98) that we need to find the infimum of $\text{Per}(B)/\mathfrak{m}(B)$. In a solution with singularities, it is natural to first check what happens when $B$ surrounds one of them. In [1, (4.12)] we took $B$ to be a tubular neighborhood of radius $R$ around a D$p$ singularity, obtaining $\text{Per}(B)/\mathfrak{m}(B) \sim R^{(5-p)/2}$, which is arbitrarily small for $p < 5$ and large otherwise. A similar computation for an O$p$ ($p < 8$) singularity gives, in the same notation of [1],

$$\frac{\text{Per}(B)}{\mathfrak{m}(B)} = \frac{\int_{\partial B} \sqrt{\bar{g}_{\partial B}} \, e^{(D-2)A} \, \mathrm{d}^{n-1}x}{\int_B \sqrt{\bar{g}} \, e^{(D-2)A} \, \mathrm{d}^n x} = \frac{R^{8-p}\sqrt{H(R)}}{\int_{R_0}^R r^{8-p} H(r) \, \mathrm{d}r} \sim (R - R_0)^{-3/2}, \tag{136}$$

where $R_0 \sim l_s (g_s)^{1/(7-p)}$ is the radius below which the O$p$ metric becomes imaginary and loses meaning (see [71, Sec. 2] for a quick review of O$p$ solutions). We see $\text{Per}(B)/\mathfrak{m}(B)$ diverges when the neighborhood gets close to the O$p$ singularity, so this is not a good candidate to obtain the infimum that defines $h_1$. (The $p = 8$ case has to be treated separately, but the same conclusion holds.)

This logic can be applied to the famous proposal in [36]. The KK scale was already estimated there using an effective $d = 4$ theory as $m_{\text{KK}} \sim N^{-1/4} \ll 1/L_{\text{AdS}} \sim N^{3/4}$, where $N$ is the $F_4$ flux quantum. The geometry of the internal $M_6$ was given in [72, 73], again in an approximation where $\Lambda$ is small. The overall length scale of $M_6$ is $N^{1/4}$, which confirms the $d = 4$ estimate, but one might wonder if the backreaction of the O6s might affect this result significantly.

The result (136) indicates that this does not happen. Since taking $B$ near the O6s gives a large result, $h_1$ is more likely to be minimized by taking $B$ away from them. In such a region, the metric is approximately Calabi–Yau. For example, for a torus orbifold such as $T^6/\mathbb{Z}_2^3$ in [73, Sec. 6.2], [74], we can take $B = \{x^1 = 1/4\}$; the integrals of the $B_i$ functions is zero, and we obtain

$$h_1 \sim N^{-1/4}, \tag{137}$$

---

[12]Strictly speaking, the solution was shown in [37] to be supersymmetric only in the smeared limit, while for the localized version only the equations of motion were checked; while it does not seem very likely, in principle supersymmetry might be broken, and the vacuum might be unstable. We thank T. Van Riet and V. Van Hemelryck for discussions on this solution.

[13]This property might even extend to the corrected O-plane geometry in full string theory. It is actually generally expected that source singularities are smoothed out in quantum gravity, so this does not seem much of a stretch. A much bigger issue, which we do not tackle here, is that the spin-two operator itself might be affected significantly by string corrections near O-planes.

which by (99) gives $m_{\mathrm{KK}} > N^{-1/4}$, in line with the above-mentioned estimates. Of course this result is only relevant at the level of approximation considered in [72,73]; it is in principle still possible that the solution would somehow be destroyed in full string theory, were one able to perform such a computation.

## 7.2 Scale-separated solutions with Casimir energy

In this Section, we construct a new example of a scale-separated AdS solution with energy sources that violate the Reduced Energy Condition (Sec. 2.2).[14] When such sources are present in an AdS compactification ($\Lambda < 0$) equation (11a) for $\mathrm{Ric}_f^{(2-d)}$ reads schematically

$$\mathrm{Ric}_f^{(2-d)} = -|\Lambda| + \mathrm{REC} + \delta\mathrm{REC}, \tag{138}$$

where $\delta\mathrm{REC} < 0$ refers to terms that violate the REC. Compared to a situation where $\delta\mathrm{REC} = 0$, in which case $\mathrm{Ric}_f^{(2-d)}$ has at least a positive direction in all the known AdS examples, the negative term $\delta\mathrm{REC}$ can provide a mechanism to tune $\mathrm{Ric}_f^{(2-d)} = 0$ and thus decouple the internal curvature scale from the scale of the cosmological constant.[15] While it is not proved whether such a decoupling is necessary, for this reason it is believed that sources that violate the REC can be useful to achieve separation of scales, as already noticed in [2,77] (although they are not sufficient, e.g. [78, Sec. 2.2] [79, Sec. 7]).[16] In the DGKT example analyzed in Sec. 7.1 this is achieved through O6-planes, which violate the REC and stabilize a Ricci-flat internal space. In the following, we construct a new explicit example of an AdS scale-separated solution of the equations of motion with a Ricci flat internal space and with parametrically large ratio between the first Kaluza-Klein modes and $|\Lambda|$, by violating the REC through quantum effects.

We work in M-theory through its low energy description in terms of 11-dimensional supergravity, and we aim to construct a semi-classical solution of the equations of motion in which the quantum energy densities generated by the low-energy fields enter as a source in the Einstein equations (5) through the stress-energy tensor

$$T_{MN} = T_{MN}^{\mathrm{classical}} + \langle T_{MN}^{\mathrm{quantum}} \rangle. \tag{139}$$

The semi-classical approximation consists in choosing a geometry and topology for the space-time and computing the quantum effects with this assumption. Self-consistency requires then that the chosen space-time solves the equations of motion with the induced $\langle T_{MN}^{\mathrm{quantum}} \rangle$. This approach has been employed to construct various semi-classical gravity solutions such as compactifications of the Standard Model [81], traversable wormholes in four dimensions [82] and $\mathrm{dS}_4$ compactifications of M-theory [75].

We consider an $\mathrm{AdS}_4$ compactification on a 7-dimensional torus, so that the 11-dimensional space-time metric has the form

$$\mathrm{d}s_{11}^2 = R_4^2\,\mathrm{d}s_{\mathrm{AdS}_4}^2 + R_7^2\,\mathrm{d}s_{\mathrm{T}^7}^2, \tag{140}$$

where in this decomposition the metric on the $\mathrm{AdS}_4$ and $\mathrm{T}^7$ factors have unit radii.

---

[14]We thank Eva Silverstein and Gonzalo Torroba for discussions about this solution, its properties and related work [75].

[15]Another possible mechanism is to use codimension-2 sources, which have the appropriate scaling to cancel the internal curvature and achieve separation of scales [76].

[16]It is also known that sources that violate the REC are necessary in order to obtain de Sitter compactifications ($\Lambda > 0$) [53, 54, 80].

The zero point energy of fields in flat space and in curved backgrounds with different topologies can been computed explicitly in many cases (see e.g. [83] for a book-length review). The massless fields of eleven-dimensional supergravity are the metric $g_{11}$, the four-form $F_4$ and the gravitino $\psi$, and in order to generate a non-trivial zero point energy we break supersymmetry by imposing anti-periodic boundary conditions for fermions on the torus cycles. The contribution of massive states is exponentially suppressed and we need not consider them. Since we are considering an isotropic and homogenous internal torus, we can constrain the form of the induced effective action energy by requiring that: *i)* The energy density obtained from it is an eleven-dimensional energy density, depending only on the circle size $R_7$ and growing when it shrinks *ii)* It is homogenous and isotropic along the internal directions *iii)* The overall sign is due to bosons. These requirements are enough to impose that the leading term in the effective action has the form $S^{\text{eff}} = (2\pi)^{-8}|\rho_c| \int_{M_{11}} \sqrt{-g_{11}} R_7^{-11}$, giving the stress-energy tensor

$$\langle T_{\mu\nu}^{\text{Cas}} \rangle = (2\pi)^{-8}|\rho_c|\ell_{11}^9 R_7^{-11} g_{\mu\nu}, \qquad \langle T_{mn}^{\text{Cas}} \rangle = -\frac{4}{7}(2\pi)^{-8}|\rho_c|\ell_{11}^9 R_7^{-11} g_{mn}, \tag{141}$$

where $m, n$ are the internal torus directions, $\ell_{11}^9$ is the eleven-dimensional Planck length and $|\rho_c|$ is a positive order one numerical coefficient depending on the topology as well as on the number of degrees of freedom. This coefficient can be computed explicitly with a one-loop calculation of the Casimir energy on a torus (see e.g. [84, Sec. 3] for a computation in general higher-dimensional supergravity theories or [81, App. A]), but it is not important for our purposes since we will obtain parametric control. It is an easy check that (141) violates the REC (13):

$$T_{mn} - \frac{1}{4} g_{mn} T^{(4)} \Big|_{\text{Cas}} = -\frac{11}{7}(2\pi)^{-8}|\rho_c|\ell_{11}^9 R_7^{-11} g_{mn} < 0. \tag{142}$$

This property makes it promising for stabilizing a flat internal space, through the mechanism in (138). The Casimir energy (141) tends to make the torus expand, and we can stabilize its effect with an energy contribution of the opposite sign, such as a flux. In M-theory, we can consider a simple homogeneous configuration on AdS$_4$ for the four-form flux:

$$F_4 = f_4 \text{vol}_{\text{AdS}_4}, \tag{143}$$

with $f_4$ a real constant. Flux quantization of its magnetic counterpart $F_7 := \star F_4 = -f_4 \frac{R_7^7}{R_4^4} \text{vol}_{T^7}$ requires to relate $f_4$ to an integer $N_7$ as:

$$\frac{1}{(2\pi\ell_{11})^6} \int F_7 = N_7 \implies f_4^2 = \frac{N_7^2}{4\pi^2} \ell_{11}^{12} \frac{R_4^8}{R_7^{14}}. \tag{144}$$

Plugging these sources in the equations of motion (11) specialized to the ansatz (140), we get a solution if the radii are fixed as

$$R_4^2 = \ell_{11}^2 \left(\frac{N_7}{2\pi}\right)^{22/3} |\rho_c|^{-14/3} \frac{7^{14/3}}{2^{11} \times 3^{8/3}}, \qquad R_7^{11} = \ell_{11}^{11} \left(\frac{N_7}{2\pi}\right)^{22/3} |\rho_c|^{-11/3} \frac{7^{11/3}}{2^{11} \times 3^{11/3}}. \tag{145}$$

We want to assess the validity of this solution in the parametric regime $N_7 \gg 1$. We first notice from (145) that in this regime all the radii are large in Planck units ($\gg \ell_{11}$), and thus we have a competition of classical and quantum effect without needing subplanckian regimes. With this parametric control, other quantum and non-perturbative effects are suppressed.

In the large $N_7$ regime we then find a parametric separation between the AdS and internal scale as

$$\frac{R_7^2}{R_4^2} = \frac{2^9 \times 3^2}{7^4} |\rho_c|^4 \left(\frac{N_7}{2\pi}\right)^{-6}. \tag{146}$$

Since $R_7$ is proportional to the internal diameter, from (133) we get that this ratio directly controls the spectrum of the KK modes, achieving parametric separation of scales

$$\frac{m_{\text{KK}}^2}{|\Lambda|} \propto N_7^6 \gg 1, \quad \text{when } N_7 \gg 1. \tag{147}$$

For $N_7 \to \infty$, the KK modes go to zero very slowly relatively to the cosmological constant, as $m \sim |\Lambda|^\alpha$ with $\alpha = 1/11$. This result can be compared with the statement of the AdS Distance Conjecture of [85] that posits $\alpha$ to be an order one number (and $1/2$ for supersymmetric solutions). More recent studies [86, 87] that suggest $\alpha > 1/d$ (where $d = 4$ in the present case) are in tension with the solution (145). In particular this has implications for the Dark Dimension proposal of [88], which is based on the bound $\alpha > 1/d$.

The solution (145) has been computed by exploiting the fact that by tuning the flux integer $N_7 \gg 1$ the background is parametrically close to flat space. Moreover, being non-supersymmetric, it might be unstable for deformations of the torus or other effects. While as a function of the volume modulus this solution is at a minimum (obtained by balancing the negative contribution to the effective potential due to the Casimir with the positive potential from the flux), a more general perturbative analysis would require to compute the form of the Casimir stress energy tensor on the seven-dimensional torus away from the symmetric point. Being in AdS, a parametric analysis of this effect might not suffice, since tachyons of the scale of the cosmological constant could still be allowed if above the BF bound [89]. However, a naive probe computation suggests that it is unstable for nucleation of M2 bubbles in AdS$_4$. It would be interesting to assess in detail its stability with a more careful analysis, taking into account possible corrections of the M2 action due to the Casimir effect, and to understand the role of subleading effects.

# Acknowledgements

We thank A. Legramandi, A. Shahbazi-Moghaddam, E. Silverstein, G. Torroba, V. Van Hemelryck, T. Van Riet for discussions.

**Funding information** GBDL is supported in part by the Simons Foundation Origins of the Universe Initiative (modern inflationary cosmology collaboration) and by a Simons Investigator award. AM is supported by the ERC Starting Grant 802689 "CURVATURE". AT is supported in part by INFN and by MIUR-PRIN contract 2017CC72MK003.

# A  Raychaudhuri and Bochner

Here we will review some facts about the Raychaudhuri and Bochner identities, familiar in physics and geometry respectively, and derive their weighted counterparts.

We begin with the physics side. Consider a vector field $\xi^m$ to which geodesics are tangent. (We will use Latin indices, even though usually this logic is employed in Lorentzian signature.) The *expansion*

$$\theta := \nabla_m \xi^m \tag{A.1}$$

measures if nearby geodesics tend to attract or repel, as we will motivate shortly. We can compute its derivative along the geodesic:

$$\xi^m \nabla_m (\nabla_n \xi^n) = \xi^m \nabla_n \nabla_m \xi^n + \xi^m R^n{}_{pmn} \xi^p. \tag{A.2}$$

Using the geodesic equation $\xi^m \nabla_m \xi^n = 0$ we obtain the *Raychaudhuri equation*

$$-\nabla_\xi \theta = \nabla_n \xi_m \nabla^m \xi^n + R_{mn} \xi^m \xi^n \,. \tag{A.3}$$

We now motivate $\theta$'s name. When $\xi^m$ it is timelike, one usually normalizes it to $\xi^2 = -1$ so that the affine parameter is proper time $\tau$. Moreover, one often assumes that the field is *hypersurface orthogonal*: there exists a family of hypersurfaces to which $\xi$ is everywhere orthogonal, representing "space slices". By the Frobenius theorem and the normalization we chose, this implies $\mathrm{d}\xi = 0$. Consider now a family of geodesics depending on a parameter $s$. The vector field $d := \partial_s$ represents the "displacement" between two nearby geodesics; it commutes with $\xi = \partial_\tau$, since together they define a two-dimensional sheet. But then we know $\nabla_\xi d^n = \xi^m \nabla_m d^n = \nabla_m \xi^n d^m$. Since $\nabla_\xi$ represents evolution along geodesics, the matrix $\nabla_m \xi^n$ gives its action on the displacement $d$. But then its trace $\theta$ measures the overall attraction around a geodesic, as claimed.

Let us now instead assume $\xi$ to be a closed one-form $\xi$, $\mathrm{d}\xi = 0$, but not necessarily related to geodesics. Recall that the Laplace–de Rham operator acts on forms as $\Delta = \{\mathrm{d}, \mathrm{d}^\dagger\}$. In particular

$$-\Delta \xi_n = -(\mathrm{d}\mathrm{d}^\dagger \xi)_n = \nabla_n \nabla^m \xi_m = \nabla^m \nabla_n \xi_m - R^p{}_{mn}{}^m \xi_p = \nabla^m \nabla_m \xi_n - R_{np} \xi^p \,, \tag{A.4}$$

with the third step similar to (A.2). We now compute

$$\frac{1}{2} \nabla^2 \xi^2 = \nabla^m (\xi^n \nabla_m \xi_n) = \nabla^m \xi^n \nabla_m \xi_n + \xi^n \nabla^m \nabla_m \xi_n \,. \tag{A.5}$$

Using (A.4) we arrive at the *Bochner identity*:

$$\xi \cdot \Delta \xi + \frac{1}{2} \nabla^2 (\xi^2) = \nabla_m \xi_n \nabla^m \xi^n + R_{mn} \xi^m \xi^n \,. \tag{A.6}$$

Essentially the difference with Raychaudhuri is in how one handles the term $\xi^m \nabla_n \nabla_m \xi^n$, which is rewritten either using the geodesic equation or using closure of $\xi$. Other than this, the first term in (A.6) is $\xi^n \Delta \xi_n = -\xi^n \nabla_n \nabla_m \xi^m = -\nabla_\xi \theta$; recalling the normalization $\xi^2 = -1$, (A.6) reduces to the Raychaudhuri equation (A.3).

(A.6) has several interesting mathematical applications, such as to show that when the Ricci tensor is positive there cannot be any harmonic one-forms. When $\xi = \mathrm{d}\eta$, one can rewrite the first term as $\nabla \eta \cdot \nabla (\Delta \eta) = -\nabla \eta \cdot \nabla (\nabla^2 \eta)$, using that $\mathrm{d}$ and $\Delta$ commute.

We will now show how (A.6) is modified when the measure is of the form $\int e^f \sqrt{g}$. The adjoint of the exterior derivative is now

$$\mathrm{d}^\dagger_f := e^{-f} \mathrm{d}^\dagger e^f \,. \tag{A.7}$$

The associated Laplacian is $\Delta_f = \{\mathrm{d}, \mathrm{d}^\dagger_f\}$; on a function, this reproduces (31). With these definitions and using (A.4), we replace it by

$$\begin{aligned}
-\Delta_f \xi_n = -(\mathrm{d}\mathrm{d}^\dagger_f \xi)_n &= \nabla_n (e^{-f} \nabla^m (e^f \xi_m)) = \nabla_n (\nabla^m \xi_m + \nabla^m f \xi_m) \\
&= \nabla^m \nabla_m \xi_n - R_{np} \xi^p + \nabla_n \nabla^m f \xi_m + \nabla^m f \nabla_n \xi_m \\
&= e^{-f} \nabla^m (e^f \nabla_m \xi_n) - (R_{mn} - \nabla_m \nabla_n f) \xi^n \,.
\end{aligned} \tag{A.8}$$

We see the appearance here of $(\mathrm{Ric}^\infty_f)_{mn} = R_{mn} - \nabla_m \nabla_n f$, the $N \to \infty$ limit of (10). Using this and the definition (31) we arrive at the *weighted Bochner* identity:

$$\xi \cdot \Delta_f \xi + \frac{1}{2} \nabla^2_f (\xi^2) = \nabla_m \xi_n \nabla^m \xi^n + (R_{mn} - \nabla_m \nabla_n f) \xi^m \xi^n \,. \tag{A.9}$$

With this result in hand we can now also find the weighted Raychaudhuri equation. We can define the weighted expansion

$$\theta_f = \mathrm{e}^{-f} \nabla_m(\mathrm{e}^f \xi^m) = -\mathrm{d}_f^\dagger \xi. \tag{A.10}$$

Similar to the unweighted case, when $\xi^2 = -1$ we can simplify (A.9) to

$$-\nabla_\xi \theta_f = \nabla_m \xi_n \nabla^m \xi^n + (R_{mn} - \nabla_m \nabla_n f)\xi^m \xi^n. \tag{A.11}$$

# B  Flows of probability distributions

In this appendix we provide more details on the formal computations of the first and second derivatives of functionals along geodesics on $\mathcal{P}_2(X)$. These formal manipulations can be found in [19, Chap. 15] and are based on a formalism introduced by Otto [10, 39]. As in the main text, given a probability distribution $\mu$ on $X$, we can define its density $\rho$ with respect to the (possibly weighted) volume form as

$$\mu := \rho \sqrt{g} \mathrm{e}^f \mathrm{d}x^n, \tag{B.1}$$

where $\mathrm{e}^f$ is the weight function. We then represent a generic functional $\mathcal{F}$ as

$$\mathcal{F}[\mu] := \int_X \sqrt{g} \mathrm{e}^f F(\rho). \tag{B.2}$$

When $\mu$ is time-dependent, $\rho$ will depend on time too and we have for the derivative of $\mathcal{F}$ the expression

$$\frac{\mathrm{d}}{\mathrm{d}t}\mathcal{F}[\mu] = \int_X \sqrt{g} \mathrm{e}^f \dot{\rho} F'(\rho) = \int_X \dot{\mu} F'(\rho). \tag{B.3}$$

Using the continuity equation $\dot{\mu} := -\nabla \cdot (\mu \nabla \eta)$, (16), and integrating by parts we have

$$\frac{\mathrm{d}}{\mathrm{d}t}\mathcal{F}[\mu] = -\int_X \sqrt{g} \mathrm{e}^f \rho \nabla \eta \cdot \nabla(F'(\rho)) = -\int_X \sqrt{g} \mathrm{e}^f \nabla \eta \cdot \nabla(P(\rho)), \tag{B.4}$$

where we used $\nabla(P(\rho)) = \rho \nabla(F'(\rho))$ which directly follows from the definition $P(\rho) := \rho F'(\rho) - F(\rho)$. Integrating by parts one last time we finally get

$$\frac{\mathrm{d}}{\mathrm{d}t}\mathcal{F}[\mu] = \int_X \sqrt{g} \mathrm{e}^f P(\rho)\Delta_f(\eta), \tag{B.5}$$

where the warped Laplacian was defined in (31). Equation (22) in the main text is obtained from (B.5) for $f = 0$.

We can then compute an extra time-derivative of (B.5). This time we have two terms:

$$\frac{\mathrm{d}^2}{\mathrm{d}t^2}\mathcal{F}[\mu] = \int_X \sqrt{g} \mathrm{e}^f \dot{\rho} P'(\rho)\Delta_f(\eta) + \int_X \sqrt{g} \mathrm{e}^f P(\rho)\Delta_f(\dot{\eta}). \tag{B.6}$$

The first term on the right-hand side can be manipulated as

$$
\begin{aligned}
\int_X \dot{\mu} P'(\rho)\Delta_f(\eta) &= -\int_X \nabla\cdot(\mu\nabla\eta)P'(\rho)\Delta_f(\eta) \\
&= \int_X \mu\nabla\eta\cdot\nabla(P'(\rho))\Delta_f(\eta) + \int_X \mu P'(\rho)\nabla\eta\cdot\nabla(\Delta_f(\eta)) \\
&= \int_X \sqrt{g}e^f\nabla\eta\cdot\nabla P_2(\rho)\Delta_f(\eta) + \int_X \mu P'(\rho)\nabla\eta\cdot\nabla(\Delta_f(\eta)) \\
&= \int_X \sqrt{g}e^f P_2(\rho)(\Delta_f(\eta))^2 + \int_X \sqrt{g}e^f P(\rho)\nabla\eta\cdot\nabla(\Delta_f(\eta)),
\end{aligned}
\tag{B.7}
$$

where we used the relation $\rho\nabla(P'(\rho)) = \nabla P_2(\rho)$ that follows from the definition $P_2(\rho) := \rho P'(\rho) - P(\rho)$. For the second term in (B.6), we need to use the fact that the motion is along a geodesic in Wasserstein space, as opposed to a generic curve. In this case, $\eta$ satisfies the Hamilton–Jacobi equation (19). Plugging it in (B.6) and using (B.7) we finally arrive at

$$
\frac{d^2}{dt^2}\mathcal{F}[\mu] = -\int_X \sqrt{g}e^f P(\rho)\left[\Delta_f\left(\frac{|\nabla\eta|^2}{2}\right) - \nabla(\Delta_f(\eta))\cdot\nabla\eta\right] + \int_X \sqrt{g}e^f P_2(\rho)(\Delta_f(\eta))^2.
\tag{B.8}
$$

Equation (23) in the main text is obtained for $f = 0$, while in the proof of Th. 5.1 we used the expression with $f \neq 0$ being the warping.

Finally, for our proofs we also relied on the following lemma [19, p. 402], which is easy to show in normal coordinates.

**Lemma B.1.** *For any $x_0 \in X$, $\xi_0 \in T_{x_0}X$, and a function $f$, there exists another function $\eta$ such that*

$$
\left(\nabla_m\nabla_n\eta - \frac{1}{n}g_{mn}\nabla^2\eta\right)(x_0) = 0, \qquad (\nabla^2\eta + \nabla f\cdot\nabla\eta)(x_0) = 0,
\tag{B.9}
$$

*and*

$$
(\nabla\eta)(x_0) = \xi_0.
\tag{B.10}
$$

# C Geodesics

We will now show several facts about geodesics that we need in the main text.

## C.1 Space-time geodesics from Wasserstein geodesics

We used in the main text that geodesics in the Wasserstein space $\mathcal{P}_2(X)$ satisfy (19), where $\xi = \nabla\eta$ is the time-dependent velocity vector field appearing in the continuity equation (16)

$$
\dot{\mu} = -\nabla\cdot(\mu\xi).
\tag{C.1}
$$

Thus, the vector field $\xi(x, t)$ describes the motion of the bit of mass at $x$ at time $t$, and as a consequence of all the bits of mass composing $\mu$ moving along to $\xi$, $\mu$ changes in time as in (C.1). We will now show that when the motion is geodesic on $\mathcal{P}_2(X)$, then the trajectories of the individual bits of mass follow geodesics on $(X, g)$.

To see this, consider a bit of mass in $\mu$ that at $t = 0$ starts at $x = x_0$. It will then follow $\xi(x_0, 0)$ and after an amount of time $\Delta t$ it will end up in $x_1 = x_0 + \xi(x_0, 0)\Delta t + O(\Delta t^2)$. Once

there, it will follow $\xi(x_1, \Delta t)$ and so on. Thus, along the trajectory $x(t)$ its tangent vector will be

$$\zeta(t) := \xi(x(t), t). \tag{C.2}$$

For each $t$ this is a tangent vector in $T_{x(t)}(X)$. Lowering an index (i.e. considering the one-form $g(\zeta, \cdot)$), and taking a derivative along the curve we get

$$\begin{aligned}
\frac{\mathrm{d}}{\mathrm{d}t}\zeta_m &= \partial_t \xi_m + \dot{x}^n \partial_n \xi_m \\
&= \partial_t \xi_m + \xi^n \partial_n \xi_m,
\end{aligned} \tag{C.3}$$

where the right hand side is understood at $x = x(t)$, and thus we could substitute $\dot{x}^n$ with $\xi^n$.

Since $\eta = \nabla \xi$, we can take a covariant derivative of (19) to obtain

$$\partial_t \xi_m = -\xi^n \nabla_m \xi_n = -\xi^n \nabla_n \xi_m, \tag{C.4}$$

where we also used $\mathrm{d}\xi = 0$ to commute the indices. Plugging this into (C.3) we get

$$\begin{aligned}
\frac{\mathrm{d}}{\mathrm{d}t}\zeta_m &= \xi^n(\partial_n \xi_m - \nabla_n \xi_m) = \xi^n \Gamma^p_{nm}\xi_p \\
&= \frac{1}{2}\partial_m(g_{np})\xi^n \xi^p \\
&= \frac{1}{2}\partial_m(g_{np})\zeta^n \zeta^p.
\end{aligned} \tag{C.5}$$

This shows that when the probability distribution $\mu$ follows a geodesic on the probability space $\mathcal{P}_2(X)$, the individual particles follow geodesics on $(X, g)$. Tracing back the steps shows also the other implication.

A similar analysis can be performed for the Lorentzian case studied in Sec. 4.2. Specifically, a direct computation following the one given above shows that imposing the equations of motions (60) on a distribution of massive particles $\mu$ is equivalent to the requirement that each particle in the distribution follows a (time-like) geodesic. This formalism needs to be modified for massless particles since the $q$-gradient (59) would not be defined for light-like geodesics.

## C.2 Internal geodesics

In this section, we will show that massless geodesics on the $D$-dimensional warped product space-time (6), where $A$ is function depending only on the coordinate on $n$-dimensional Riemannian manifold $X_n$, can be projected to Riemannian geodesics for $X_n$, and viceversa.

This is a direct consequence of the well-known fact that only massless geodesics are mapped into geodesics upon a conformal transformation. [90, App. D], which we quickly review as follows. Take the geodesic equation on $\mathcal{M}_D$:

$$\frac{\mathrm{d}}{\mathrm{d}\sigma}(g_{MN}\partial_\sigma X^N) - \frac{1}{2}\partial_M(g_{QP})\partial_\sigma X^P \partial_\sigma X^Q = 0, \tag{C.6}$$

where $X^M$ are local coordinates on $\mathcal{M}_D$ and $\sigma$ is the coordinate along the geodesics. Defining $\bar{g} := \mathrm{e}^{-2A}g_D$, a new coordinate $\bar{\sigma} = \bar{\sigma}(\sigma)$ along geodesics with $\frac{\partial \bar{\sigma}}{\partial \sigma} = \mathrm{e}^{-2A}$, and recalling that $m^2 = -g_{QP}\partial_\sigma X^P \partial_\sigma X^Q$, we obtain

$$\frac{\mathrm{d}}{\mathrm{d}\bar{\sigma}}(\bar{g}_{MN}\partial_{\bar{\sigma}} X^N) - \frac{1}{2}\partial_M(\bar{g}_{QP})\partial_{\bar{\sigma}} X^P \partial_{\bar{\sigma}} X^Q + \frac{m^2}{2}\partial_M(\mathrm{e}^{2A}) = 0. \tag{C.7}$$

This shows that when the warp function $A$ is not constant only massless geodesics on $\mathcal{M}_D$ are geodesic on the unwarped product space-time. Since the latter is a simple product its geodesics then directly split into geodesics on the $d$ dimensional space-time and geodesics on $X_n$.

# D   Lorentzian transport

In this Appendix we briefly derive some formulas we need in Sec. 4.2 to prove the entropic reformulation of Einstein gravity in the Lorentzian case.

In particular, we want to compute derivatives of functionals on a space-time $M$ along time-like geodesics. Given a probability distribution $\mu$ on $M$, we write a generic functional as

$$\mathcal{F}[\mu] := \int_M \sqrt{-g} F(\rho), \qquad \text{with} \qquad \mu := \sqrt{-g} \rho\,.$$

Calling $\sigma$ the coordinate describing the geodesic evolution, we get

$$\frac{\mathrm{d}}{\mathrm{d}\sigma} \mathcal{F}[\mu] = \int_M \dot{\mu} F'(\rho) = \int_M \sqrt{-g} \nabla^q \eta \cdot \nabla P(\rho)$$
$$= \int_M \sqrt{-g} \Box_q(\eta) P(\rho)\,,$$

where we used that curves in probability space are parametrized in terms of vector fields on $M$ through the non-linear continuity equation in (60); we defined again $P(\rho) := \rho F'(\rho) - F(\rho)$. We also introduced the non-linear $q$-box operator

$$\Box_q(\eta) := -\nabla^M \nabla_M^q \eta\,,$$

as the natural second order operator associated to the $q$-gradient (59). We can now take another derivative of $\mathcal{F}$ and evaluate it at $\sigma = 0$ (which is, without loss of generality, a generic point along the geodesic). After a lengthy computation, in which we also make use of the geodesic equation in (60), we get

$$\frac{\mathrm{d}^2}{\mathrm{d}\sigma^2} \mathcal{F}[\mu] = \int_M \sqrt{-g} P(\rho) \left[ \frac{\mathrm{d}}{\mathrm{d}\sigma} \Box_q(\eta) + \nabla_M^q \nabla^M(\Box_q(\eta)) \right] + \int_M \sqrt{-g} P_2(\rho) (\Box_q(\eta))^2\,,$$

where we defined $P_2(\rho) := \rho P'(\rho) - P(\rho)$. Defining the linear operator

$$\mathcal{L}_{\eta,q}(\phi) := \frac{\mathrm{d}}{\mathrm{d}\sigma} (\Box_q(\eta + \sigma\phi)) \Big|_{\sigma=0}\,,$$

which along geodesics satisfies the relation

$$\frac{\mathrm{d}}{\mathrm{d}\sigma} (\Box_q(\eta)) \Big|_{\sigma=0} = \mathcal{L}_{\eta,q}(\dot{\eta}) = -\frac{1}{q} \mathcal{L}_{\eta,q}(|\nabla\eta|^q)\,,$$

we then get the expression

$$\frac{\mathrm{d}^2}{\mathrm{d}\sigma^2} \mathcal{F}[\mu] \Big|_{\sigma=0} = \int_M \sqrt{-g} P(\rho) \left[ -\frac{1}{q} \mathcal{L}_{\eta,q}(|\nabla\eta|^q) + \nabla_\nu^q \eta \nabla^\nu(\Box_q \eta) \right] + \int_M \sqrt{-g} (\Box_q \eta)^2 P_2(\rho)\,,$$
$$\tag{D.1}$$

where on the right hand side we have omitted the evaluation symbol.

Finally, we also need the $q$-analogue of the Bochner equation (24). An explicit computation (see for instance [15, App. A]) gives

$$-\frac{1}{q} \mathcal{L}_{\eta,q}(|\nabla\eta|^q) + \nabla_q \eta \nabla(\Box_q \eta) = (q-2)^2 |\nabla\eta|^{2(q-4)} (\nabla_P \nabla_M \eta \nabla^P \eta \nabla^M \eta)^2$$
$$- 2(q-2)|\nabla\eta|^{2(q-3)} \nabla^N \eta \nabla_M \nabla_N \eta \nabla_S \eta \nabla^M \nabla^S \eta$$
$$+ |\nabla\eta|^{2(q-2)} (R_{MN} \nabla^M \eta \nabla^N \eta + \nabla^M \nabla^N \eta \nabla_M \nabla_N \eta)\,. \tag{D.2}$$

We can now specialize our formulas to the Shannon entropy functional

$$\mathcal{S}[\mu] := -\int_M \sqrt{-g}\,\rho \ln \rho\,. \tag{D.3}$$

Combining (D.1) and (D.2) and collecting the $q$-gradients, we finally get the expression (62) for the second derivative of the Shannon entropy:

$$\frac{\mathrm{d}^2}{\mathrm{d}\sigma^2}\Big|_{\sigma=0}\mathcal{S} = -\int_M \sqrt{-g}\,\rho\left[R_{MN}\nabla_q^M\eta\nabla_q^N\eta + \nabla_M\nabla_N^q\eta\nabla^M\nabla_q^N\eta\right]. \tag{D.4}$$

For the proof of Th. 4.2 we also need the following

**Lemma D.1.** *On a D-dimensional Lorentzian space-time, for any function $\eta$ with time-like gradient and $q < 1$ we have the following inequality*

$$\nabla_M\nabla_N^q\eta\nabla^M\nabla_q^N\eta \geqslant 0\,, \tag{D.5}$$

*where the q-gradient $\nabla^q$ is defined as in (59).*

*Proof.* Given a time-like vector field $\xi$ consider the *q-Hamiltonian*

$$H_q(\xi) := -\frac{1}{q}(-g_{MN}\xi^M\xi^N)^{q/2}\,, \tag{D.6}$$

and define the quantity

$$H_{MN} := \frac{\partial^2 H_q(\xi)}{\partial \xi^M \partial \xi^N} = (2-q)\xi_M\xi_N|\xi|^{q-4} + |\xi|^{q-2}g_{MN}\,, \tag{D.7}$$

where $|\xi| := (-g_{MN}\xi^M\xi^N)^{\frac{1}{2}}$. Now, defining the matrix $B_M{}^P := H_{MN}\nabla^M\xi^P$, we have

$$B_{MP} = -(q-2)\xi_M|\xi|^{q-4}\xi^S\nabla_S\xi_P + |\xi|^{q-2}\nabla_M\xi_P = -\nabla_M\nabla_N^q\eta\,, \tag{D.8}$$

so that

$$B_{MP}B^{PM} = \nabla_M\nabla_N^q\eta\nabla^M\nabla_q^N\eta\,, \tag{D.9}$$

where as usual $\xi_M = \nabla_M\eta$. To prove the lemma we then need to show that $B_{MP}B^{PM} \geqslant 0$. To do that, let us consider a vielbein $e^A$ adapted to the time-like vector field $\xi$, that is $e^0 = \frac{\xi}{|\xi|}$ and $e^a$, $a = 1,\ldots,D-1$, being $D-1$ normalized space-like vectors orthogonal to $e^0$ and to each other. In this basis, the matrix (D.7) has the non-vanishing components (in flat indices):

$$H_{00} = |\xi|^{q-2}(1-q)\,, \qquad H_{ab} = |\xi|^{q-2}\delta_{ab}\,. \tag{D.10}$$

Then, defining $b_{MN} := \nabla_M\xi_N$, which is symmetric as a consequence of $\xi_M := \nabla_M\eta$, we have

$$B_{AB}B^{AB} = |\xi|^{2(q-2)}\left((1-q)^2 b_{00}^2 + 2(1-q)b_{0i}b^{0i} + b_{ik}b_{ki}\right). \tag{D.11}$$

This shows that $\nabla_M\nabla_N^q\eta\nabla^M\nabla_q^N\eta \geqslant 0$ if $q < 1$. $\qquad\square$

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
