# Peer review of "Gravity from thermodynamics: optimal transport and negative effective dimensions"

_SciPost Physics, doi:SciPost Phys. 15, 039 (2023)_

## Round 1 · Referee Report · Anonymous · 2023-2-9

Strengths
1. Timely topic
2. Out of the box ideas (the part on entropy and gravity)
3. Introduces math to the physics community that is highly needed.
4. Presents a very simple and new AdS vacuum with remarkable properties
Weaknesses
1. Certain sections are not easy to digest for physicists as they are definition- and notation-heavy.
2. Would be nice to make more clear how the optimal transport issues relate exactly to bounds on Laplacians for spin two modes.
3.
Report
I support this paper for publication. Below I explain why.
The paper is quite original and out of the box. The idea of getting Einsteins equations from links with entropy considerations in the theory of optimal transport is intriguing. It is then also very nice that the authors have attempted to be sufficiently comprehensive and recall some basic facts from the theory of optimal transport. Although I must admit that some sections are very heavy notation wise, especially the beginning of section 6. But I believe the authors have understood that and for that reason the extended introduction, where the main results are highlighted is very necessary.
I want to further make clear that as a referee I was not able to find the time to digest everything in all detail. Especially the discussions in section 6.2 and section 6.3 are beyond my understanding. I leave it to the editor to potentially include a more mathematical-oriented referee who can properly judge the mathematical results of this paper.
The way I regard this paper is an important piece of work that introduces mathematical tools for the physics community which recently got more interested in questions regarding Kaluza-Klein mass scales (the issue of scale separation).
The paper considers two main directions: the link between Einstein equations and optimal transport and the theory behind bounds on eigenvalues of generalised Laplace operators. The two topics are related of course and this is why the authors probably decided to write one long paper. Yet, it means this paper is not an easy read. Reading the paper I did wonder whether it would have been more natural to split the paper into two parts since the connection between Einsteins gravity and optimal transport can be of interest to readers who care less about recent problems in flux compactifications or issues regarding the singularities of physical sources (branes/planes) in string theory.
Requested changes
1. I would appreciate if the authors could be slightly more clear to what extend they derive new general results and to what extend they summarize existing results of the literature. I would not only help the general readers, but also the referees.
2. Is there a relation between the reduced energy condition and the claim by Douglas and Kallosh in
https://arxiv.org/abs/1001.4008 regarding the matter needed to sustain negatively curved internal dimensions? If this is not interesting, the authors do not need to make changes in the paper to accommodate this question.
3. Maybe this question takes one too much off topic, but I cannot resist asking whether the authors have contemplated a connection between getting Einsteins equation from optimal transport and Verlinde's ideas on gravity as an entropic force? Also here, if this is not interesting, the authors do not need to make changes in the paper to accommodate this question.
3. Since the Swampland conjectures involving scale separation do not mention stability, could the authors perhaps comment on whether there example in section 7 clearly demonstrates that the conjectures, as currently written down, are false?
4. Are the authors certain that the example of section 7.2 is under full control? Is there any insight on perturbative stability of this AdS vacuum? Because if this solution is truly perturbatively stable and scale separated, it is simply remarkable, yet it only takes two pages in this large paper.
5. Finally a small, irrelevant detail: on page 9 there is a mall typo below eq 2.9: it should say:" ....for a collection OF fields..."
Author: Giuseppe Bruno De Luca on 2023-04-21 [id 3607]
(in reply to Report 1 on 2023-02-09)
We thank the referee for the detailed comments and suggestions. Below we address them individually, stressing how we plan to update the manuscript to address them.
Weaknesses:
1) We acknowledge that some parts of the paper (in particular Section 6) are a bit more in a mathematical rather physical style of writing; this is why the introduction is a bit longer than usual, so to explain the ideas in a plain and accessible way to a non-expert reader. Moreover, also the more abstract parts of the paper contain comments aimed to guide the reader, who can follow the stream of the paper by reading the plain text and the statements (and skipping the proofs if under time constraints).
2) We will add a discussion at page 6 of the paper in order to comment the role of optimal transport in proving one of the main results on spectral bounds of the paper (Th. 6.16). Summarizing it here, Optimal transport plays a key role in the proof of Th. 6.16, based on the so-called “L1 -localization method”: the basic idea is that using L1 -optimal transport (i.e. optimal transport with cost function given by the distance function), it is possible to partition the (possibly singular) space X (up to a set of measure zero) into geodesics, each being endowed with a Borel non-negative measure. This reduces the proof of the desired inequality (1.3) in X, to proving a family of corresponding inequalities on 1-dimensional weighted spaces. Such a powerful dimension reduction argument has older roots and it was developed via optimal transport tools for smooth Riemannian manifolds by Klartag and for (possibly non-smooth) metric measure spaces satisfying synthetic Ricci lower bounds and dimensional upper bounds by Cavalletti and Mondino.
Requested changes:
1) We thank the referee for pointing out possible ambiguities in the presentation. The results for the purely Riemannian case (Thm. 4.1) and for the pure Lorentzian case (Thm. 4.2) are known in the mathematical literature, as we mention at the beginning of Secc. 4.1 and 4.2. The purpose of these sections is to re-obtain these mathematical results in a unified way for the smooth case using methods more familiar to the physics community and with the formalism we introduce in Sec. 2. The results for warped compactifications (Sec. 5, culminating in Thm. 5.1) are instead entirely new and we prove them for the smooth case using the same methods. We will add to the introduction of the manuscript a more detailed description of earlier work, remarking the difference between "rigorous" and "formal" results.
2) The energy condition by Douglas and Kallosh, eq. (2.32) in 1001.4008, looks similar to a trace of the REC (with d -> d -2). However, the REC is a stronger condition in that it controls all the internal directions, not just the trace. The need to constrain all the directions simultaneously, and not just the trace, stems from the fact that knowing only the Ricci scalar is too weak a constraint on the geometry for the purposes of obtaining rigorous bounds on the KK spectrum. The same is true for the warped case. We added a footnote about this above (2.9).
On the other hand, in (2.34) Douglas and Kallosh also consider a stronger condition. However, as they notice, their condition would imply a positive Ricci tensor, which is in general not true for warped compactifications even without O-planes or quantum effects; the REC indeed allows for negative directions.
3) We thank the referee for this suggestion. We are also intrigued by the possibility that a connection of this sort might exist; in the introduction (pp. 4–5) we had a general discussion in this vein, citing there paper [21] and [23]. The paper by Verlinde is similarly relevant, and we have added it to this list. A comment we made there is that our connection is purely classical, while this literature is more concerned with black hole entropy, an inherently quantum phenomenon. Nevertheless one can imagine merging the two approaches somehow; this would be very exciting.
3') We thank the referee for this important question. In the example in Sec. 7, the masses of the KK modes behave as m ~ |Λ|^α for α = 1/11 . The original AdS Distance conjecture of 1906.05225 claims α to be an order 1 number and it is compatible with this non-supersymmetric solution. However, more recent proposals that suggest that α > 1/d seem to be in tension with it. It would be interesting to come back to this question with a more detailed analysis of subleading quantum effects. We will expand the discussion at the end of Sec. 7 to mention these developments and comparisons with the literature.
4) We thank the referee for the question on this important point. A full perturbative stability analysis would require to consider arbitrary perturbations of the internal torus geometry and to recompute the Casimir energy for the perturbed configuration. This is a well-defined but technically challenging computation, as the form of the Casimir stress energy tensor away from the symmetric point can be highly non-trivial. We have added a sentence about this at the end of section 7. (The radii of the external and internal spaces are both large when $N \gg 1$, as we see from (7.12); so the solution is under perturbative control.)
5) We thank the referee for pointing this out, we will fix it in the revised manuscript.
Author: Giuseppe Bruno De Luca on 2023-05-12 [id 3668]
(in reply to Report 2 on 2023-04-24)We thank the referee for their kind words, and for their detailed comments and questions. Below we articulate how we intend to change the manuscript to address them.
1) In this work we considered general orientifold singularities, without a sufficient number of branes on top as to cancel the total tension. These have more severe curvature singularities that cannot be resolved by going to the covering space. In general, if a space is RCD its Z2 quotient is also RCD [ Galaz-Garcia,Kell, Mondino, Sosa, On quotients of spaces with Ricci curvature bounded below], but due to the repulsive nature of orientifold singularities they fall outside this class, even for negative N (cf. Sec.6.4.1).
2) Regarding infinitesimal Hilbertianity of intersecting O6 solution, to perform a detailed supergravity analysis in the spirit of the results in this paper would require to know their complete backreaction, which is currently not known. However, in general, infinitesimal Hilbertianity is a weak enough assumption that we expect it to hold also for those cases. For example, as we stress in Proposition 6.4 and Remark 6.5, this condition holds when the physical singularity is of measure zero. We also thank the referee for encouraging us to explore how the matter content on the localized sources affects the entropy comparison.
3) In terms of the volume modulus, the solution is at a minimum. Computing the masses corresponding to deformations that make the torus not square is indeed a good idea, and we thank the referee for it. However, upon reflection a more precise computation of the Casimir stress energy tensor as function of general moduli seems to us a bit challenging and out of the scope of the present paper, in particular given that parametric estimates might not suffice since tachyons of the scale of the cosmological constant could be allowed if above the BF bound. We will extend the discussion on the paper stressing these points, to which we hope to come back in the future with a more precise treatment.

---

## Round 1 · Referee Report · Anonymous · 2023-4-24

Report
This points out a beautiful connection between the theory of optimal transport and the equations of motion governing gravity compactifications. The paper is well-written, clear, and points out a novel connection between two very different fields, thereby satisfying the publication requirements at SciPost. I therefore recommend publication without a doubt.
There are several minor improvements that could be implemented in the paper, at the author's discretion (I'm also happy if they leave as is):\begin{itemize}
\item The authors point out that O-planes are not RCD but Hilbertian. I wonder if bounds can be obtained by passing to the orientifold double cover, where the orientifold planes are smooth loci. A small comment/footnote in the introduction commenting on why this is/is not a good idea would be nice.
\item I regard (7.4) as a very important result of this paper, showing that the DGKT vacuum cannot be so warped by the O6's so as to lose scale separation. There are two comments on this. First, O6 solutions usually (but maybe not always) have intersectiong O6's. As it is commonplace in the community to believe this is the case, it would be useful to comment on whether the results apply to intersecting O6's. Are these infinitesimally Hilbertian? etc. Secondly, a possible caveat is that the intersecting O6's have a large amount of localized degrees of freedom that backreact in the geometry. Now this is a question that one cannot usually say anything about in supergravity. But the very original approach of the authors makes me hopeful that maybe someone can, and I encourage them to explore it. I mean since they rewrite Einstein's eqs. in terms of concavity of entropy, it makes sense that they could say something about the scenario where one tries to hide many, many degrees of freedom inside of a brane. More generally, can they use this angle to put upper bounds on e.g. the rank of the gauge group in a brane or other singular objects?
\item The Casimir vacuum in section 7.1 is beautiful, and I think should be given more prominence in the introduction etc. As the authors state, they did not check instabilities for $T^7$. However I believe that Casimir energies of $T^n$ are not too difficult to compute and it may be possible to find minima for special tori (like the ``cubic'' torus with all sides equal). Maybe the authors can quickly check the hessian there? It would certainly add a lot of solidity to the construction at a very small cost in effort.
\end{itemize}
To be clear, I do NOT require the authors to implement \textbf{any} of the above changes. They are just suggestions, the paper may be published as-is.

---

## Editorial Decision

published